# Beating Adversarial Low-Rank MDPs with Unknown Transition and Bandit Feedback

**Haolin Liu**[*]
University of Virginia
srs8rh@virginia.edu

**Zakaria Mhammedi**[*]
Google Research
mhammedi@google.com

**Chen-Yu Wei**[*]
University of Virginia
chenyu.wei@virginia.edu

**Julian Zimmert**[*]
Google Research
zimmert@google.com

## Abstract

We consider regret minimization in low-rank MDPs with fixed transition and adversarial losses. Previous work has investigated this problem under either full-information loss feedback with unknown transitions (Zhao et al., 2024), or bandit loss feedback with known transition (Foster et al., 2022). First, we improve the $\text{poly}(d, A, H)T^{5/6}$ regret bound of Zhao et al. (2024) to $\text{poly}(d, A, H)T^{2/3}$ for the full-information unknown transition setting, where $d$ is the rank of the transitions, $A$ is the number of actions, $H$ is the horizon length, and $T$ is the number of episodes. Next, we initiate the study on the setting with bandit loss feedback and unknown transitions. Assuming that the loss has a linear structure, we propose both model-based and model-free algorithms achieving $\text{poly}(d, A, H)T^{2/3}$ regret, though they are computationally inefficient. We also propose oracle-efficient model-free algorithms with $\text{poly}(d, A, H)T^{4/5}$ regret. We show that the linear structure is necessary for the bandit case—without structure on the reward function, the regret has to scale polynomially with the number of states. This is contrary to the full-information case (Zhao et al., 2024), where the regret can be independent of the number of states even for unstructured reward function.

## 1 Introduction

We study online reinforcement learning (RL) in low-rank Markov Decision Processes (MDPs). Low-rank MDPs is a class of MDPs where the transition probability can be decomposed as an inner product between two low-dimensional features, i.e., $P(x' \mid x, a) = \phi^\star(x, a)^\top \mu^\star(x')$, where $P(x' \mid x, a)$ is the probability of transitioning to state $x'$ when the learner takes action $a$ on state $x$, and $\phi^\star$, $\mu^\star$ are two feature mappings. The ground truth features $\phi^\star$ and $\mu^\star$ are unknown to the learner. This setting has recently caught theoretical attention due to its simplicity and expressiveness (Agarwal et al., 2020; Uehara et al., 2021; Zhang et al., 2022a; Cheng et al., 2023; Modi et al., 2024; Zhang et al., 2022b; Mhammedi et al., 2024a; Huang et al., 2023). In particular, since the learner does not know the features, it is necessary for the learner to perform *feature learning* (or *representation learning*) to approximate them. This allows low-rank MDPs to model the additional difficulty not present in traditional linear function approximation schemes where the features are given, such as in linear MDPs (Jin et al., 2020b) and in linear mixture MDPs (Ayoub et al., 2020). Since feature learning is an indispensable part of modern deep RL pipelines, low-rank MDP is a model that is closer to practice than traditional linear function approximation.

---

[*]The authors are listed in alphabetical order.

38th Conference on Neural Information Processing Systems (NeurIPS 2024).

Table 1: Comparison of adversarial low-rank MDP algorithms. $\mathcal{O}$ here hides factors of order $\text{poly}(d, |\mathcal{A}|, \log T, \log |\Phi\|\Upsilon|)$. $^\dagger$Algorithm 5 assumes access to $\phi(x, a)$ for any $(\phi, x, a) \in \Phi \times \mathcal{X} \times \mathcal{A}$, while other algorithms only require access to $\phi(x, a)$ for any $(\phi, a) \in \Phi \times \mathcal{A}$ on *visited* $x$.

| Feedback | Algorithm | Algorithm type | Regret | Efficiency | Loss |
|---|---|---|---|---|---|
| Full-info | Zhao et al. (2024) | Model-based | $\mathcal{O}(T^{5/6})$ | Oracle-efficient | Arbitrary |
| | Algorithm 1 | Model-based | $\mathcal{O}(T^{2/3})$ | Oracle-efficient | Arbitrary |
| | **Lower Bound** | | $\Omega(\sqrt{|\mathcal{A}|T})$ | | Arbitrary |
| Bandit | Algorithm 2 | Model-based | $\mathcal{O}(T^{2/3})$ | Inefficient | Linear loss Unknown loss feature |
| | Algorithm 5$^\dagger$ | Model-free | $\mathcal{O}(T^{2/3})$ | Inefficient | Linear loss Unknown loss feature |
| | Algorithm 3 (for oblivious adversary) | Model-free | $\mathcal{O}(T^{4/5})$ | Oracle-efficient | Linear loss Unknown loss feature |
| | Algorithm 4 (for adaptive adversary) | Model-free | $\mathcal{O}(T^{4/5})$ | Oracle-efficient | Linear loss Known loss feature |
| | **Lower Bound** | | $\Omega(\sqrt{|\mathcal{X}\|\mathcal{A}|T})$ | | Arbitrary |

Most prior theoretical work on low-rank MDPs focuses on reward-free learning; this is a setting where instead of focusing on a particular reward function, the goal is to learn a model for the transitions (or, in the model-free setting, a small set of policies with good state cover), that enables policy optimization for *any* downstream reward functions. While this is a reasonable setup in some cases, in other applications, the learner can only obtain loss information from interactions with the environment, and only observes the loss on the state-actions that have been visited (i.e., bandit feedback). This introduces additional challenges to the learner.

Furthermore, in many online learning scenarios, the loss function may change over time, reflecting the non-stationary nature of the environment or task switches (Padakandla et al., 2020). This could be modeled by the *adversarial* MDP setting, where the loss function changes arbitrarily from one episode to the next, and the changes might even depend on the behavior of the learner. This setting is also extensively studied, but mostly restricted to tabular MDPs (Rosenberg and Mansour, 2019; Jin et al., 2020a; Shani et al., 2020; Luo et al., 2021) or traditional linear function approximation schemes (Cai et al., 2020; Luo et al., 2021; He et al., 2022; Zhao et al., 2022; Sherman et al., 2023b; Dai et al., 2023; Liu et al., 2023). The work by Zhao et al. (2024) initiated the study on adversarial MDPs in low-rank MDPs, but their work is restricted to full-information loss feedback.

When feature learning, bandit feedback, and adversarial losses are combined, the problem becomes highly challenging, and to the best of our knowledge their are no provably efficient algorithms to tackle this setting. In this work, we provide the first result for this combination. We hope that our result would bring new ideas to RL in practice, where all three elements are usually present simultaneously. We give several main results, targeting at either tighter regret (i.e., the performance gap between the optimal policy and the learner) or computational efficiency, as summarized in Table 1. Below we give a brief introduction for each of them. A more thorough related work review is in Appendix A.

- $T^{2/3}$-**regret algorithm under full-information feedback (Algorithm 1).** This setting is studied by the only prior work in adversarial low-rank MDPs (Zhao et al., 2024), and we greatly improve their $T^{5/6}$ regret bound to $T^{2/3}$. Our algorithm begins with a model-based initial exploration phase to estimate the transition. It then performs policy optimization where the critic is the $Q$ value induced by the estimated transition and the full information loss.

- $T^{2/3}$-**regret model-based/model-free inefficient algorithm under bandit feedback (Algorithm 2, Algorithm 5).** Algorithm 2 starts with a model-based initial exploration phase to learn an estimated transition, and then runs exponential weights over policy space for regret minimization in the second phase. To tackle bandit feedback, we construct a novel loss estimator that leverages the structure of low-rank MDP to perform accurate off-policy evaluation. Algorithm 5 starts with a

different exploration phase, where it calls VoX (Mhammedi et al., 2023) to learn a policy cover; VoX is a model-free, reward-free exploration algorithm. After this initial exploratory phase, the algorithm also applies exponential weights and utilizes the same loss estimator as in Algorithm 2. However, due to its model-free nature, certain components of the estimator cannot be directly accessed and must be derived through specific optimizations.

- $T^{4/5}$-**regret model-free oracle-efficient algorithm under bandit feedback (Algorithm 3, Algorithm 4).** Algorithm 3 also starts with the model-free exploration algorithm VoX (Mhammedi et al., 2023) to learn a policy cover. After that, the algorithm operates in epochs; during epoch $k$, the algorithm commits to a fixed mixture of policies. This mixture consists of certain exploratory policies (based on the policy cover from the initial phase) and a policy computed using an online learning algorithm based on estimated $Q$-functions from previous epochs (these serve as loss functions). Algorithm 4 deals with the much more challenging setting of an *adaptive* adversary with bandit feedback. Here, we make the additional assumption that the loss feature, which may be different from the feature of the low-rank decomposition, is given. The algorithm is similar to Algorithm 3 with key differences outlined in Section 4.3.

## 2    Preliminaries

We study the episodic online reinforcement learning setting with horizon $H$. We consider an MDP $\mathcal{M} = (\mathcal{X}, \mathcal{A}, P^\star_{1:H})$, where $\mathcal{X}$ represents a countable (possibly infinite) state space[2], $\mathcal{A}$ is a finite action space, and $P^\star_h : \mathcal{X} \times \mathcal{A} \to \Delta(\mathcal{X})$ denotes the transition kernel from layer $h$ to $h + 1$. We assume that the initial state $x_1 \in \mathcal{X}$ is fixed for simplicity without loss of generality. For any policy $\pi : \mathcal{X} \mapsto \Delta(\mathcal{A})$ and arbitrary set of transition kernels $\{P_h\}_{h \in [H]}$, we let $\mathbb{P}^{P,\pi}$ denote the law over $(\boldsymbol{x}_1, \boldsymbol{a}_1, \ldots, \boldsymbol{x}_H, \boldsymbol{a}_H)$ induced by the process of setting $\boldsymbol{x}_1 = x_1$, sampling $\boldsymbol{a}_1 \sim \pi_1(\cdot \mid \boldsymbol{x}_1)$, then for $h = 2, \ldots, H$, $\boldsymbol{x}_h \sim P_{h-1}(\cdot \mid \boldsymbol{x}_{h-1}, \boldsymbol{a}_{h-1})$ and $\boldsymbol{a}_h \sim \pi_h(\cdot \mid \boldsymbol{x}_h)$. We let $\mathbb{E}^{P,\pi}$ denote the corresponding expectations. Further, we let $d^{P,\pi}_h(x) \coloneqq \mathbb{P}^{P,\pi}[\boldsymbol{x}_h = x]$ denote the *occupancy* of $x \in \mathcal{X}$. We also let $d^{P,\pi}_h(x,a) \coloneqq \mathbb{P}^{P,\pi}[\boldsymbol{x}_h = x, \boldsymbol{a}_h = a]$. Further, we let $\mathbb{E}^\pi = \mathbb{E}^{P^\star,\pi}$, $\mathbb{P}^\pi = \mathbb{P}^{P^\star,\pi}$, and $d^\pi_h = d^{P^\star,\pi}_h$. We use $\pi \circ_h \pi'$ to denote a policy that follows $\pi_k(\cdot \mid \cdot)$ for $k < h$ and $\pi'_k(\cdot \mid \cdot)$ for $k \geq h$. Similarly, $\pi \circ_h \pi' \circ_{h'} \pi''$ denotes a policy that follows $\pi_k$ for $k < h$, $\pi'_k$ for $h \leq k < h'$ and $\pi''_k$ for $k > h'$.

We consider a learner interacting with the MDP $\mathcal{M}$ for $T$ episodes with adversarial loss functions. Before the game starts, an oblivious adversary chooses the loss functions for all episodes ($\ell^t_{1:H} : \mathcal{X} \times \mathcal{A} \to [0,1])^T_{t=1}$. For each episode $t \in [T]$, the learner starts at state $\boldsymbol{x}^t_1 = x_1$, then for each step $h \in [H]$ within episode $t$, the learner observes state $\boldsymbol{x}^t_h \in \mathcal{X}_h$, chooses an action $\boldsymbol{a}^t_h \in \mathcal{A}$, then suffers loss $\ell^t_h(\boldsymbol{x}^t_h, \boldsymbol{a}^t_h)$. The state $\boldsymbol{x}^t_{h+1}$ at the next step is drawn from transition $P^\star_h(\cdot \mid \boldsymbol{x}^t_h, \boldsymbol{a}^t_h)$. We consider *bandit feedback* setting where the learner could only observe the losses $\ell^t_1(\boldsymbol{x}^t_1, \boldsymbol{a}^t_1), \ldots, \ell^t_H(\boldsymbol{x}^t_H, \boldsymbol{a}^t_H)$ at the visited state-action pairs.

We let $\Pi \coloneqq \{\pi : \mathcal{X} \to \Delta(\mathcal{A})\}$ denote the set of Markovian policies. For policy $\pi \in \Pi$, loss $\ell$ and transition kernels $P_{1:H}$, we denote by $Q^{P,\pi}_h(\cdot, \cdot; \ell)$ the *state-action* value function (a.k.a. $Q$-function) at step $h \in [H]$ with respect to the transitions $P_{1:H}$ and loss $\ell$; that is

$$Q^{P,\pi}_h(x, a; \ell) \coloneqq \mathbb{E}^{P,\pi}\left[\sum_{s=h}^H \ell_s(\boldsymbol{x}^t_s, \boldsymbol{a}^t_s) \mid \boldsymbol{x}_h = x, \boldsymbol{a}_h = a\right], \tag{1}$$

for all $(x, a) \in \mathcal{X} \times \mathcal{A}$. We let $V^{P,\pi}_h(x; \ell) \coloneqq \max_{a \in \mathcal{A}} Q^{P,\pi}_h(x, a; \ell)$ be the corresponding *state* value function at layer $h$. Further, we write $Q^\pi_h(\cdot, \cdot; \ell) \coloneqq Q^{P^\star,\pi}_h(\cdot, \cdot; \ell)$ and $V^\pi_h(\cdot; \ell) \coloneqq V^{P^\star,\pi}(\cdot; \ell)$.

For all of our algorithms except for Algorithm 4, we aim to construct (possibly randomized) policies $\{\boldsymbol{\pi}^t\}_{t \in [T]}$ that ensure a sublinear *pseudo-regret* with respect to the best-fixed policy; that is,

$$\mathrm{Reg}_T \coloneqq \min_{\pi \in \Pi} \mathrm{Reg}_T(\pi) \quad \text{where} \quad \mathrm{Reg}_T(\pi) \coloneqq \mathbb{E}\left[\sum_{t=1}^T V^{\boldsymbol{\pi}^t}_1(x_1; \ell^t) - \sum_{t=1}^T V^\pi_1(x_1; \ell^t)\right]. \tag{2}$$

For Algorithm 4, we bound the standard *regret*

$$\overline{\mathrm{Reg}}_T \coloneqq \sum_{t=1}^T V^{\boldsymbol{\pi}^t}_1(x_1; \ell^t) - \min_{\pi \in \Pi} \sum_{t=1}^T V^\pi_1(x_1; \ell^t) \tag{3}$$

---

[2]We assume that $\mathcal{X}$ is countable only to simplify the presentation. Our results can easily be extended to a continuous state space with an appropriate measure-theoretic treatment (see e.g. Mhammedi et al. (2023)).

with high probability. This allows it to handle adaptive adversary.

Throughout, we will assume that the MDP $\mathcal{M}$ is low-rank with *unknown* feature maps $\phi_h^\star$ and $\mu_h^\star$.

**Assumption 2.1** (Low-Rank MDP). *There exist (unknown) features maps $\phi_{1:H}^\star : \mathcal{X} \times \mathcal{A} \to \mathbb{R}^d$ and $\mu_{1:H}^\star : \mathcal{X} \to \mathbb{R}^d$, such that for all $h \in [H-1]$ and $(x, a, x') \in \mathcal{X} \times \mathcal{A} \times \mathcal{X}$:*

$$\mathbb{P}[\boldsymbol{x}_{h+1} = x' \mid \boldsymbol{x}_h = x, \boldsymbol{a}_h = a] = \phi_h^\star(x, a)^\top \mu_{h+1}^\star(x'). \tag{4}$$

*Furthermore, for all $h \in [H]$, the feature maps $\mu_h^\star$ and $\phi_h^\star$ are such that $\sup_{(x,a) \in \mathcal{X} \times \mathcal{A}} \|\phi_h^\star(x,a)\| \le 1$ and $\|\sum_{x \in \mathcal{X}} g(x) \cdot \mu_h^\star(x)\| \le \sqrt{d}$, for all $g : \mathcal{X} \to [0,1]$.*

**Loss function under bandit feedback.** For bandit feedback setting, we make the following additional linear assumption on the losses; in the sequel, we will argue that this is necessary to avoid a sample complexity scaling with the number of states. This linear loss assumption also appears in Ren et al. (2022); Zhang et al. (2022a) for stochastic low-rank MDPs. Note that for the full-information feedback setting, such an assumption is not required.

**Assumption 2.2** (Loss Representation). *For any $t \in [T]$ and layer $h$, there is a vector $g_h^t \in \mathbb{B}_d(1)$ such that the loss $\ell_h^t(x, a)$ at round $t$ satisfies:*

$$\forall (x, a) \in \mathcal{X} \times \mathcal{A}, \quad \ell_h^t(x, a) = \phi_h^\star(x, a)^\top g_h^t. \tag{5}$$

We note that there is no loss of generality in assuming that the losses are expressed using the same features $\phi_{1:H}^\star$ as the low-rank structure in (4). This is because if the losses have different features, we can simply combine these features with the low-rank features, and redefine $\phi^\star$ accordingly. For the bulk of our results (and as stated in the prequel), we will assume that the losses $\{\ell_h^t(\cdot, \cdot)\}_{h \in [H], t \in [T]}$ (or equivalently $\{g_h^t\}_{h \in [H], t \in [T]}$ under Assumption 2.2) are chosen by an aversary before the start of the game (i.e. oblivious adversary). In Section 4.3, we will present a model-free, oracle-efficient algorithm for an adaptive adversary.

**Function approximation.** So far, Assumption 2.1 and Assumption 2.2 are in line with assumptions made in the linear MDP setting (Jin et al., 2020b). However, unlike in linear MDPs, we do not assume that the feature maps $\phi_{1:H}^\star$ are known. To facilitate representation learning and ultimately a sublinear regret, we need to make *realizability* assumptions. In particular, in the model-free setting, we assume we have a function class $\Phi$ that contains the true features $\phi_{1:H}^\star$. In the model-based setting, we additionally assume access to a function class $\Upsilon$ that contains the feature maps $\mu_{1:H}^\star$[3]. We will formalize these assumptions in their corresponding sections in the sequel.

**Other notation.** For $\psi : \Pi \to \mathbb{R}^d$, we define $\texttt{John}(\psi, \Pi)$ as a distribution $\mu \in \Delta(\Pi)$ such that $\|\psi(\pi)\|_{G^{-1}}^2 \le d$ for all $\pi \in \Pi$, where $G = \sum_{\pi \in \Pi} \mu(\pi) \cdot \psi(\pi)\psi(\pi)^\top$. This is the standard John's exploration or $G$-optimal design, which always exists.

# 3 Model-based Algorithms for Adversarial Low-rank MDPs

In this section, we discuss adversarial low-rank MDPs under model-based assumption. The model-based assumption is formalized in the Assumption 3.1 below. This assumption is standard which also appears in prior works on model-based learning in low-rank MDPs (Agarwal et al., 2020; Uehara et al., 2021; Zhang et al., 2022a; Cheng et al., 2023; Zhao et al., 2024).

**Assumption 3.1** (Model-based assumption). *The learner has access to two model spaces $\Phi$ and $\Upsilon$ such that $\phi^\star \in \Phi$ and $\mu^\star \in \Upsilon$. Moreover, for any $\phi \in \Phi$, $\mu \in \Upsilon$, and $h \in [2..H]$, we have $\sup_{(x,a) \in \mathcal{X} \times \mathcal{A}} \|\phi_{h-1}(x,a)\| \le 1$, $\sum_{x' \in \mathcal{X}} \phi_{h-1}(x,a)^\top \mu_h(x') = 1$ and $\|\sum_{x \in \mathcal{X}} g(x) \cdot \mu_h(x)\| \le \sqrt{d}$, for all $g : \mathcal{X} \to [0,1]$.*

## 3.1 Adversarial Low-rank MDPs with Full Information

We first discuss learning adversarial low-rank MDPs with full information and model-based assumption. This setting aligns with Zhao et al. (2024), and our Algorithm 1 successfully improves their regret from $T^{5/6}$ to $T^{2/3}$.

---

[3]The setting where we assume access to function classes that realize both $\phi_{1:H}^\star$ and $\mu_{1:H}^\star$ is called *model-based* because it allows one to model the transition probabilities, thanks to the low-rank structure in (4).

---

**Algorithm 1** Model-Based Algorithm for Full-Information Feedback

---

1: Let $\eta = \frac{1}{H\sqrt{T}}$, $\epsilon = \left(Hd^2|\mathcal{A}|(d^2 + |\mathcal{A}|)\right)^{\frac{1}{3}} T^{-\frac{1}{3}}$, and $T_0 = \widetilde{\mathcal{O}}(\epsilon^{-2} H^3 d^2 |\mathcal{A}|(d^2 + |\mathcal{A}|))$. Let $\pi^1$ be a uniform policy.

2: Run (Cheng et al., 2023, Algorithm 1) for $T_0$ episodes and get outputs $\hat{\phi} \in \Phi$, $\hat{\mu} \in \Upsilon$.

3: Define transitions $\widehat{P}_{1:H-1}$ as

$$\widehat{P}_h(x' \mid x, a) = \hat{\phi}_h(x, a)^\top \hat{\mu}_{h+1}(x'), \quad \forall (x, a, x') \in \mathcal{X} \times \mathcal{A} \times \mathcal{X}.$$

4: **for** $t = T_0 + 1, T_0 + 2, \ldots, T$ **do**

5:     Execute policy $\pi^t$ and observe trajectory $(\boldsymbol{x}_{1:H}, \boldsymbol{a}_{1:H})$ and full information loss $\ell^t$.

6:     Update policy for all $h \in [H]$:

$$\pi_h^{t+1}(a \mid x) \propto \exp\left(-\eta \sum_{i=1}^t \widehat{Q}_h^i(x, a)\right) \text{ where } \widehat{Q}_h^t(x, a) = Q_h^{\widehat{P}, \pi^t}(x, a; \ell^t).$$

7: **end for**

---

As argued in Zhao et al. (2024), the challenge of learning adversarial low-rank MDPs lies in the need for balancing exploration and exploitation both in representation learning and policy optimization over adversarial losses. To tackle this doubled exploration and exploitation challenge, the algorithm of Zhao et al. (2024) performs *simultaneous* representation learning and policy optimization. With a closer look at their analysis, we find that there is a drawback of this approach: because their algorithm handles the two tasks at the same time, it spends less exploration for representation learning in the early phase of the algorithm. This results in larger error in the estimated Q-values (i.e., critic) fed to policy optimization, and worsens the overall regret.

To address this issue, we design Algorithm 1 as a simple two-phase algorithm that separates representation learning and policy optimization. In the first phase, following Cheng et al. (2023), we perform optimal reward-free exploration for low-rank MDPs to estimate the transition. The resulted estimator, $\widehat{P}$, is able to accurately approximate the true transition and give accurate Q-value estimators for any policy. The more accurate Q-estimator allows for more effective policy optimization in the second phase. Theorem 3.1 shows the guarantee of Algorithm 1 where $\widetilde{\mathcal{O}}$ hides logarithmic factors of $d, H, T, |\Phi\|\Upsilon|$.

**Theorem 3.1.** *Algorithm 1 ensures* $\mathrm{Reg}_T \le \widetilde{\mathcal{O}}\left(H^3 \left(d^2 + |\mathcal{A}|\right) T^{\frac{2}{3}}\right)$.

The proof for Theorem 3.1 is given in Appendix B. Zhao et al. (2024) also constructs a lower bound $\Omega\left(H\sqrt{d|\mathcal{A}|T}\right)$ for this settings. Thus, the $\mathrm{poly}(|\mathcal{A}|)$-dependence is unavoidable.

### 3.2 Model-Based, Computationally Inefficient Algorithm for Bandit Feedback

In this section, following Assumption 3.1, we introduce the first (model-based) algorithm (Algorithm 2) for adversarial low-rank MDPs with bandit feedback and sublinear regret. Compared with linear MDPs, the key challenge for more general low-rank MDPs is to construct a proper loss estimator. For linear MDP, since the feature is known, the loss estimator closely resembles that of linear bandits. However, low-rank MDPs lack such structural simplicity, making standard loss estimators invalid. To overcome this challenge, we propose a new loss estimator that works for any loss function based on off-policy evaluation and the low-rank structure of transition. In this section, $\widetilde{\mathcal{O}}$ hides logarithmic factors of $d, H, T, |\Phi\|\Upsilon|$.

In Algorithm 2, we first conduct an initial representation learning phase to establish accurate transition estimator $\widehat{P}$ and its corresponding features $\hat{\phi}$ and $\hat{\mu}$ based on reward-free exploration algorithms in Cheng et al. (2023). Then, in the second phase, we use exponential weights to maintain a distribution over the policy space $\Pi'$ where we mix a uniform policy with $\Pi$ to enhance exploration. At every round $t$, a behavior policy $\pi^t$ is chosen from the current policy distribution, and we use the data collected by $\pi^t$ to estimate the value for every $\pi \in \Pi'$. The success of such off-policy evaluation is based on the following observations of low-rank MDP. Using the low-rank transition structure, for $h \ge 2$, we have

$$\forall \pi \in \Pi, \quad d_h^\pi(x) = \phi_{h-1}^\star(\pi)^\top \mu_h^\star(x), \quad \text{where} \quad \phi_{h-1}^\star(\pi) := \mathbb{E}^\pi[\phi_{h-1}^\star(\boldsymbol{x}_{h-1}, \boldsymbol{a}_{h-1})]. \tag{6}$$

**Algorithm 2** Model-Based Algorithm for Bandit Feedback

**Input:** A policy class $\Pi$.

1: Set $\epsilon = T^{-\frac{1}{3}}$, $\gamma = T^{-\frac{1}{3}}$, $\beta = T^{-\frac{1}{3}}$, $\eta = (4Hd|\mathcal{A}|)^{-1}T^{-\frac{2}{3}}$, and $T_0 = \widetilde{\mathcal{O}}(\epsilon^{-2}H^3d^2|\mathcal{A}|(d^2 + |\mathcal{A}|))$.

2: Run (Cheng et al., 2023, Algorithm 1) for $T_0$ episodes and get outputs $\hat{\phi} \in \Phi, \hat{\mu} \in \Upsilon$.

3: Define transitions $\widehat{P}_{1:H-1}$ as

$$\widehat{P}_h(x' \mid x, a) = \hat{\phi}_h(x,a)^\top \hat{\mu}_{h+1}(x'), \quad \forall(x, a, x') \in \mathcal{X} \times \mathcal{A} \times \mathcal{X}.$$

4: For all $h \in [H-1]$, define $\hat{\phi}_h(\pi) = \sum_{(x,a)\in\mathcal{X}\times\mathcal{A}} \hat{d}_h^\pi(x,a) \cdot \hat{\phi}_h(x,a)$, where $\hat{d}_h^\pi \coloneqq d_h^{\widehat{P},\pi}$.

5: Define the policy space $\Pi' = \{\pi' : \exists \pi \in \Pi, \ \pi_h'(\cdot \mid x) = (1-\beta)\pi_h(\cdot \mid x) + \beta/|\mathcal{A}|, \ \forall x, h\}$.

6: **for** $t = T_0 + 1, T_0 + 2, \ldots, T$ **do**

7:     Define $p^t(\pi) \propto \exp\left(-\eta \sum_{i=1}^{t-1}\left(\hat{\ell}^i(\pi) - b^i(\pi)\right)\right)$, for all $\pi \in \Pi'$.

8:     Let $\rho^t(\pi) = (1-\gamma)p^t(\pi) + \frac{\gamma}{H-1}\sum_{h=1}^{H-1}J_h$, where $J_h = \texttt{John}(\hat{\phi}_h(\cdot), \Pi')$. `// John as in §2`

9:     Execute policy $\boldsymbol{\pi}^t \sim \rho^t$ and observe trajectory $(\boldsymbol{x}_{1:H}^t, \boldsymbol{a}_{1:H}^t)$ and losses $\boldsymbol{\ell}_h^t = \ell_h^t(\boldsymbol{x}_h^t, \boldsymbol{a}_h^t)$.

10:     Define $\Sigma_h^t = \sum_{\pi\in\Pi'} \rho^t(\pi) \cdot \hat{\phi}_h(\pi)\hat{\phi}_h(\pi)^\top$, $b^t(\pi) = \sqrt{d}H\epsilon \cdot \sum_{h=1}^{H-1} \|\hat{\phi}_h(\pi)\|_{(\Sigma_h^t)^{-1}}$, and

$$\hat{\ell}^t(\pi) = \frac{\pi_1(\boldsymbol{a}_1^t \mid \boldsymbol{x}_1^t)}{\boldsymbol{\pi}_1^t(\boldsymbol{a}_1^t \mid \boldsymbol{x}_1^t)}\boldsymbol{\ell}_1^t + \sum_{h=2}^H \hat{\phi}_{h-1}(\pi)^\top \left(\Sigma_{h-1}^t\right)^{-1} \hat{\phi}_{h-1}(\boldsymbol{\pi}^t)\frac{\pi_h(\boldsymbol{a}_h^t \mid \boldsymbol{x}_h^t)}{\boldsymbol{\pi}_h^t(\boldsymbol{a}_h^t \mid \boldsymbol{x}_h^t)}\boldsymbol{\ell}_h^t.$$

11: **end for**

Thus, using the definition of the $V$-function from Section 2, we have for any loss function $\ell$ and $\pi$:

$$V_1^\pi(x_1; \ell) - \mathbb{E}[\ell_1(\boldsymbol{x}_1, \boldsymbol{a}_1)] = \sum_{h=2}^H \sum_{(x,a)\in\mathcal{X}\times\mathcal{A}} \phi_{h-1}^\star(\pi)^\top \mu_h^\star(x)\pi_h(a \mid x) \cdot \ell_h(x,a). \tag{7}$$

Letting $\Lambda_h^t \coloneqq (\mathbb{E}_{\boldsymbol{\pi}^t\sim p^t}[\phi_h^\star(\boldsymbol{\pi}^t)\phi_h^\star(\boldsymbol{\pi}^t)^\top])^{-1}$ and ingnoring the loss term $\mathbb{E}[\ell_1(\boldsymbol{x}_1, \boldsymbol{a}_1)]$ from the first step (this term can easily be treated as in a bandit setting with $H = 1$), we have for all $\pi \in \Pi'$:

$$V_1^\pi(x_1; \ell) = \mathbb{E}_{\boldsymbol{\pi}^t\sim p^t}\left[\sum_{h=2}^H \sum_{(x,a)\in\mathcal{X}\times\mathcal{A}} \phi_{h-1}^\star(\pi)^\top \Lambda_{h-1}^t \phi_{h-1}^\star(\boldsymbol{\pi}^t)\phi_{h-1}^\star(\boldsymbol{\pi}^t)^\top \mu_h^\star(x)\pi_h(a \mid x) \cdot \ell_h(x,a)\right],$$

$$= \mathbb{E}_{\boldsymbol{\pi}^t\sim p^t}\left[\sum_{h=2}^H \sum_{x,a} \phi_{h-1}^\star(\pi)^\top \Lambda_{h-1}^t \phi_{h-1}^\star(\boldsymbol{\pi}^t) \cdot d_h^{\boldsymbol{\pi}^t}(x)\boldsymbol{\pi}_h^t(a \mid x)\frac{\pi_h(a \mid x)}{\boldsymbol{\pi}_h^t(a \mid x)} \cdot \ell_h(x,a)\right],$$

$$= \mathbb{E}_{\boldsymbol{\pi}^t\sim p^t}\mathbb{E}^{\boldsymbol{\pi}^t}\left[\sum_{h=2}^H \phi_{h-1}^\star(\pi)^\top \Lambda_{h-1}^t \phi_{h-1}^\star(\boldsymbol{\pi}^t) \cdot \frac{\pi_h(\boldsymbol{a}_h \mid \boldsymbol{x}_h)}{\boldsymbol{\pi}_h^t(\boldsymbol{a}_h \mid \boldsymbol{x}_h)} \cdot \ell_h(\boldsymbol{x}_h, \boldsymbol{a}_h)\right].$$

Thus, for all $\ell$ and $\pi$, $\sum_{h=2}^H \phi_{h-1}^\star(\pi)^\top \Lambda_{h-1}^t \phi_{h-1}^\star(\boldsymbol{\pi}^t) \cdot \pi_h(\boldsymbol{a}_h \mid \boldsymbol{x}_h)\boldsymbol{\pi}_h^t(\boldsymbol{a}_h \mid \boldsymbol{x}_h)^{-1} \cdot \ell_h(\boldsymbol{x}_h, \boldsymbol{a}_h)$ for $\boldsymbol{\pi}^t \sim p^t$ and $(\boldsymbol{x}_h, \boldsymbol{a}_h) \sim d_h^{\boldsymbol{\pi}^t}$ is an unbiased estimator of $V_1^\pi(x_1, \ell)$. However, $\phi_{h-1}^\star(\pi)$ is not accessible because both the true feature $\phi^\star$ and occupancy $d^\pi$ for the true transition are unknown. Thus, our estimator incorporates the learned feature $\hat{\phi}$ and the occupancy of $\widehat{P}$ instead as shown in Line 4 and Line 10. Utilizing estimated features and transition could introduce additional bias but the initial representation learning already ensures such bias is small enough to tackle. We compensate for the bias by incorporating an exploration bonus $b^t(\pi)$ in exponential weights. To further encourage exploration, we additionally perform John's exploration together with exponential weights when selecting behavior policies. The main guarantee of Algorithm 2 is given in Theorem 3.2.

**Theorem 3.2.** *Algorithm 2 achieves* $\text{Reg}_T(\pi) \le \widetilde{\mathcal{O}}\left(d^2H^3|\mathcal{A}|(d^2 + |\mathcal{A}|)T^{2/3}\log|\Pi|\right)$ *for any* $\pi \in \Pi$.

Note that the guarantee in Theorem 3.2 only holds for policy $\pi \in \Pi$. To ensure our regret bound is meaningful, at least a near-optimal policy should be contained in the given policy set $\Pi$. In general, the size of such a policy set would grow exponentially with the number of states (e.g. covering of all Markovian policies), making the regret have polynomial dependence on the number of states. In Theorem 3.3, we show that even for low-rank MDPs, if the loss function lacks structure, the regret cannot avoid polynomial dependence on the number of states. The detailed construction for this lower-bound is given in Appendix H.

**Theorem 3.3.** *There exists a low-rank MDP with $|\mathcal{X}|$ states, $|\mathcal{A}|$ actions and sufficiently large $T$ with unstructured losses such that any agent suffers at least regret of $\Omega(\sqrt{|\mathcal{X}||\mathcal{A}|T})$.*

Theorem 3.3 shows that under bandit feedback, in general, we could not gain too much from low-rank transition structure compared with tabular MDPs. This contrasts with the $\Omega(\sqrt{|\mathcal{A}|T})$ lower bound in the full information settings (Zhao et al. (2024)). To get rid of any dependence on the number of states, we additionally introduce Assumption 2.2 to impose linear structure on the loss function. Unlike linear MDPs that require the loss feature to be known, our algorithm can even handle linear loss with unknown feature, since our Algorithm 2 never explicitly uses the loss feature. The linear structure is only used to control the size of the candidate policy class in the analysis (i.e., making $\log|\Pi|$ irrelevant to the number of states). Specifically, when both loss and transition are linear, the Q-function is also linear, making it sufficient to consider the following linear policy space:

$$\Pi_{\text{lin}} = \left\{ \pi : \mathcal{X} \to \mathcal{A} \;\middle|\; \pi_h(a \mid x) = \mathbb{I}\left\{a = \operatorname*{argmin}_{a \in \mathcal{A}} \phi_h(x,a)^\top \theta_h\right\}, \; h \in [H], \; \|\theta_h\|_2 \le \sqrt{d}HT, \; \phi \in \Phi \right\}.$$

The $\frac{1}{T}$-cover of $\Pi_{\text{lin}}$ only have size $|\Phi| \cdot T^{\mathcal{O}(d)}$ following standard arguments (e.g, Exercise 27.6 of Lattimore and Szepesvári (2020)) and if we feed it into Algorithm 2, our regret could avoid dependece on the size of state space as shown in Corollary 3.1.

**Corollary 3.1.** *If the loss function satisfies Assumption 2.2, applying Algorithm 2 with $\Pi$ as the $\frac{1}{T}$-cover of $\Pi_{\text{lin}}$ ensures $\operatorname{Reg}_T \le \widetilde{\mathcal{O}}\left(d^3 H^3 |\mathcal{A}|(d^2 + |\mathcal{A}|)T^{2/3}\right)$.*

## 4  Model-free Algorithms for Adversarial Low-rank MDPs

In this section, we consider the model-free setting, where we only assume access to a feature class $\Phi$ that contains the true feature map $\phi^\star$.

**Assumption 4.1** (Model-free realizability). *The learner has access to a function class $\Phi$ such that*

$$\phi^\star \in \Phi \qquad and \qquad \sup_{\phi \in \Phi} \sup_{(x,a) \in \mathcal{X} \times \mathcal{A}} \|\phi(x,a)\| \le 1. \tag{8}$$

This is a standard assumption in the context of model-free RL (Modi et al., 2024; Zhang et al., 2022b; Mhammedi et al., 2024a). We note that having access to the function class $\Phi$ alone (instead of both $\Phi$ and $\Upsilon$ as in Assumption 3.1) is not sufficient to model the transition probabilities (unlike in the model-based case). This makes the model-free setting much more challenging; in fact, until the recent work by Mhammedi et al. (2023) there were no model-free, oracle-efficient algorithms for this setting that do not require any additional structural assumptions on the MDP.

### 4.1  Model-free, Inefficient Algorithm for Bandit Feedback

Our first algorithm follows the same structure as Algorithm 2 but incorporates a model-free initial exploration phase introduced by Mhammedi et al. (2023). Unlike the model-based exploration phase, which directly provides an estimated transition, the model-free exploration phase outputs a policy cover. This policy cover can be combined with the optimization in Algorithm 1 of Liu et al. (2023) to solve the expected feature $\hat{\phi}$, which is then used in the loss estimator in Line 10 of Algorithm 2. The algorithm, summarized in Algorithm 5, is inefficient but achieves $T^{2/3}$ regret. More details and proofs can be found in Appendix D.

### 4.2  Model-free, Oracle Efficient Algorithm for Bandit Feedback (Oblivious Adversary)

We now descibe the key component of our efficient model-free algorithm (Algorithm 3).

**Exploration phase and policy cover.**   Similar to algorithms in the previous section, Algorithm 3 begins with a reward-free exploratorion phase; Line 2 of Algorithm 3. However, unlike in the previous sections where the role of this exploration phase was to learn a model for the transition probabilities, here the goal is to compute a, so called, *policy cover* which is a small set of policies that can be used to effectively explore the state space.

**Definition 4.1** (Approximate policy cover). *For $\alpha, \varepsilon \in (0,1]$ and $h \in [H]$, a subset $\Psi \subseteq \Pi$ is an $(\alpha, \varepsilon)$-policy cover for layer $h$ if*

$$\max_{\pi \in \Psi} d_h^\pi(x) \ge \alpha \cdot \max_{\pi' \in \Pi} d_h^{\pi'}(x), \quad \text{for all } x \in \mathcal{X} \text{ such that } \max_{\pi' \in \Pi} d_h^{\pi'}(x) \ge \varepsilon \cdot \|\mu_h^\star(x)\|. \tag{10}$$

**Algorithm 3** Oracle Efficient Algorithm for Adversarial Low-Rank MDPs (Oblivious Adversary).

---

**Input:** Number of rounds $T$, feature class $\Phi$, confidence parameter $\delta \in (0, 1)$.

1: Set $\varepsilon \leftarrow T^{-1/3}$, $N_{\text{reg}} \leftarrow T^{2/3}$, $\nu \leftarrow N_{\text{reg}}^{-1/2}$, and $T_0 \leftarrow \varepsilon^{-2} A d^{13} H^6 \log(\Phi/\delta)$.

2: Get $\Psi_{1:H}^{\text{cov}} \leftarrow \text{VoX}(\Phi, \varepsilon, \delta)$.     `// Compute policy cover with VoX (Mhammedi et al., 2023).`

3: **for** $k = 1, \ldots, (T - T_0)/N_{\text{reg}}$ **do**

4:      Define $\widehat{\pi}_h^{(k)}(a \mid x) \propto \exp\left(-\eta \sum_{s<k} \widehat{Q}_h^{(s)}(x, a)\right)$ for $h \in [H]$.

5:      **for** $t = T_0 + (k-1) \cdot N_{\text{reg}} + 1, \ldots, T_0 + k \cdot N_{\text{reg}}$ **do**

6:          Sample variables $\zeta^t \sim \text{Ber}(\nu)$, $\boldsymbol{h}^t \sim \text{unif}([H])$, and $\boldsymbol{\pi}^t \sim \text{unif}(\Psi_{\boldsymbol{h}^t}^{\text{cov}})$.

7:          Set $\widehat{\boldsymbol{\pi}}^t = \mathbb{I}\{\zeta^t = 0\} \cdot \widehat{\pi}^{(k)} + \mathbb{I}\{\zeta^t = 1\} \cdot \boldsymbol{\pi}^t \circ_{\boldsymbol{h}^t} \pi_{\text{unif}} \circ_{\boldsymbol{h}^t + 1} \widehat{\pi}^{(k)}$.

8:          Execute $\widehat{\boldsymbol{\pi}}^t$, and observe trajectory $(\boldsymbol{x}_1^t, \boldsymbol{a}_1^t, \ldots, \boldsymbol{x}_H^t, \boldsymbol{a}_H^t)$.

9:          For $h \in [H]$, observe loss $\boldsymbol{\ell}_h^t := \ell_h^t(\boldsymbol{x}_h^t, \boldsymbol{a}_h^t)$.

10:      **end for**

11:      For $h \in [H]$ and $\mathcal{I}^{(k)} = \{T_0 + (k-1) \cdot N_{\text{reg}} + 1, \ldots, T_0 + k \cdot N_{\text{reg}}\}$, compute $(\hat{\phi}_h^{(k)}, \hat{\theta}_h^{(k)})$:

$$(\hat{\phi}_h^{(k)}, \hat{\theta}_h^{(k)}) \leftarrow \underset{(\phi, \theta) \in \Phi \times \mathbb{B}_d(H\sqrt{d})}{\text{argmin}} \sum_{t \in \mathcal{I}^{(k)}} \left( \phi_h(\boldsymbol{x}_h^t, \boldsymbol{a}_h^t)^\top \theta - \sum_{s=h}^{H} \boldsymbol{\ell}_s^t \right)^2 \cdot \mathbb{I}\{\zeta^t = 0 \text{ or } \boldsymbol{h}^t \leq h\}. \quad (9)$$

12:      Set $\widehat{Q}_h^{(k)}(x, a) = \hat{\phi}_h^{(k)}(x, a)^\top \hat{\theta}_h^{(k)}$, for all $(x, a) \in \mathcal{X} \times \mathcal{A}$.

13: **end for**

---

In Line 2, Algorithm 3 calls VoX (Mhammedi et al., 2023), a reward-free and model-free exploration algorithm to compute $(1/(8Ad), \varepsilon)$-policy covers $\Psi_1^{\text{cov}}, \ldots, \Psi_H^{\text{cov}}$ for layers $1, \ldots, H$, respectively, with $|\Psi_h| = d$ for all $h \in [H]$. This call to VoX requires $O(1/\varepsilon^2)$ episodes; see the guarantee of VoX in Lemma G.1. After this initial phase, the algorithm operates in epochs, each consisting of $N_{\text{reg}} \in \mathbb{N}$ episodes, where in each epoch $k \in [K]$, the algorithm commits to executing policies sampled from a fixed policy distribution $\rho^{(k)} \in \Delta(\Pi)$ with support on the policy covers $\Psi_{1:H}^{\text{cov}}$ and a policy $\widehat{\pi}^{(k)}$ specified by an online learning algorithm. Next, we describe in more detail how $\rho^{(k)}$ is constructed and motivate the elements of its construction starting with the online learning policies $\{\widehat{\pi}^{(k)}\}_{k \in [K]}$.

**Online learning policies.** Given estimates $\{\widehat{Q}_{1:H}^{(s)}\}_{s<k}$ of the average $Q$-functions

$$\left\{ \frac{1}{N_{\text{reg}}} \sum_{t \text{ in epoch } s} Q_{1:H}^{\widehat{\pi}^{(s)}}(\cdot, \cdot; \ell^t) \right\}_{s<k} \quad (11)$$

from the previous epoch (we will describe how these estimates are computed in the sequel), Algorithm 3 computes policy $\widehat{\pi}^{(k)}$ for epoch $k$ according to

$$\widehat{\pi}_h^{(k)}(a \mid x) \propto \exp\left( -\eta \sum_{s<k} \widehat{Q}_h^{(s)}(x, a) \right), \quad (12)$$

for all $h \in [H]$. Given a state $x \in \mathcal{X}$, such exponential weight update ensures a sublinear regret with respect to the sequence of loss functions given by $\{\pi(\cdot \mid x) \mapsto \sum_{h \in [H]} \widehat{Q}_h^{(k)}(x, \pi_h(\cdot \mid x))\}_{k \in [K]}$. Thanks to the performance difference lemma, and as shown in Luo et al. (2021), a sublinear regret with respect to these "surrogate" loss functions translates into a sublinear regret in the low-rank MDP game we are interested in, granted that $\{\widehat{Q}_{1:H}^{(k)}\}_{k \in [K]}$ are good estimates of the average $Q$-functions (Luo et al., 2021). In line with previous analyses, we require the $Q$-function estimates to ensure that the following bias term

$$\mathbb{E}^\pi \left[ \max_{a \in \mathcal{A}} \left( \frac{1}{N_{\text{reg}}} \sum_{t \text{ in epoch } k} Q_h^{\widehat{\pi}^{(k)}}(\boldsymbol{x}_h, a; \ell^t) - \widehat{Q}_h^{(k)}(\boldsymbol{x}_h, a) \right)^2 \right] \quad (13)$$

is small for all $h \in [H]$, $k \in [K]$, and $\pi \in \Pi$.

*Q*-function estimates.    Thanks to the low-rank MDP structure and the linear loss representation assumption (Assumption 2.2), the avegare *Q*-functions in (11) are linear in the feature maps $\phi^\star$. Thus, using the function class $\Phi$ in Assumption 2.1 we can estimate these average *Q*-functions by regressing the sum of losses $\sum_{s=h}^{H} \ell_s^t$ onto $(\boldsymbol{x}_h^t, \boldsymbol{a}_h^t)$ for $t$ in the $k$th epoch (as in (9)). However, naïvely doing this using only trajectories generated by $\widehat{\pi}^{(k)}$ would only ensure that the bias term in (13) is small for $\pi = \widehat{\pi}^{(k)}$. To ensure that it is small for all possible policies $\pi$'s, we need to estimate the *Q*-functions on the trajectories of policies that are guaranteed to have good state coverage; this is where we use the policy cover from the initial phase.

**Mixture of policies.**    At episode $t$ in each epoch $k \in [K]$, we execute policy $\widehat{\boldsymbol{\pi}}^t$ sampled from $\rho^{(k)}$, where $\rho^{(k)}$ is the distribution of the random policy:

$$\mathbb{I}\{\boldsymbol{\zeta}^t = 0\} \cdot \widehat{\pi}^{(k)} + \mathbb{I}\{\boldsymbol{\zeta}^t = 1\} \cdot \boldsymbol{\pi}^t \circ_{\boldsymbol{h}^t} \pi_{\texttt{unif}} \circ_{\boldsymbol{h}^t+1} \widehat{\pi}^{(k)}, \tag{14}$$

with $\boldsymbol{\zeta}^t \sim \mathrm{Ber}(\nu)$, $\boldsymbol{h}^t \sim \texttt{unif}([H])$, and $\boldsymbol{\pi}^t \sim \texttt{unif}(\Psi_{\boldsymbol{h}^t}^{\texttt{cov}})$. In words, at the start of each episode of any epoch $k$, we execute $\widehat{\pi}^{(k)}$ (see (12)) with probability $1 - \nu$; and with probability $\nu$, we execute a policy in $\Psi_{1:H}^{\texttt{cov}}$ selected uniformly at random. As explained in the previous paragraph, this ensures a small bias for all choices of $\pi$ in (13) thanks the policy cover property of $\Psi_{1:H}^{\texttt{cov}}$. We now state the guarantee of Algorithm 3.

**Theorem 4.1.** *Let $\delta \in (0,1)$ be given and suppose Assumption 2.1 and Assumption 2.2 hold. This, for $T = \mathrm{poly}(A, d, H, \log(|\Phi|/\delta))$ sufficiently large, Algorithm 3 guarantees $\mathrm{Reg}_T \leq \mathrm{poly}(A, d, H, \log(|\Phi|/\delta)) \cdot T^{4/5}$ regret against an oblivious adversary.*

The proof is in Appendix E. Note that the $T$-dependence in this regret even outperforms that of the previous best bound by Zhao et al. (2024) (see Table 1). Compared to their algorithm, Algorithm 3 is model-free and only requires bandit feedback. This makes the result in Theorem 4.1 rather surprising.

### 4.3    Model-free, Oracle Efficient Algorithm (Adaptive Adversary)

In this section, we present a variant of Algorithm 3 (Algorithm 4) that guarantees a sublinear regret against an adaptive adversary. Given the difficulty of this setting, we make the additional assumption that the algorithm has access to the loss feature $\phi^{\mathrm{loss}}$, which may be different than the low-rank MDP feature $\phi^\star$ (unlike in Assumption 2.2).

**Assumption 4.2** (Loss Representation)**.** *There is a (known) feature map $\phi^{\mathrm{loss}}$ satisfying $\sup_{h \in [H], (x,a) \in \mathcal{X} \times \mathcal{A}} \|\phi_h^{\mathrm{loss}}(x,a)\| \leq 1$ and such that for any round $h \in [H]$, $t \in [T]$, and history $\mathcal{H}^{t-1} = (x_{1:H}^{1:t-1}, a_{1:H}^{1:t-1})$, the loss function at round $t$ satisfies*

$$\forall (x,a) \in \mathcal{X} \times \mathcal{A}, \quad \ell_h(x, a; \mathcal{H}^{t-1}) = \phi_h^{\mathrm{loss}}(x,a)^\top g_h^t, \tag{15}$$

*for some $g_h^t \in \mathbb{B}_d(1)$.*

Note that Assumption 4.2 asserts that the loss at round $t$ depends only on the history $\mathcal{H}^{t-1}$ and the current state action pair. Before moving forward, we introduce some additional notation we will use throughout this section.

**Additional notation.**    For any two feature maps $\phi, \psi : \mathcal{X} \times \mathcal{A} \to \mathbb{R}^d$, we denote by $[\phi, \psi] : \mathcal{X} \times \mathcal{A} \to \mathbb{R}^{2d}$ the vertical concatenation of the two feature maps. For any $h \in [H]$, $t \in [T]$, policy $\pi \in \Pi$, and history $\mathcal{H}^{t-1} = (x_{1:H}^{1:t-1}, a_{1:H}^{1:t-1})$, we denote by $Q_h^\pi(\cdot, \cdot; \mathcal{H}^{t-1})$ the *Q*-function at layer $h$ corresponding to rollout policy $\pi$; that is,

$$Q_h^\pi(x, a; \mathcal{H}^{t-1}) \coloneqq \mathbb{E}^\pi \left[ \sum_{s=h}^{H} \ell_s(\boldsymbol{x}_s, \boldsymbol{a}_s; \mathcal{H}^{t-1}) \mid \boldsymbol{x}_h = x, \boldsymbol{a}_h = a \right]. \tag{16}$$

Finally, we let $V_h^\pi(x; \mathcal{H}^{t-1}) \coloneqq \max_{a \in \mathcal{A}} Q_h^\pi(x, a; \mathcal{H}^{t-1})$ denote the corresponding *V*-function.

Algorithm 4 is similar to Algorithm 3 with the following key differences; after computing a policy cover, the algorithm calls `RepLearn` (a representation learning algorithm initially introduced by Modi et al. (2024) and subsequently refined by Mhammedi et al. (2023)) to compute a feature map $\phi^{\mathrm{rep}}$. Then, for every $h \in [H]$, the algorithm computes a *spanner*; a set of policies $\Psi_h^{\mathrm{span}} = \{\pi_{h,1}, \ldots, \pi_{h,2d}\}$ that act as an approximate spanner for the set $\{\mathbb{E}^\pi[\phi_h^{\mathrm{rep}}(\boldsymbol{x}_h, \boldsymbol{a}_h), \phi_h^{\mathrm{loss}}(\boldsymbol{x}_h, \boldsymbol{a}_h)] : \pi \in \Pi\} \subseteq \mathbb{R}^{2d}$, where we use $[\cdot, \cdot]$ to denote the vertical concatenation of vectors. These spanner

policies are then used as the exploratory policies after the initial phase; that is, at episode $t$ in each epoch $k \in [K]$, we execute policy $\boldsymbol{\pi}^{(k)}$ sampled from $\rho^{(k)}$, where $\rho^{(k)}$ is set to be the distribution of the random policy: $\mathbb{I}\{\boldsymbol{\zeta}^t = 0\} \cdot \widehat{\pi}^{(k)} + \mathbb{I}\{\boldsymbol{\zeta}^t = 1\} \cdot \boldsymbol{\pi}^t \circ_{\boldsymbol{h}^t+1} \widehat{\pi}^{(k)}$, with $\boldsymbol{\zeta}^t \sim \mathrm{Ber}(\nu)$, $\boldsymbol{h}^t \sim \mathrm{unif}([H])$, and $\boldsymbol{\pi}^t \sim \mathrm{unif}(\Psi_{\boldsymbol{h}^t}^{\mathrm{span}})$. Here, the main difference to Algorithm 3 (see also (47)) is that we use $\boldsymbol{\pi}^t \sim \mathrm{unif}(\Psi_{\boldsymbol{h}^t}^{\mathrm{span}})$ instead of $\boldsymbol{\pi}^t \sim \mathrm{unif}(\Psi_{\boldsymbol{h}^t}^{\mathrm{cov}})$. We require these spanner policies instead of policies in the policy cover, as an adaptive adversary's history-dependent losses prevent standard least squares regression due to the lack of permutation invariance of state-action pairs across episodes within an epoch. Estimating the Q-functions is thus more complex, and we approach it in expectation over roll-ins using policies in $\Psi^{\mathrm{span}}$, the "in-expectation" estimation task is in a sense easier.

We now state the guarantee of Algorithm 4.

**Theorem 4.2.** *Let $\delta \in (0,1)$ be given and suppose that Assumption 2.1 and Assumption 4.2 hold. Then, for $T = \mathrm{poly}(A, d, H, \log(|\Phi|/\delta))$ sufficiently large, Algorithm 4 guarantees with probability at least $1 - \delta$,*

$$\sum_{t \in [T]} V_h^{\boldsymbol{\pi}^t}(x_1; \mathcal{H}^{t-1}) - \min_{\pi \in \Pi} \sum_{t \in [T]} V_h^{\pi}(x_1; \mathcal{H}^{t-1}) \leq \mathrm{poly}(A, d, H, \log(|\Phi|/\delta)) \cdot T^{4/5}, \quad (17)$$

*where $\boldsymbol{\pi}^t$ is the policy that Algorithm 4 executes at episode $t \in [T]$.*

---

**Algorithm 4** Oracle Efficient Algorithm for Adversarial Low-Rank MDPs (Adaptive Adversary).

---

**Input:** Number of rounds $T$, feature class $\Phi$, loss feature $\phi^{\mathrm{loss}}$, confidence parameter $\delta \in (0,1)$.

1: Set $\varepsilon \leftarrow T^{-1/3}$, $N_{\mathrm{reg}} \leftarrow T^{2/3}$, $\nu \leftarrow N_{\mathrm{reg}}^{-1/4}$, and $\alpha \leftarrow (8Ad)^{-1}$, $T_{\mathrm{cov}} \leftarrow \varepsilon^{-2} A d^{13} H^6 \log(\Phi/\delta)$.

2: Set $T_{\mathrm{rep}} \leftarrow \alpha^{-1} \varepsilon^{-2} A H \log(|\Phi|/\delta)$, $T_{\mathrm{span}} \leftarrow \alpha^{-2} \varepsilon^{-2} A \log(dH|\Phi|\varepsilon^{-1}\delta^{-1})$,

3: Define $\mathcal{F}_h = \left\{ (x,a) \mapsto \max_{a \in \mathcal{A}} \bar{\phi}_h(x,a)^\top \bar{\theta} \mid \bar{\phi}_h = [\phi_h^{\mathrm{loss}}, \phi_h], \phi \in \Phi, \bar{\theta} \in \mathbb{B}_{2d}(1) \right\}$, $\forall h \in [H]$.

4: Get $\Psi_{1:H}^{\mathrm{cov}} \leftarrow \mathrm{VoX}(\Phi, \varepsilon, \delta/4)$.

5: Get $\phi_h^{\mathrm{rep}} \leftarrow \mathrm{RepLearn}(h, \mathcal{F}_{h+1}, \Phi, \mathrm{unif}(\Psi_h^{\mathrm{cov}}), T_{\mathrm{rep}})$, for all $h \in [H-1]$. // RepLearn as in Mhammedi et al. (2023)

6: For all $h \in [H]$, set $\bar{\phi}_h^{\mathrm{rep}} \leftarrow [\phi_h^{\mathrm{loss}}, \phi_h^{\mathrm{rep}}] \in \mathbb{R}^{2d}$.

7: For $h \in [H]$, set $\Psi_h^{\mathrm{span}} \leftarrow \mathrm{Spanner}(h, \Phi, \Psi_{1:h}^{\mathrm{cov}}, \bar{\phi}_h^{\mathrm{rep}}, T_{\mathrm{span}})$.      // Algorithm 7

8: Set $T_0 \leftarrow T_{\mathrm{cov}} + T_{\mathrm{rep}} + T_{\mathrm{span}}$.

9: **for** $k = 1, \ldots, (T - T_0)/N_{\mathrm{reg}}$ **do**

10:     Define $\widehat{\pi}_h^{(k)}(a \mid x) \propto \exp\left( -\eta \sum_{s<k} \widehat{Q}_h^{(s)}(x,a) \right)$ for $h \in [H]$.

11:     **for** $t = T_0 + (k-1) \cdot N_{\mathrm{reg}} + 1, \ldots, T_0 + k \cdot N_{\mathrm{reg}}$ **do**

12:         Define the random variables $\boldsymbol{\zeta}^t \sim \mathrm{Ber}(\nu)$, $\boldsymbol{h}^t \sim \mathrm{unif}([H])$, and $\boldsymbol{\pi}^t \sim \mathrm{unif}(\Psi_{\boldsymbol{h}^t}^{\mathrm{span}})$.

13:         Set $\widehat{\boldsymbol{\pi}}^t = \mathbb{I}\{\boldsymbol{\zeta}^t = 0\} \cdot \widehat{\pi}^{(k)} + \mathbb{I}\{\boldsymbol{\zeta}^t = 1\} \cdot \boldsymbol{\pi}^t \circ_{\boldsymbol{h}^t+1} \widehat{\pi}^{(k)}$.

14:         Execute $\widehat{\boldsymbol{\pi}}^t$, and observe trajectory $(\boldsymbol{x}_1^t, \boldsymbol{a}_1^t, \ldots, \boldsymbol{x}_H^t, \boldsymbol{a}_H^t)$.

15:         For $h \in [H]$, observe loss $\boldsymbol{\ell}_h^t := \ell_h(\boldsymbol{x}_h^t, \boldsymbol{a}_h^t; \mathcal{H}^{t-1})$, where $\mathcal{H}^{t-1} := (\boldsymbol{x}_{1:H}^{1:t-1}, \boldsymbol{a}_{1:H}^{1:t-1})$.

16:     **end for**

17:     For $h \in [H]$ and $\mathcal{I}^{(k)} = \{T_0 + (k-1) \cdot N_{\mathrm{reg}} + 1, \ldots, T_0 + k \cdot N_{\mathrm{reg}}\}$, compute $\hat{\theta}_h^{(k)}$ such that

$$\hat{\theta}_h^{(k)} \leftarrow \operatorname*{argmin}_{\theta \in \mathbb{B}_{2d}(4Hd^2)} \sum_{\pi \in \Psi_h^{\mathrm{span}}} \left| \sum_{t \in \mathcal{I}^{(k)}} \mathbb{I}\{\boldsymbol{h}^t = h, \boldsymbol{\pi}^t = \pi, \boldsymbol{\zeta}^t = 1\} \cdot \left( \bar{\phi}_h^{\mathrm{rep}}(\boldsymbol{x}_h^t, \boldsymbol{a}_h^t)^\top \theta - \sum_{s=h}^H \boldsymbol{\ell}_s^t \right) \right| \quad (18)$$

18:     Set $\widehat{Q}_h^{(k)}(x,a) = \bar{\phi}_h^{\mathrm{rep}}(x,a)^\top \hat{\theta}_h^{(k)}$, for all $(x,a) \in \mathcal{X} \times \mathcal{A}$.

19: **end for**

---

## 5 Conclusion

In this paper, we focus on learning low-rank MDPs with unknown transitions and adversarial losses. For the full-information setting, we improve upon previous regret bounds. More importantly, we initiate the study of the challenging bandit feedback setting, developing various algorithms that achieve sublinear regret under different assumptions. However, the optimal $\sqrt{T}$ regret remains out of reach due to the limitations of our two-phase design. An interesting direction for future work is to perform on-the-fly representation learning to adapt to adversarial losses and achieve optimal regret.

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

# Appendices

# A    Related Work

**Learning low-rank MDPs in the stochastic setting.**    In the absent of adversarial losses, several general learning frameworks have been developed for super classes of low-rank MDPs which offer tight sample complexity for either reward-based (Jiang et al., 2017; Jin et al., 2021a; Du et al., 2021; Foster et al., 2021; Zhong et al., 2022) or reward-free (Chen et al., 2022b,a; Xie et al., 2022) settings. However, these algorithms require solving non-convex optimization problems on non-convex version spaces, making them computationally inefficiency. Oracle-efficient algorithms for low-rank MDPs are first obtained by Agarwal et al. (2020) using a model-based approach, and the sample complexity bound has been largely improved in subsequent works (Uehara et al., 2021; Zhang et al., 2022a; Cheng et al., 2023). The model-based approach, however, necessitates the function class to accurately model the transition, which is a strong requirement. To relax it, Modi et al. (2024); Zhang et al. (2022b) developed oracle-efficient model-free algorithms, but both of them require additional assumptions on the MDP structure. Recently, Mhammedi et al. (2024a) proposed a satisfactory model-free algorithm that removes all these assumptions. Our work leverages their techniques to tackle the more challenging adversarial setting.

**Learning adversarial MDPs.**    Learning adversarial tabular MDPs under bandit feedback and unknown transition has been extensively studied (Rosenberg and Mansour, 2019; Jin et al., 2020a; Lee et al., 2020; Jin et al., 2021b; Shani et al., 2020; Chen and Luo, 2021; Luo et al., 2021; Dai et al., 2022; Dann et al., 2023). This line of work has demonstrated not only $\sqrt{T}$ regret bounds but also several data-dependent bounds.

For adversarial MDPs with a large state space which necessitates the use of function approximation, if the transition is known, Foster et al. (2022) shows that adversarial setting is as easy as the stochastic setting even under general function approximation. For full-information loss feedback with unknown transition, $\sqrt{T}$ bound is derived for both linear mixture MDPs (Cai et al., 2020; He et al., 2022) and linear MDPs (Sherman et al., 2023a). For more challenging low-rank MDPs with unknown features, the best result only achieves $T^{5/6}$ regret (Zhao et al., 2024).

For function approximation with bandit feedback and unknown transition, Zhao et al. (2022) provides $\sqrt{T}$ bound for linear mixture MDPs, but their regret has polynomial dependence on the size of the state space due to the lack of structure on the loss function. For linear MDPs, a series of recent work has made significant progress in improving the regret bound (Luo et al., 2021; Dai et al., 2023; Sherman et al., 2023b; Kong et al., 2023; Liu et al., 2023). The state-of-the-art result by Liu et al. (2023) gives an inefficient algorithm with $\sqrt{T}$ regret and an efficient algorithm with $T^{3/4}$ regret. These regret bounds for linear MDPs do not depend on the state space size because of the linear loss assumption. We show in Appendix H that cross-state structure on the losses is necessary for low-rank MDPs with bandit feedback to achieve regret bound that do not scale with the number of states.

# B  Proof of Theorem 3.1 (Model-Based, Full Information)

**Theorem B.1** (Theorem 3 in Cheng et al. (2023))**.** *With probability $1 - \delta$, for any policy $\pi$ and layer $h$, Algorithm 1 in Cheng et al. (2023) outputs transition $\widehat{P}_{1:H}$ and features $\hat{\phi}_h$, $\hat{\mu}_h$ such that $\widehat{P}_h(x' \mid x, a) = \hat{\phi}_h(x, a)^\top \hat{\mu}_{h+1}(x')$ and*

$$\mathbb{E}^\pi \left[ \| \widehat{P}_h \left( \cdot \mid \boldsymbol{x}_h, \boldsymbol{a}_h \right) - P_h^\star \left( \cdot \mid \boldsymbol{x}_h, \boldsymbol{a}_h \right) \|_1 \right] \le \epsilon,$$

*if the number of collected trajectories is at least $\mathcal{O}\left( \frac{H^3 d^2 |\mathcal{A}|(d^2 + |\mathcal{A}|)}{\epsilon^2} \log^2 \left( TdH|\Phi\|\Upsilon| \right) \right)$.*

Define $\widehat{V}_1^\pi(x_1; \ell)$ as the value function of policy $\pi$ under transition $\{\widehat{P}_h\}_{h=1}^H$ and loss $\ell$. We have

$$
\begin{aligned}
\text{Reg}_T(\pi^\star) &= \sum_{t=1}^T V_1^{\pi^t}(x_1; \ell^t) - \sum_{t=1}^T V_1^{\pi^\star}(x_1; \ell^t) \\
&= \underbrace{\sum_{t=1}^T \left( V_1^{\pi^t}(x_1; \ell^t) - \widehat{V}_1^{\pi^t}(x_1; \ell^t) \right)}_{\textbf{Bias1}} + \underbrace{\sum_{t=1}^T \left( \widehat{V}_1^{\pi^\star}(x_1; \ell^t) - V_1^{\pi^\star}(x_1; \ell^t) \right)}_{\textbf{Bias2}} \\
&\quad + \underbrace{\sum_{t=1}^T \left( \widehat{V}_1^{\pi^t}(x_1; \ell^t) - \widehat{V}_1^{\pi^\star}(x_1; \ell^t) \right)}_{\textbf{FTRL}} + \mathcal{O}\left( \frac{H^3 d^2 |\mathcal{A}|(d^2 + |\mathcal{A}|)}{\epsilon^2} \log^2 \left( TdH|\Phi\|\Upsilon| \right) \right)
\end{aligned}
$$

**Bounding the bias term.** By Theorem B.1 and Lemma I.6, we have

$$\textbf{Bias1} + \textbf{Bias2} \le 2H^2 \epsilon T.$$

**Bounding the FTRL term.** Since $\eta = \frac{1}{H\sqrt{T}}$, we have

$$
\begin{aligned}
\textbf{FTRL} &\le \sum_{t=1}^T \sum_{h=1}^H \mathbb{E}^{\widehat{P}, \pi^\star} \left[ \langle \widehat{Q}_h^t(\boldsymbol{x}_h, \cdot), \pi_h^t(\cdot \mid \boldsymbol{x}_h) - \pi_h^\star(\cdot \mid \boldsymbol{x}_h) \rangle \right] && \text{(Lemma I.7)} \\
&\le \frac{H \log |\mathcal{A}|}{\eta} + \eta \sum_{h=1}^H \sum_{t=1}^T \mathbb{E}^{\widehat{P}, \pi^\star \circ_h \pi^t} \left[ \left( \widehat{Q}_h^t(\boldsymbol{x}_h, \boldsymbol{a}_h) \right)^2 \right], && \text{(Lemma I.5)} \\
&\le \frac{H \log |\mathcal{A}|}{\eta} + 2H^3 \eta T && (\widehat{Q}_h^t(\boldsymbol{x}_h, \boldsymbol{a}_h) \le H) \\
&= \mathcal{O}\left( H^2 \sqrt{T} \log |\mathcal{A}| \right).
\end{aligned}
$$

Thus, by setting $\epsilon = (Hd^2|\mathcal{A}|(d^2 + |\mathcal{A}|))^{\frac{1}{3}} T^{-\frac{1}{3}}$, we have

$$
\begin{aligned}
\text{Reg}_T &\le \mathcal{O}\left( \frac{H^3 d^2 |\mathcal{A}|(d^2 + |\mathcal{A}|)}{\epsilon^2} \log^2 \left( TdH|\Phi\|\Upsilon| \right) + 2H^2 \epsilon T + H^2 \sqrt{T} \log |\mathcal{A}| \right) \\
&\le \mathcal{O}\left( H^3 \left( d^2 + |\mathcal{A}| \right) T^{\frac{2}{3}} \log \left( |\mathcal{A}| + dH|\Phi\|\Upsilon|T \right) \right).
\end{aligned}
$$

## C Proof of Theorem 3.2 (Model-Based, Bandit Feedback)

**Lemma C.1.** *With $\widehat{P}_{1:H}$ as in Theorem B.1, we have for any $h \in [H]$ and any policy $\pi$,*

$$\sum_{x \in \mathcal{X}} \left| \hat{d}_h^\pi(x) - d_h^\pi(x) \right| \le \sum_{i=1}^{h-1} \mathbb{E}^\pi \left[ \|\widehat{P}_i(\cdot \mid \boldsymbol{x}_i, \boldsymbol{a}_i) - P_i(\cdot \mid \boldsymbol{x}_i, \boldsymbol{a}_i)\|_1 \right] \le (h-1) \cdot \epsilon.$$

*where $\hat{d}_h^\pi := d_h^{\widehat{P}, \pi}$.*

**Proof.** We prove the claim by induction. When $h = 1$, given that the $\|d_1^\pi - \hat{d}_1^\pi\|_1 = 0$. Assume

$$\sum_{x \in \mathcal{X}} \left| \hat{d}_h^\pi(x) - d_h^\pi(x) \right| \le \sum_{i=1}^{h-1} \mathbb{E}^\pi \left[ \|\widehat{P}_i(\cdot \mid \boldsymbol{x}_i, \boldsymbol{a}_i) - P_i(\cdot \mid \boldsymbol{x}_i, \boldsymbol{a}_i)\|_1 \right].$$

We have

$$\sum_{x \in \mathcal{X}_{h+1}} \left| \hat{d}_{h+1}^\pi(x) - d_{h+1}^\pi(x) \right|$$

$$= \sum_{x \in \mathcal{X}} \sum_{a \in \mathcal{A}} \sum_{x' \in \mathcal{X}_{h+1}} \left| \hat{d}_h^\pi(x)\pi_h(a \mid x) \cdot \widehat{P}_h(x'|x,a) - d_h^\pi(x)\pi_h(a \mid x) \cdot P_h^\star(x'|x,a) \right|,$$

$$\le \sum_{x \in \mathcal{X}} \sum_{a \in \mathcal{A}} \sum_{x' \in \mathcal{X}_{h+1}} \left| \hat{d}_h^\pi(x)\pi_h(a \mid x) \cdot \widehat{P}_h(x'|x,a) - d_h^\pi(x)\pi_h(a \mid x) \cdot \widehat{P}_h(x'|x,a) \right|$$

$$+ \sum_{x \in \mathcal{X}} \sum_{a \in \mathcal{A}} \sum_{x' \in \mathcal{X}_{h+1}} \left| d_h^\pi(x)\pi_h(a \mid x) \cdot \widehat{P}_h(x'|x,a) - d_h^\pi(x)\pi_h(a \mid x) \cdot P_h^\star(x'|x,a) \right|,$$

$$\le \sum_{x \in \mathcal{X}} \sum_{a \in \mathcal{A}} \sum_{x' \in \mathcal{X}_{h+1}} \widehat{P}_h(x'|x,a)\pi_h(a \mid x) \cdot |\hat{d}_h^\pi(x) - d_h^\pi(x)|$$

$$+ \sum_{x \in \mathcal{X}} \sum_{a \in \mathcal{A}} \sum_{x' \in \mathcal{X}_{h+1}} d_h^\pi(x)\pi_h(a \mid x) \cdot |\widehat{P}_h(x'|x,a) - P_h(x'|x,a)|,$$

$$\le \sum_{x \in \mathcal{X}} |\hat{d}_h^\pi(x) - d_h^\pi(x)| + \mathbb{E}^\pi \left[ \|\widehat{P}_h(\cdot \mid \boldsymbol{x}_h, \boldsymbol{a}_h) - P_h(\cdot \mid \boldsymbol{x}_h, \boldsymbol{a}_h)\|_1 \right],$$

$$\le \sum_{i=1}^{h} \mathbb{E}^\pi \left[ \|\widehat{P}_i(\cdot \mid \boldsymbol{x}_i, \boldsymbol{a}_i) - P_i(\cdot \mid \boldsymbol{x}_i, \boldsymbol{a}_i)\|_1 \right],$$

where the last step follows by the induction hypothesis. The second inequality of Lemma C.1 directly comes from Theorem B.1. $\qquad\square$

Our candidate policy space $\Pi'$ has a $\beta$ mixture of the random policy. For any deterministic policy $\pi_0^\star$, define policy $\pi^\star$ such that for any state $x \in \mathcal{X}$, we have $\pi^\star(\cdot \mid x) = (1 - \beta)\pi_0^\star(\cdot \mid x) + \frac{\beta}{|\mathcal{A}|}$. We have $\pi^\star \in \Pi'$. Define $\widehat{V}_1^\pi(x_1; \ell)$ as the value function of policy $\pi$ under transition $\{\widehat{P}_h\}_{h=1}^H$ and loss $\ell$.

For any policy $\pi_0^\star$, we have

$$\text{Reg}_T(\pi_0^\star) \tag{19}$$

$$= \mathbb{E}\left[ \sum_{t=1}^T \sum_\pi \rho^t(\pi) V_1^\pi(\boldsymbol{x}_1; \ell^t) - V_1^{\pi_0^\star}(\boldsymbol{x}_1; \ell^t) \right] + \mathcal{O}\left( \frac{H^3 d^2 |\mathcal{A}|(d^2 + |\mathcal{A}|)}{\epsilon^2} \log^2 (TdH|\Phi\|\Upsilon|) \right),$$

$$= \mathbb{E}\left[ \sum_{t=1}^T \sum_\pi p^t(\pi) V_1^\pi(\boldsymbol{x}_1; \ell^t) - V_1^{\pi^\star}(\boldsymbol{x}_1; \ell^t) \right] + \mathcal{O}\left( \frac{H^3 d^2 |\mathcal{A}|(d^2 + |\mathcal{A}|)}{\epsilon^2} \log^2 (TdH|\Phi\|\Upsilon|) \right) \tag{20}$$

$$+ \underbrace{\mathbb{E}\left[ \sum_{t=1}^T \sum_\pi \left( \rho^t(\pi) - p^t(\pi) \right) V_1^\pi(\boldsymbol{x}_1; \ell^t) \right]}_{\textbf{Error1}} + \underbrace{\mathbb{E}\left[ \sum_{t=1}^T V_1^{\pi^\star}(\boldsymbol{x}_1; \ell^t) - V_1^{\pi_0^\star}(\boldsymbol{x}_1; \ell^t) \right]}_{\textbf{Error2}},$$

$$= \underbrace{\mathbb{E}\left[ \sum_{t=1}^T \sum_\pi p^t(\pi) \left( V_1^\pi(\boldsymbol{x}_1; \ell^t) - \widehat{V}_1^\pi(\boldsymbol{x}_1; \ell^t) \right) \right]}_{\textbf{Bias1}} + \underbrace{\mathbb{E}\left[ \sum_{t=1}^T \widehat{V}_1^{\pi^\star}(\boldsymbol{x}_1; \ell^t) - V_1^{\pi^\star}(\boldsymbol{x}_1; \ell^t) \right]}_{\textbf{Bias2}}$$

$$+ \mathbb{E}\left[\underbrace{\sum_{t=1}^{T}\sum_{\pi} p^t(\pi)\widehat{V}_1^{\pi}(\boldsymbol{x}_1;\ell^t) - \widehat{V}_1^{\pi^{\star}}(\boldsymbol{x}_1;\ell^t)}_{\textbf{EXP}}\right] + \textbf{Error1} + \textbf{Error2}$$

$$+ \mathcal{O}\left(\frac{H^3 d^2 |\mathcal{A}|(d^2+|\mathcal{A}|)}{\epsilon^2}\log^2\left(TdH|\Phi\|\Upsilon|\right)\right). \tag{21}$$

Recall that $J = \mathrm{John}\{\hat{\phi}_h(\pi)\}_{\pi\in\Pi',h\in[H]} \in \Delta(\Pi'\times H)$, and we have $\rho^t(\pi) = (1-\gamma)p^t(\pi) + \gamma\sum_{h=1}^H J(\pi,h)$ where $p^t(\pi)$ is defined in Line 7 of Algorithm 2.

**Lemma C.2.** *We have*

$$\textbf{Error1} + \textbf{Error2} \le H\gamma T + 2H^2\beta T.$$

**Proof.**

$$\textbf{Error1} = \gamma\mathbb{E}\left[\sum_{t=1}^{T}\sum_{\pi}\left(\sum_{h=1}^H J(\pi,h) - p^t(\pi)\right)\cdot V_1^\pi(\boldsymbol{x}_1;\ell^t)\right] \le H\gamma T.$$

$$\textbf{Error2} = \sum_{t=1}^{T}\sum_{h=1}^H \mathbb{E}^{\pi^\star}\left[\sum_{a\in\mathcal{A}}|\pi_0^\star(a\mid\boldsymbol{x}_h) - \pi^\star(a\mid\boldsymbol{x}_h)|\cdot Q_h^{\pi_0^\star}(\boldsymbol{x}_h,a;\ell^t)\right] \le 2H^2\beta T,$$

where the last step follows by Lemma I.7. $\qquad\square$

**Lemma C.3.** *We have*

$$\textbf{Bias1} + \textbf{Bias2} \le H^2 T\epsilon.$$

**Proof.** This is a direct result combing Theorem B.1 and Lemma I.6. $\qquad\square$

For $(x_{1:H},a_{1:H}) \in \mathcal{X}^H\times\mathcal{A}^H$ and $\pi\in\Pi'$ where $\Pi'$ defined in Line 5 in Algorithm 2 is the mix of a given policy class $\Pi$ and a uniform policy, and is also the policy class we play with. Recall that the loss estimator

$$\hat{\ell}^t(\pi;\pi^t,x_{1:H},a_{1:H}) := \frac{\pi_1(a_1\mid x_1)}{\pi_1^t(a_1\mid x_1)}\ell_1^t(x_1,a_1)$$
$$+ \sum_{h=2}^H \hat{\phi}_{h-1}(\pi)^\top\left(\Sigma_{h-1}^t\right)^{-1}\hat{\phi}_{h-1}(\pi^t)\frac{\pi_h(a_h\mid x_h)}{\pi_h^t(a_h\mid x_h)}\ell_h^t(x_h,a_h). \tag{22}$$

defined in Line 10 of Algorithm 2.

**Lemma C.4.** *For any episode $t\in[T]$, for any policy $\pi$ we have*

$$\widehat{V}_1^\pi(x_1;\ell^t) \le \mathbb{E}_{\boldsymbol{\pi}^t\sim\rho^t}\mathbb{E}^{\boldsymbol{\pi}^t}\left[\hat{\ell}^t(\pi;\boldsymbol{\pi}^t,\boldsymbol{x}_{1:H},\boldsymbol{a}_{1:H})\right] + \sqrt{d}H\epsilon\sum_{h=2}^H\left\|\hat{\phi}_{h-1}(\pi)\right\|_{(\Sigma_{h-1}^t)^{-1}},$$

$$\widehat{V}_1^\pi(x_1;\ell^t) \ge \mathbb{E}_{\boldsymbol{\pi}^t\sim\rho^t}\mathbb{E}^{\boldsymbol{\pi}^t}\left[\hat{\ell}^t(\pi;\boldsymbol{\pi}^t,\boldsymbol{x}_{1:H},\boldsymbol{a}_{1:H})\right] - \sqrt{d}H\epsilon\sum_{h=2}^H\left\|\hat{\phi}_{h-1}(\pi)\right\|_{(\Sigma_{h-1}^t)^{-1}}.$$

**Proof.** First, from the definition in Line 4 of Algorithm 2, for any $x\in\mathcal{X}$ with $h\ge 2$, we have

$$\hat{d}_h^\pi(x) = \sum_{x',a'}\hat{d}_{h-1}^\pi(x',a')\cdot\widehat{P}_{h-1}(x\mid x',a') = \hat{\phi}_{h-1}(\pi)^\top\hat{\mu}_h(x)$$

where $\hat{\phi}_{h-1}(\pi) = \sum_{x',a'}\hat{d}_{h-1}^\pi(x',a')\hat{\phi}_{h-1}(x',a')$.

We now prove the first inequality:

$$\widehat{V}_1^\pi(x_1;\ell^t)$$

$$= \sum_{h=1}^{H} \sum_{x_h \in \mathcal{X}} \sum_{a_h \in \mathcal{A}} \hat{d}_h^{\pi}(x_h) \pi_h(a_h \mid x_h) \cdot \ell_h^t(x_h, a_h),$$

$$= \underbrace{\sum_{a_1 \in \mathcal{A}} \pi_1(a_1 \mid x_1) \cdot \ell_1^t(x_1, a_1)}_{\textbf{First}} + \underbrace{\sum_{h=2}^{H} \sum_{x_h \in \mathcal{X}} \sum_{a_h \in \mathcal{A}} \hat{\phi}_{h-1}(\pi)^\top \hat{\mu}_h(x_h) \pi_h(a_h \mid x_h) \cdot \ell_h^t(x_h, a_h)}_{\textbf{Remain}} \quad (23)$$

Through importance sampling, we have

$$\textbf{First} = \mathbb{E}_{\boldsymbol{\pi}^t \sim \rho^t} \mathbb{E}^{\boldsymbol{\pi}^t} \left[ \frac{\pi_1(\boldsymbol{a}_1 \mid \boldsymbol{x}_1)}{\pi_1^t(\boldsymbol{a}_1 \mid \boldsymbol{x}_1)} \cdot \ell_1^t(\boldsymbol{x}_1, \boldsymbol{a}_1) \right].$$

We now bound the remaining term in (42) Since $\Sigma_{h-1}^t = \mathbb{E}_{\boldsymbol{\pi}^t \sim \rho^t} \left[ \hat{\phi}_{h-1}(\boldsymbol{\pi}^t) \hat{\phi}_{h-1}(\boldsymbol{\pi}^t)^\top \right]$, we have

**Remain**

$$= \sum_{h=2}^{H} \sum_{x_h \in \mathcal{X}} \sum_{a_h \in \mathcal{A}} \hat{\phi}_{h-1}(\pi)^\top \left( \Sigma_{h-1}^t \right)^{-1} \mathbb{E}_{\boldsymbol{\pi}^t \sim \rho^t} \left[ \hat{\phi}_{h-1}(\boldsymbol{\pi}^t) \hat{\phi}_{h-1}(\boldsymbol{\pi}^t)^\top \right] \hat{\mu}_h(x_h) \pi_h(a_h \mid x_h) \ell_h^t(x_h, a_h),$$

$$= \mathbb{E}_{\boldsymbol{\pi}^t \sim \rho^t} \left[ \sum_{h=2}^{H} \sum_{x_h \in \mathcal{X}} \sum_{a_h \in \mathcal{A}} \hat{\phi}_{h-1}(\pi)^\top \left( \Sigma_{h-1}^t \right)^{-1} \hat{\phi}_{h-1}(\boldsymbol{\pi}^t) \underbrace{\hat{\phi}_{h-1}(\boldsymbol{\pi}^t)^\top \hat{\mu}_h(x_h)}_{\hat{d}_h^{\boldsymbol{\pi}^t}(x_h)} \pi_h(a_h \mid x_h) \ell_h^t(x_h, a_h) \right],$$

$$= \mathbb{E}_{\boldsymbol{\pi}^t \sim \rho^t} \left[ \sum_{h=2}^{H} \sum_{x_h \in \mathcal{X}} \sum_{a_h \in \mathcal{A}} \hat{\phi}_{h-1}(\pi)^\top \left( \Sigma_{h-1}^t \right)^{-1} \hat{\phi}_{h-1}(\boldsymbol{\pi}^t) \hat{d}_h^{\boldsymbol{\pi}^t}(x_h) \pi_h(a_h \mid x_h) \ell_h^t(x_h, a_h) \right],$$

$$= \mathbb{E}_{\boldsymbol{\pi}^t \sim \rho^t} \left[ \sum_{h=2}^{H} \sum_{x_h \in \mathcal{X}} \sum_{a_h \in \mathcal{A}} \hat{\phi}_{h-1}(\pi)^\top \left( \Sigma_{h-1}^t \right)^{-1} \hat{\phi}_{h-1}(\boldsymbol{\pi}^t) d_h^{\boldsymbol{\pi}^t}(x_h) \pi_h(a_h \mid x_h) \ell_h^t(x_h, a_h) \right]$$

$$+ \mathbb{E}_{\boldsymbol{\pi}^t \sim \rho^t} \left[ \sum_{h=2}^{H} \sum_{x_h \in \mathcal{X}} \sum_{a_h \in \mathcal{A}} \hat{\phi}_{h-1}(\pi)^\top \left( \Sigma_{h-1}^t \right)^{-1} \hat{\phi}_{h-1}(\boldsymbol{\pi}^t) \left( \hat{d}_h^{\boldsymbol{\pi}^t}(x_h) - d_h^{\boldsymbol{\pi}^t}(x_h) \right) \pi_h(a_h \mid x_h) \ell_h^t(x_h, a_h) \right],$$

$$(24)$$

$$\leq \mathbb{E}_{\boldsymbol{\pi}^t \sim \rho^t} \left[ \sum_{h=2}^{H} \sum_{x_h \in \mathcal{X}} \sum_{a_h \in \mathcal{A}} \hat{\phi}_{h-1}(\pi)^\top \left( \Sigma_{h-1}^t \right)^{-1} \hat{\phi}_{h-1}(\boldsymbol{\pi}^t) d_h^{\boldsymbol{\pi}^t}(x_h, a_h) \frac{\pi_h(a_h \mid x_h)}{\pi_h^t(a_h \mid x_h)} \ell_h^t(x_h, a_h) \right]$$

$$+ \sum_{h=2}^{H} \left\| \hat{\phi}_{h-1}(\pi) \right\|_{(\Sigma_{h-1}^t)^{-1}} \mathbb{E}_{\boldsymbol{\pi}^t \sim \rho^t} \left[ \left\| \hat{\phi}_{h-1}(\boldsymbol{\pi}^t) \right\|_{(\Sigma_{h-1}^t)^{-1}} \right] \sum_{x_h \in \mathcal{X}} \left| \hat{d}_h^{\boldsymbol{\pi}^t}(x_h) - d_h^{\boldsymbol{\pi}^t}(x_h) \right|, \quad (25)$$

$$\leq \mathbb{E}_{\boldsymbol{\pi}^t \sim \rho^t} \mathbb{E}^{\boldsymbol{\pi}^t} \left[ \sum_{h=2}^{H} \hat{\phi}_{h-1}(\pi)^\top \left( \Sigma_{h-1}^t \right)^{-1} \hat{\phi}_{h-1}(\boldsymbol{\pi}^t) \frac{\pi_h(\boldsymbol{a}_h \mid \boldsymbol{x}_h)}{\pi_h^t(\boldsymbol{a}_h \mid \boldsymbol{x}_h)} \ell_h^t(\boldsymbol{x}_h, \boldsymbol{a}_h) \right]$$

$$+ \sqrt{d} H \epsilon \sum_{h=2}^{H} \left\| \hat{\phi}_{h-1}(\pi) \right\|_{(\Sigma_{h-1}^t)^{-1}}, \quad (26)$$

where (25) follows by Cauchy-Schwarz, and the last inequality uses Lemma C.1. Adding up **First** and **Remain**, and using the defintion of $\hat{\ell}$ in (22) implies the first inequality of the lemma.

The second inequality follows the same procedure except for applying Cauchy-Schwarz inequality in the opposite direction in Eq. (24). $\qquad\square$

**Lemma C.5.** If $\eta \leq \left( \frac{2|\mathcal{A}|Hd}{\beta\gamma} + dH^2 \frac{\epsilon}{\sqrt{\gamma}} \right)^{-1}$, then

$$\textbf{EXP} \leq \frac{\log |\Pi|}{\eta} + \frac{6d\eta H^2 |\mathcal{A}| T}{\beta} + 4\eta d^2 H^4 \epsilon^2 T + 4dH^2 \epsilon T,$$

where **EXP** is as in (21).

**Proof.** Recall in Line 10 of Algorithm 2, $b^t(\pi) \coloneqq \sqrt{d} H \epsilon \sum_{h=2}^{H} \left\| \hat{\phi}_{h-1}(\pi) \right\|_{(\Sigma_{h-1}^t)^{-1}}$, for all $\pi \in \Pi'$. By Lemma C.4, we have

$$\textbf{EXP} = \mathbb{E} \left[ \sum_{t=1}^{T} \sum_{\pi} p^t(\pi) \widehat{V}_1^{\pi}(\boldsymbol{x}_1; \ell^t) - \widehat{V}_1^{\pi^\star}(\boldsymbol{x}_1; \ell^t) \right]$$

$$\leq \sum_{t=1}^{T} \sum_{\pi} p^t(\pi) \mathbb{E}_{\boldsymbol{\pi}^t \sim \rho^t} \mathbb{E}^{\boldsymbol{\pi}^t} \left[ \hat{\ell}^t(\pi; \boldsymbol{\pi}^t, \boldsymbol{x}_{1:H}, \boldsymbol{a}_{1:H}) \right] - \sum_{t=1}^{T} \mathbb{E}_{\boldsymbol{\pi}^t \sim \rho^t} \mathbb{E}^{\boldsymbol{\pi}^t} \left[ \hat{\ell}^t(\pi^\star; \boldsymbol{\pi}^t, \boldsymbol{x}_{1:H}, \boldsymbol{a}_{1:H}) \right]$$

$$+ \sqrt{d} H \epsilon \sum_{t=1}^{T} \sum_{\pi} p^t(\pi) \sum_{h=2}^{H} \left\| \hat{\phi}_{h-1}(\pi) \right\|_{(\Sigma_{h-1}^t)^{-1}} + \sqrt{d} H \epsilon \sum_{t=1}^{T} \sum_{h=2}^{H} \left\| \hat{\phi}_{h-1}(\pi^\star) \right\|_{(\Sigma_{h-1}^t)^{-1}},$$

$$= \sum_{t=1}^{T} \sum_{\pi} p^t(\pi) \cdot \left( \mathbb{E}_{\boldsymbol{\pi}^t \sim \rho^t} \mathbb{E}^{\boldsymbol{\pi}^t} \left[ \hat{\ell}^t(\pi; \boldsymbol{\pi}^t, \boldsymbol{x}_{1:H}, \boldsymbol{a}_{1:H}) \right] - b^t(\pi) \right)$$

$$- \sum_{t=1}^{T} \left( \mathbb{E}_{\boldsymbol{\pi}^t \sim \rho^t} \mathbb{E}^{\boldsymbol{\pi}^t} \left[ \hat{\ell}^t(\pi^\star; \boldsymbol{\pi}^t, \boldsymbol{x}_{1:H}, \boldsymbol{a}_{1:H}) \right] - b^t(\pi^\star) \right)$$

$$+ \sum_{t=1}^{T} \sum_{\pi} p^t(\pi) b^t(\pi) - \sum_{t=1}^{T} b^t(\pi^\star) + \sqrt{d} H \epsilon \sum_{t=1}^{T} \sum_{\pi} p^t(\pi) \sum_{h=2}^{H} \left\| \hat{\phi}_{h-1}(\pi) \right\|_{(\Sigma_{h-1}^t)^{-1}}$$

$$+ \sqrt{d} H \epsilon \sum_{t=1}^{T} \sum_{h=2}^{H} \left\| \hat{\phi}_{h-1}(\pi^\star) \right\|_{(\Sigma_{h-1}^t)^{-1}}$$

$$= \sum_{t=1}^{T} \sum_{\pi} p^t(\pi) \cdot \left( \mathbb{E}_{\boldsymbol{\pi}^t \sim \rho^t} \mathbb{E}^{\boldsymbol{\pi}^t} \left[ \hat{\ell}^t(\pi; \boldsymbol{\pi}^t, \boldsymbol{x}_{1:H}, \boldsymbol{a}_{1:H}) \right] - b^t(\pi) \right)$$

$$- \sum_{t=1}^{T} \left( \mathbb{E}_{\boldsymbol{\pi}^t \sim \rho^t} \mathbb{E}^{\boldsymbol{\pi}^t} \left[ \hat{\ell}^t(\pi^\star; \boldsymbol{\pi}^t, \boldsymbol{x}_{1:H}, \boldsymbol{a}_{1:H}) \right] - b^t(\pi^\star) \right)$$

$$+ 4\sqrt{d} H \epsilon \sum_{t=1}^{T} \sum_{\pi} \rho^t(\pi) \sum_{h=2}^{H} \left\| \hat{\phi}_{h-1}(\pi) \right\|_{(\Sigma_{h-1}^t)^{-1}}, \quad \text{(since } p^t(\pi) \leq 2\rho^t(\pi)) \tag{27}$$

$$\leq \textbf{FTRL} + 4dH^2 \epsilon T,$$

where

$$\textbf{FTRL} := \sum_{t=1}^{T} \sum_{\pi} p^t(\pi) \cdot \left( \mathbb{E}_{\boldsymbol{\pi}^t \sim \rho^t} \mathbb{E}^{\boldsymbol{\pi}^t} \left[ \hat{\ell}^t(\pi; \boldsymbol{\pi}^t, \boldsymbol{x}_{1:H}, \boldsymbol{a}_{1:H}) \right] - b^t(\pi) \right)$$

$$- \sum_{t=1}^{T} \left( \mathbb{E}_{\boldsymbol{\pi}^t \sim \rho^t} \mathbb{E}^{\boldsymbol{\pi}^t} \left[ \hat{\ell}^t(\pi^\star; \boldsymbol{\pi}^t, \boldsymbol{x}_{1:H}, \boldsymbol{a}_{1:H}) \right] - b^t(\pi^\star) \right). \tag{28}$$

We now bound the FTRL term. Since $\rho^t(\pi) = (1 - \gamma) p^t(\pi) + \gamma \sum_{h=1}^{H} J(\pi, h)$, define $\Sigma_J = \sum_{\pi \in \Pi} \sum_{h=1}^{H} J(\pi, h) \hat{\phi}_h(\pi) \hat{\phi}_h(\pi)^\top$. Note that for any $t \in [T]$ and $h \in [H]$, we have $\Sigma_h^t \geq \gamma \Sigma_J$.

By the triangle inequality, we have for any $\pi, \pi^t$ and $(x_{1:H}, a_{1:H})$,

$$\left| \hat{\ell}^t(\pi; \pi^t, x_{1:H}, a_{1:H}) \right| \leq \left| \frac{\pi_1(\boldsymbol{a}_1 \mid \boldsymbol{x}_1)}{\pi_1^t(\boldsymbol{a}_1 \mid \boldsymbol{x}_1)} \cdot \ell_1^t(\boldsymbol{x}_1, \boldsymbol{a}_1) \right|$$

$$+ \sum_{h=2}^{H} \left| \hat{\phi}_{h-1}(\pi)^\top \left( \Sigma_{h-1}^t \right)^{-1} \hat{\phi}_{h-1}(\pi^t) \frac{\pi_h(a_h \mid x_h)}{\pi_h^t(a_h \mid x_h)} \ell_h^t(x_h, a_h) \right|,$$

$$\leq \frac{|\mathcal{A}|}{\beta} + \frac{|\mathcal{A}|}{\beta} \sum_{h=2}^{H} \left| \hat{\phi}_{h-1}(\pi)^\top \left( \Sigma_{h-1}^t \right)^{-1} \hat{\phi}_{h-1}(\pi^t) \right|,$$

$$\text{(}\beta\text{-mixture of uniform policy)}$$

$$\leq \frac{|\mathcal{A}|}{\beta} + \frac{|\mathcal{A}|}{\beta \gamma} \sum_{h=2}^{H} \left\| \hat{\phi}_{h-1}(\pi) \right\|_{\Sigma_J^{-1}} \left\| \hat{\phi}_{h-1}(\pi^t) \right\|_{\Sigma_J^{-1}},$$

$$\leq \frac{|\mathcal{A}|}{\beta} + \frac{|\mathcal{A}| H d}{\beta \gamma},$$

$$\leq \frac{2|\mathcal{A}| H d}{\beta \gamma},$$

and

$$\left| b^t(\pi) \right| = \left| \sqrt{d} H \epsilon \sum_{h=2}^{H} \left\| \hat{\phi}_{h-1}(\pi) \right\|_{(\Sigma_{h-1}^t)^{-1}} \right| \le dH^2 \frac{\epsilon}{\sqrt{\gamma}}.$$

To ensure $\eta \left| \hat{\ell}^t(\pi; \pi^t, x_{1:H}, a_{1:H}) - b^t(\pi) \right| \le 1$, it suffices to set $\eta \le \left( \frac{2|\mathcal{A}|Hd}{\beta\gamma} + dH^2 \frac{\epsilon}{\sqrt{\gamma}} \right)^{-1}$. Under this constraint, from Lemma I.5, we have

$$\textbf{FTRL} \le \frac{\log|\Pi|}{\eta} + \underbrace{2\eta \sum_{t=1}^{T} \sum_{\pi \in \Pi'} p^t(\pi) \cdot \mathbb{E}_{\boldsymbol{\pi}^t \sim \rho^t} \mathbb{E}^{\boldsymbol{\pi}^t} \left[ \hat{\ell}^t(\pi; \boldsymbol{\pi}^t, \boldsymbol{x}_{1:H}, \boldsymbol{a}_{1:H})^2 \right]}_{\textbf{Stability-1}}$$

$$+ \underbrace{\mathbb{E} \left[ 2\eta \sum_{t=1}^{T} \sum_{\pi \in \Pi'} p^t(\pi) b^t(\pi)^2 \right]}_{\textbf{Stability-2}}$$

For any $t \in [T]$, we have

$$\sum_{\pi \in \Pi'} p^t(\pi) \cdot \mathbb{E}_{\boldsymbol{\pi}^t \sim \rho^t} \mathbb{E}^{\boldsymbol{\pi}^t} \left[ \hat{\ell}^t(\pi; \boldsymbol{\pi}^t, \boldsymbol{x}_{1:H}, \boldsymbol{a}_{1:H})^2 \right]$$

$$\le H \mathbb{E}_{\boldsymbol{\pi}^t \sim \rho^t} \mathbb{E}^{\boldsymbol{\pi}^t} \left[ \frac{\pi_1(\boldsymbol{a}_1 \mid \boldsymbol{x}_1)^2}{\boldsymbol{\pi}_1^t(\boldsymbol{a}_1 \mid \boldsymbol{x}_1)^2} \ell_1^t(\boldsymbol{x}_1, \boldsymbol{a}_1)^2 \right]$$

$$+ H \sum_{h=2}^{H} \mathbb{E}_{\boldsymbol{\pi}^t \sim \rho^t} \mathbb{E}^{\boldsymbol{\pi}^t} \left[ \sum_{\pi \in \Pi'} p^t(\pi) \hat{\phi}_{h-1}(\pi)^\top \left( \Sigma_{h-1}^t \right)^{-1} \hat{\phi}_{h-1}(\boldsymbol{\pi}^t) \hat{\phi}_{h-1}(\boldsymbol{\pi}^t)^\top \left( \Sigma_{h-1}^t \right)^{-1} \hat{\phi}_{h-1}(\pi) \frac{\pi_h(\boldsymbol{a}_h \mid \boldsymbol{x}_h)^2}{\boldsymbol{\pi}_h^t(\boldsymbol{a}_h \mid \boldsymbol{x}_h)^2} \ell_h^t(\boldsymbol{x}_h, \boldsymbol{a}_h)^2 \right],$$

$$\le \frac{H|\mathcal{A}|}{\beta} + 2H \sum_{h=2}^{H} \mathbb{E}_{\boldsymbol{\pi}^t \sim \rho^t} \mathbb{E}^{\boldsymbol{\pi}^t} \left[ \text{Tr} \left( \hat{\phi}_{h-1}(\boldsymbol{\pi}^t) \hat{\phi}_{h-1}(\boldsymbol{\pi}^t)^\top \left( \Sigma_{h-1}^t \right)^{-1} \right) \sum_{a \in \mathcal{A}} \frac{\pi_h(a \mid \boldsymbol{x}_h)^2}{\boldsymbol{\pi}_h^t(a \mid \boldsymbol{x}_h)} \ell_h^t(\boldsymbol{x}_h, a)^2 \right],$$

$$\tag{29}$$

$$\le \frac{H|\mathcal{A}|}{\beta} + \frac{2H|\mathcal{A}|}{\beta} \sum_{h=2}^{H} \mathbb{E}_{\boldsymbol{\pi}^t \sim \rho^t} \left[ \text{Tr} \left( \hat{\phi}_{h-1}(\boldsymbol{\pi}^t) \hat{\phi}_{h-1}(\boldsymbol{\pi}^t)^\top \left( \Sigma_{h-1}^t \right)^{-1} \right) \right],$$

$$\le \frac{3dH^2|\mathcal{A}|}{\beta}.$$

where (29) follows by the fact that $p^t(\pi) \le \frac{1}{1-\gamma} \rho^t(\pi)$ and $\frac{1}{1-\gamma} \le 2$. Thus,

$$\textbf{Stability-1} = 2\eta \sum_{\pi \in \Pi'} p^t(\pi) \cdot \mathbb{E}_{\boldsymbol{\pi}^t \sim \rho^t} \mathbb{E}^{\boldsymbol{\pi}^t} \left[ \hat{\ell}^t(\pi; \boldsymbol{\pi}^t, \boldsymbol{x}_{1:H}, \boldsymbol{a}_{1:H})^2 \right] \le \frac{6d\eta H^2 |\mathcal{A}| T}{\beta}.$$

Moreover,

$$\textbf{Stability-2} = 2\eta \mathbb{E} \left[ \sum_{t=1}^{T} \sum_{\pi \in \Pi'} p^t(\pi) b^t(\pi)^2 \right],$$

$$= 2\eta dH^3 \epsilon^2 \mathbb{E} \left[ \sum_{t=1}^{T} \sum_{h=2}^{H} \sum_{\pi \in \Pi'} p^t(\pi) \left\| \hat{\phi}_{h-1}(\pi) \right\|_{(\Sigma_{h-1}^t)^{-1}}^2 \right],$$

$$\le 4\eta dH^3 \epsilon^2 \mathbb{E} \left[ \sum_{t=1}^{T} \sum_{h=1}^{H-1} \sum_{\pi \in \Pi'} \rho^t(\pi) \left\| \hat{\phi}_{h-1}(\pi) \right\|_{(\Sigma_{h-1}^t)^{-1}}^2 \right],$$

$$\le 4\eta d^2 H^4 \epsilon^2 T.$$

$$\square$$

Combing Lemma C.2, Lemma C.3 and Lemma C.5, if $\eta \le \left( \frac{2|\mathcal{A}|Hd}{\beta\gamma} + dH^2 \frac{\epsilon}{\sqrt{\gamma}} \right)^{-1}$, we have

$$\text{Reg}_T(\pi_0^\star) \le H\gamma T + 2H^2 \beta T + H^2 T \epsilon + \frac{\log|\Pi|}{\eta} + \frac{6d\eta H^2 |\mathcal{A}| T}{\beta}$$

$$+ 4\eta d^2 H^4 \epsilon^2 T + 4dH^2 \epsilon T + \mathcal{O}\left(\frac{H^3 d^2 |\mathcal{A}|(d^2 + |\mathcal{A}|)}{\epsilon^2} \log^2\left(TdH|\Phi\|\Upsilon|\right)\right).$$

By setting $\epsilon = T^{-\frac{1}{3}}$, $\gamma = T^{-\frac{1}{3}}$, $\beta = T^{-\frac{1}{3}}$, $\eta = \frac{1}{4Hd|\mathcal{A}|}T^{-\frac{2}{3}}$, we have for any $\pi_0^\star \in \Pi$,

$$\mathrm{Reg}_T(\pi_0^\star) \le \mathcal{O}\left(d^2 H^3 |\mathcal{A}|(d^2 + |\mathcal{A}|)T^{\frac{2}{3}} \log|\Pi| \log^2\left(TdH|\Phi\|\Upsilon|\right)\right).$$

# D More Details of Inefficient Model-Free Algorithm in Section 4.1

## D.1 Algorithm Description

In this section, we give a more detailed introduction of the algorithm mentioned in Section 4.1. This algorithm is model-free and achieves $T^{\frac{2}{3}}$ regret, but it is computationally inefficient. We consider the low-rank MDPs with linear losses that satisfies Assumption 2.1 and Assumption 2.2.

Let $\mathcal{C}(S, \epsilon')$ be $\epsilon'$-net of space $S$. We define necessary policy and function classes in Definition D.1.

**Definition D.1.** *We define linear policy class and its discretization as*

$$\Pi_{\text{lin}} = \left\{ \pi : \mathcal{X} \to \Delta(\mathcal{A}) \ \middle| \ \pi_h(a \mid x) = \mathbb{I}\left\{ a = \underset{a \in \mathcal{A}}{\arg\min}\, \phi_h(x,a)^\top \theta_h \right\}, \ h \in [H], \ \theta_h \in \mathbb{B}_d\left(\sqrt{d}HT\right), \ \phi \in \Phi \right\}.$$

$$\Pi_{\text{lin}}^{\text{cov}}(\epsilon') = \left\{ \pi : \mathcal{X} \to \Delta(\mathcal{A}) \ \middle| \ \pi_h(a \mid x) = \mathbb{I}\left\{ a = \underset{a \in \mathcal{A}}{\arg\min}\, \phi_h(x,a)^\top \theta_h \right\}, \ h \in [H], \ \theta_h \in \mathcal{C}\left(\mathbb{B}_d\left(\sqrt{d}HT\right), \epsilon'\right), \ \phi \in \Phi \right\}.$$

*Define corresponding function class as follows*

$$\mathcal{F}^\pi = \left\{ f : \mathcal{X} \to [-1,1] \ \middle| \ f(x) = \sum_a \pi(a|x)\phi(x,a)^\top \theta, \ \text{for } \theta \in \mathbb{B}^d(\sqrt{d}) \text{ and } \phi \in \Phi \right\}$$

$$\mathcal{F} = \left\{ f : \mathcal{X} \to [-1,1] \ \middle| \ f \in \bigcup_{\pi \in \Pi_{\text{lin}}} \mathcal{F}^\pi \right\}.$$

Our main algorithm is given in Algorithm 5, which shares the same structure as Algorithm 2, but with a different initial phase to learn expected feature estimator $\hat{\phi}_h(\pi) = \sum_{(x,a) \in \mathcal{X} \times \mathcal{A}} \hat{d}_h^\pi(x)\pi(a|x)\hat{\phi}_h(x,a)$ to approximate $\phi_h^\star(\pi) = \sum_{(x,a) \in \mathcal{X} \times \mathcal{A}} d_h^\pi(x)\pi(a|x)\phi_h^\star(x,a)$ for every $h \in [H]$. In Algorithm 2, under Assumption 3.1, it is feasible to use established model-based approach to learn an accurate estimated transition $\widehat{P}$ together with its feature $\hat{\phi}$. The occupancy estimator $\hat{d}_{1:H}^\pi$ is induced by $\widehat{P}$ which also enjoy small errors. However, when we move to model-free settings with Assumption 4.1, there is no existing approach that could guarantee a good estimation for $\hat{d}_h^\pi(x)$ and $\hat{\phi}$.

To tackle this challenge, we first call VoX (Mhammedi et al., 2023) to construct a policy cover $\Psi_{1:H}^{\text{cov}}$, and then play every policy in $\Psi_{1:H}^{\text{cov}}$ for $n$ episodes to collect data. Subsequently, these data are fed into Algorithm 6 to jointly solve estimated occupancy $\hat{d}_{1:H}^\pi$ and feature $\hat{\phi}_{1:H}$. Algorithm 6 is similar to (Liu et al., 2023, Algorithm 1), which is used to estimate occupancy on the fly for linear MDP. In Algorithm 6, given a target policy $\pi$, we jointly solve $\hat{\phi}_{1:H} \in \Phi$, $\hat{d}_{1:H}^\pi \in [0,1]^{|\mathcal{X}|}$, and $(\hat{\xi}_{1:H,f})_{f \in \mathcal{F}^\pi} \subset \mathbb{B}^d\left(\sqrt{d}\right)$ that satisfies four constrains, where $\hat{\xi}_{h,f}$ is the estimation of $\xi_{h,f}^\star :=$ $\sum_{x' \in \mathcal{X}} \mu_{h+1}^\star(x')f(x')$. The first constraint Eq. (30) ensures the estimated occupancy $\hat{d}_{1:H}^\pi$ are valid distrbutions. The second constrant Eq. (31) enforces the estimated values to follow the dynamic programming relationship between the occupancy of layer $h$ and layer $h+1$, which helps to control the propagation of estimation errors across layers through the bias of $\hat{\phi}_h(x,a)^\top \hat{\xi}_{h,f}$. The third constrain Eq. (32) and fourth constrain Eq. (33) are then used to bound the estimated bias of $\hat{\phi}_h(x,a)^\top \hat{\xi}_{h,f}$ by utilizing the data collected from policies in policy covers $\Psi_{1:H}^{\text{cov}}$. Note that applying Eq. (33) requires access to the whole state space, which is an additional assumption not needed in previous algorithms. The gurantee of Algorithm 5 is given in Theorem D.1, where the $\widetilde{O}$ hides the logarithmic dependence on $d, H, |\mathcal{A}|, T$.

**Theorem D.1.** *Algorithm 5 ensures* $\text{Reg}_T \le \widetilde{O}\left(d^8 H^6 |\mathcal{A}| T^{\frac{2}{3}} \log(|\Phi|)\right)$.

---

**Algorithm 5** Model-Free Algorithm for Bandit Feedback

---

**Input:** Policy class $\Pi = \Pi_{\text{lin}}^{\text{cov}}(\frac{1}{T})$.

1: Set $\epsilon = 18^{-1}d^{\frac{5}{2}}T^{-\frac{1}{3}}, \gamma = T^{-\frac{1}{3}}, \beta = T^{-\frac{1}{3}}, \eta = (4Hd|\mathcal{A}|)^{-1}T^{-\frac{2}{3}}, n = 11250d^{\frac{5}{2}}|\mathcal{A}|T^{\frac{2}{3}}\log\frac{3dnHT|\Phi|}{\delta}$ and $T_0 = \widetilde{O}\left(\epsilon^{-2}|\mathcal{A}|d^{13}H^6\log(|\Phi|/\delta)\right)$.

2: Get $\Psi_{1:H}^{\text{cov}} \leftarrow \text{VoX}(\Phi, \varepsilon, \delta)$ using $T_0$ episodes.

3: For every policy $\pi' \in \Psi_{1:H}^{\text{cov}}$, play it for $n$ episodes and get the data set $\left(\mathcal{D}_h^{\pi'}\right)_{h\in[H]}$ where $\mathcal{D}_h^{\pi'}$ consists of tuples $(x, a, x')$ such that $(x, a) \sim d_h^{\pi'}$ and $x' \sim P^\star(\cdot \mid x, a)$.

4: Define the policy space $\Pi' = \{\pi' : \exists \pi \in \Pi, \ \pi_h'(\cdot \mid x) = (1-\beta)\pi_h(\cdot \mid x) + \beta/|\mathcal{A}|, \ \forall x, h\}$.

5: Get $\hat{\phi}_h(\cdot) \leftarrow \text{EOM-PC}\left(\Pi', \left(\mathcal{D}_h^{\pi'}\right)_{h\in[H],\pi'\in\Psi_{1:H}^{\text{cov}}}\right)$ from Algorithm 6.

6: **for** $t = T_0 + 1, T_0 + 2, \ldots, T$ **do**

7:    Define $p^t(\pi) \propto \exp\left(-\eta\sum_{i=1}^{t-1}\left(\hat{\ell}^i(\pi) - b^i(\pi)\right)\right)$, for all $\pi \in \Pi'$.

8:    Let $\rho^t(\pi) = (1-\gamma)p^t(\pi) + \frac{\gamma}{H-1}\sum_{h=1}^{H-1}J_h$, where $J_h = \text{John}(\hat{\phi}_h(\cdot), \Pi')$. `// John as in §2`

9:    Execute policy $\boldsymbol{\pi}^t \sim \rho^t$ and observe trajectory $(\boldsymbol{x}_{1:H}^t, \boldsymbol{a}_{1:H}^t)$ and losses $\boldsymbol{\ell}_h^t = \ell_h^t(\boldsymbol{x}_h^t, \boldsymbol{a}_h^t)$.

10:    Define $\Sigma_h^t = \sum_{\pi\in\Pi'}\rho^t(\pi)\cdot\hat{\phi}_h(\pi)\hat{\phi}_h(\pi)^\top, b^t(\pi) = d^{\frac{11}{2}}HT^{-\frac{1}{3}}\cdot\sum_{h=1}^{H-1}\|\hat{\phi}_h(\pi)\|_{(\Sigma_h^t)^{-1}}$, and

$$\hat{\ell}^t(\pi) = \frac{\pi_1(\boldsymbol{a}_1^t \mid \boldsymbol{x}_1^t)}{\boldsymbol{\pi}_1^t(\boldsymbol{a}_1^t \mid \boldsymbol{x}_1^t)}\boldsymbol{\ell}_1^t + \sum_{h=2}^{H}\hat{\phi}_{h-1}(\pi)^\top\left(\Sigma_{h-1}^t\right)^{-1}\hat{\phi}_{h-1}(\boldsymbol{\pi}^t)\frac{\pi_h(\boldsymbol{a}_h^t \mid \boldsymbol{x}_h^t)}{\boldsymbol{\pi}_h^t(\boldsymbol{a}_h^t \mid \boldsymbol{x}_h^t)}\boldsymbol{\ell}_h^t.$$

11: **end for**

---

---

**Algorithm 6** EOM-PC$\left(\Pi, \left(\mathcal{D}_h^{\pi'}\right)_{h\in[H],\pi'\in\Psi_{1:H}^{\text{cov}}}\right)$ (Estimate Occupancy Measure with Policy Cover)

---

**Input:** The policy class $\Pi$, datasets $\left(\mathcal{D}_h^{\pi'}\right)_{h\in[H]}$ for every $\pi' \in \Psi_{1:H}^{\text{cov}}$

Jointly find $\hat{\phi}_h \in \Phi, (\hat{d}_h^\pi)_{\pi\in\Pi} \in [0,1]^{|\mathcal{X}|}$, and $(\hat{\xi}_{h,f})_{f\in\mathcal{F}} \subset \mathbb{B}^d\left(\sqrt{d}\right)$ for any $h \in [H]$ such that for all $\pi \in \Pi$,

$$\sum_{x\in\mathcal{X}}\hat{d}_h^\pi(x) = 1, \qquad \forall h \in [H] \tag{30}$$

$$\sum_{x'\in\mathcal{X}}\hat{d}_{h+1}^\pi(x')f(x') = \sum_{x\in\mathcal{X}}\sum_{a\in\mathcal{A}}\hat{d}_h^\pi(x)\pi(a|x)\hat{\phi}_h(x,a)^\top\hat{\xi}_{h,f}, \qquad \forall f \in \mathcal{F}, h \in [H] \tag{31}$$

$$\sum_{x,a,x'\in\mathcal{D}_h^{\pi'}}\left(f(x') - \hat{\phi}_h(x,a)^\top\hat{\xi}_{h,f}\right)^2 - \min_{(\phi,\xi)\in\Phi\times\mathbb{B}_d(\sqrt{d})}\sum_{x,a,x'\in\mathcal{D}_h^{\pi'}}\left(f(x') - \phi(x,a)^\top\xi\right)^2$$

$$\leq 132d^{\frac{3}{2}}\log(3dnHT|\Phi|/\delta), \qquad \forall \pi' \in \Psi_{1:H}^{\text{cov}}, \ f \in \mathcal{F}, h \in [H] \tag{32}$$

$$\max_{x,a}\left|\hat{\phi}_h(x,a)^\top\hat{\xi}_{h,f}\right| \leq 1 \qquad \forall f \in \mathcal{F}, h \in [H] \tag{33}$$

**Output:** $\hat{\phi}_h : \Pi \to \mathbb{R}^d, \hat{\phi}_h(\pi) = \sum_{(x,a)\in\mathcal{X}\times\mathcal{A}}\hat{d}_h^\pi(x)\pi(a|x)\hat{\phi}_h(x,a), \forall h \in [H]$.

---

### D.2   Analysis of Occupancy Estimation from Algorithm 6

**Lemma D.1.** *With probability* $1 - \delta$, $\phi_{1:H}^\star, (d_{1:H}^\pi)_{\pi\in\Pi'}$, *and* $\xi_{h,f}^\star := \sum_{x'\in\mathcal{X}}\mu_{h+1}^\star(x')f(x'), \forall f \in \mathcal{F}, \forall h \in [H]$ *is a solution to Algorithm 6.*

**Proof.** Since for any policy $\pi$ and any $h \in [H]$, $\sum_{x \in \mathcal{X}} d_h^\pi(x) = 1$, Eq. (30) holds. For any policy $\pi$, any $f \in \mathcal{F}^\pi$ and any $h \in [H]$, we have

$$
\begin{aligned}
\sum_{x' \in \mathcal{X}} d_{h+1}^\pi(x') f(x') &= \sum_{x' \in \mathcal{X}} \sum_{x \in \mathcal{X}} \sum_{a \in \mathcal{A}} d_h^\pi(x) \pi(a|x) P_h^\star(x' \mid x, a) f(x') \\
&= \sum_{x \in \mathcal{X}_h} \sum_{a \in \mathcal{A}} d_h^\pi(x) \pi(a|x) \phi_h^\star(x,a)^\top \sum_{x' \in \mathcal{X}_{h+1}} \mu_{h+1}^\star(x') f(x') \\
&= \sum_{x \in \mathcal{X}_h} \sum_{a \in \mathcal{A}} d_h^\pi(x) \pi(a|x) \xi_{h,f}^\star.
\end{aligned}
$$

Thus, Eq. (31) holds. From Exercise 27.6 of Lattimore and Szepesvári (2020), the $\epsilon$-net of $\mathbb{B}_d(R)$ is $\left(\frac{3R}{\epsilon}\right)^d$. Thus. $|\Pi'| = \left|\Pi_{\text{lin}}^{\text{cov}}(\frac{1}{T})\right| = |\Phi| \left(3\sqrt{d}HT^2\right)^d$. We also have $\left|\mathcal{C}\left(\mathbb{B}_d(\sqrt{d}), \frac{1}{T}\right)\right| = \left(3\sqrt{d}T\right)^d$ and for any policy $\pi$, $\left|\mathcal{C}\left(\mathcal{F}^\pi, \frac{1}{T}\right)\right| = |\Phi| \left(3\sqrt{d}T\right)^d$. To consider all possible instances, define

$$
\mathcal{N}_T := |\Pi'| \left|\mathcal{C}\left(\mathbb{B}_d(\sqrt{d}), \epsilon\right)\right| |\mathcal{C}(\mathcal{F}^\pi, \epsilon)| |\Psi_{1:H}^{\text{cov}}| |\Phi| H \leq dH^2 |\Phi|^3 \left(3\sqrt{d}HT^2\right)^{3d}
$$

Thus, by union bound, with probability of $1 - \delta$, for every $\pi \in \Pi'$, every $\pi' \in \Psi_{1:H}^{\text{cov}}$, every $f \in \mathcal{C}(\mathcal{F}^\pi, \epsilon)$, every $\xi \in \mathcal{C}\left(\mathbb{B}_d(\sqrt{d}), \epsilon\right)$, every $\phi \in \Phi$ and every $h \in [H]$, we have

$$
\begin{aligned}
&\sum_{x,a,x' \in \mathcal{D}_h^{\pi'}} \left(f(x') - \phi_h^\star(x,a)^\top \xi_{h,f}^\star\right)^2 - \sum_{x,a,x' \in \mathcal{D}_h^{\pi'}} \left(f(x') - \phi(x,a)^\top \xi\right)^2 \\
&= -2 \sum_{x,a,x' \in \mathcal{D}_h^{\pi'}} \left(f(x') - \phi_h^\star(x,a)^\top \xi_{h,f}^\star\right) \left(\phi_h^\star(x,a)^\top \xi_{h,f}^\star - \phi(x,a)^\top \xi\right) \\
&\quad - \sum_{x,a,x' \in \mathcal{D}_h^{\pi'}} \left(\phi_h^\star(x,a)^\top \xi_{h,f}^\star - \phi(x,a)^\top \xi\right)^2 \\
&= -2 \sum_{x,a,x' \in \mathcal{D}_h^{\pi'}} \left(f(x') - \mathbb{E}_{x' \sim P^\star(\cdot|x,a)}[f(x')]\right) \left(\phi_h^\star(x,a)^\top \xi_{h,f}^\star - \phi(x,a)^\top \xi\right) \\
&\quad - \sum_{x,a,x' \in \mathcal{D}_h^{\pi'}} \left(\phi_h^\star(x,a)^\top \xi_{h,f}^\star - \phi(x,a)^\top \xi\right)^2 \\
&\leq 8 \sqrt{\sum_{x,a,x' \in \mathcal{D}_h^{\pi'}} \left(\phi_h^\star(x,a)^\top \xi_{h,f}^\star - \phi(x,a)^\top \xi\right)^2 \log \frac{|\mathcal{N}_T|}{\delta}} + 4\sqrt{d} \log \frac{|\mathcal{N}_T|}{\delta} \\
&\quad - \sum_{x,a,x' \in \mathcal{D}_h^{\pi'}} \left(\phi_h^\star(x,a)^\top \xi_{h,f}^\star - \phi(x,a)^\top \xi\right)^2 \qquad \text{(Freedman's Inequality)} \\
&\leq 20\sqrt{d} \log \frac{|\mathcal{N}_T|}{\delta} \qquad\qquad\qquad\qquad\qquad\qquad\qquad\qquad\quad \text{(AM-GM)} \\
&\leq 120 d^{\frac{3}{2}} \log \frac{3dHT|\Phi|}{\delta}
\end{aligned}
$$

Bounding the distance through $\frac{1}{T}$-net, we have with probability of $1 - \delta$, for every $\pi \in \Pi'$, every $\pi' \in \Psi_{1:H}^{\text{cov}}$, every $f \in \mathcal{F}^\pi$, every $\xi \in \mathbb{B}_d(\sqrt{d})$, every $\phi \in \Phi$ and every $h \in [H]$,

$$
\begin{aligned}
\sum_{x,a,x' \in \mathcal{D}_h^{\pi'}} \left(f(x') - \phi_h^\star(x,a)^\top \xi_{h,f}^\star\right)^2 - \sum_{x,a,x' \in \mathcal{D}_h^{\pi'}} \left(f(x') - \phi(x,a)^\top \xi\right)^2 &\leq 120 d^{\frac{3}{2}} \log \frac{3dHT|\Phi|}{\delta \epsilon} + \frac{12\sqrt{d}n}{T} \\
&\leq 132 d^{\frac{3}{2}} \log \frac{3dnHT|\Phi|}{\delta}
\end{aligned}
$$

Thus, Eq. (32) also holds. Finally, for all $x, a$, we have

$$
\left|\phi_h^\star(x,a)^\top \xi_{h,f}^\star\right| = \left|\sum_{x' \in \mathcal{X}} \phi_h^\star(x,a)^\top \mu_{h+1}^\star(x') f(x')\right| = \left|\sum_{x' \in \mathcal{X}} P^\star(x' \mid x, a) f(x')\right| \leq 1.
$$

Thus, $\phi_h^\star$, $d_h^\pi$, and $\xi_{h,f}^\star := \sum_{x' \in \mathcal{X}} \mu_{h+1}^\star(x') f(x')$, $\forall f \in \mathcal{F}^\pi, h \in [H]$ satisfy all Eq. (30) – Eq. (33) and is a solution to Algorithm 6. $\qquad\square$

**Lemma D.2.** *With probability* $1 - \delta$, *for all* $f \in \mathcal{F}$, *any solution* $\hat{d}^{\pi}$ *from Algorithm 6 for any* $\pi \in \Pi'$ *satisfies*

$$\left| \sum_x (\hat{d}_h^{\pi}(x) - d_h^{\pi}(x)) f(x) \right| \leq d^5 H T^{-\frac{1}{3}}$$

**Proof.**

For every solution $\hat{\phi}_{1:H}, \hat{\xi}_{1:H, f \in \mathcal{F}^{\pi}}, (\hat{d}_{1:H}^{\pi})_{\pi \in \Pi'}$ of Algorithm 6, we have

$$\sum_{x,a,x' \in \mathcal{D}_h^{\pi'}} \left( f(x') - \hat{\phi}_h(x,a)^\top \hat{\xi}_{h,f} \right)^2 - \sum_{x,a,x' \in \mathcal{D}_h^{\pi'}} \left( f(x') - \phi_h^\star(x,a)^\top \xi_{h,f}^\star \right)^2$$

$$= \sum_{x,a,x' \in \mathcal{D}_h^{\pi'}} \left( f(x') - \hat{\phi}_h(x,a)^\top \hat{\xi}_{h,f} \right)^2 - \min_{(\phi,\xi)} \sum_{x,a,x' \in \mathcal{D}_h^{\pi'}} \left( f(x') - \phi(x,a)^\top \xi \right)^2$$

$$\underbrace{+ \min_{(\phi,\xi)} \sum_{x,a,x' \in \mathcal{D}_h^{\pi'}} \left( f(x') - \phi(x,a)^\top \xi \right)^2 - \sum_{x,a,x' \in \mathcal{D}_h^{\pi'}} \left( f(x') - \phi_h^\star(x,a)^\top \xi_{h,f}^\star \right)^2}_{\leq 0}$$

$$\leq 132 d^{\frac{3}{2}} \log \frac{3dnHT|\Phi|}{\delta} \tag{34}$$

where the last step comes from the constrain Eq. (32). On the other hand

$$2 \sum_{x,a,x' \in \mathcal{D}_h^{\pi'}} \left( f(x') - \hat{\phi}_h(x,a)^\top \hat{\xi}_{h,f} \right)^2 - 2 \sum_{x,a,x' \in \mathcal{D}_h^{\pi'}} \left( f(x') - \phi_h^\star(x,a)^\top \xi_{h,f}^\star \right)^2$$

$$= 4 \sum_{x,a,x' \in \mathcal{D}_h^{\pi'}} \left( f(x') - \phi_h^\star(x,a)^\top \xi_{h,f}^\star \right) \left( \phi_h^\star(x,a)^\top \xi_{h,f}^\star - \hat{\phi}_h(x,a)^\top \hat{\xi}_{h,f} \right)$$

$$+ 2 \sum_{x,a,x' \in \mathcal{D}_h^{\pi'}} \left( \phi_h^\star(x,a)^\top \xi_{h,f}^\star - \hat{\phi}_h(x,a)^\top \hat{\xi}_{h,f} \right)^2$$

$$= 4 \sum_{x,a,x' \in \mathcal{D}_h^{\pi'}} \left( f(x') - \mathbb{E}_{x' \sim P^\star(\cdot|x,a)}[f(x')] \right) \left( \phi_h^\star(x,a)^\top \xi_{h,f}^\star - \hat{\phi}_h(x,a)^\top \hat{\xi}_{h,f} \right)$$

$$+ 2 \sum_{x,a,x' \in \mathcal{D}_h^{\pi'}} \left( \phi_h^\star(x,a)^\top \xi_{h,f}^\star - \hat{\phi}_h(x,a)^\top \hat{\xi}_{h,f} \right)^2 \tag{35}$$

Combing Eq. (34) and Eq. (35), we have

$$\sum_{x,a,x' \in \mathcal{D}_h^{\pi'}} \left( \phi_h^\star(x,a)^\top \xi_{h,f}^\star - \hat{\phi}_h(x,a)^\top \hat{\xi}_{h,f} \right)^2$$

$$\leq -4 \sum_{x,a,x' \in \mathcal{D}_h^{\pi'}} \left( f(x') - \mathbb{E}_{x' \sim P^\star(\cdot|x,a)}[f(x')] \right) \left( \phi_h^\star(x,a)^\top \xi_{h,f}^\star - \hat{\phi}_h(x,a)^\top \hat{\xi}_{h,f} \right)$$

$$- \sum_{x,a,x' \in \mathcal{D}_h^{\pi'}} \left( \phi_h^\star(x,a)^\top \xi_{h,f}^\star - \hat{\phi}_h(x,a)^\top \hat{\xi}_{h,f} \right)^2$$

$$+ 264 d^{\frac{3}{2}} \log \frac{3dnHT|\Phi|}{\delta}$$

$$\leq 288 d^{\frac{3}{2}} \log \frac{3dnHT|\Phi|}{\delta} \qquad \qquad \text{(Lemma I.2 with } \lambda = \frac{1}{8})$$

Since for every data tuple $(x,a,x') \in \mathcal{D}_h^{\pi'}$, $(x,a) \sim d_h^{\pi'}$ independently, by Lemma I.3, for every $\pi' \in \Psi_{1:H}^{\text{cov}}$ we have

$$\mathbb{E}^{\pi'} \left[ \left( \phi_h^\star(x,a)^\top \xi_{h,f}^\star - \hat{\phi}_h(x,a)^\top \hat{\xi}_{h,f} \right)^2 \right] \leq \frac{2}{n} \sum_{x,a \in \mathcal{D}_h^{\pi'}} \left( \phi_h^\star(x,a)^\top \xi_{h,f}^\star - \hat{\phi}_h(x,a)^\top \hat{\xi}_{h,f} \right)^2 + \frac{24d \log\left( \frac{6\sqrt{d}|\Phi|}{\delta} \right)}{n}$$

$$\leq \frac{600d^{\frac{3}{2}}}{n} \log \frac{3dnHT|\Phi|}{\delta}$$

This implies there exists a representation $\hat{\phi}_{1:H}(x,a)$ such that for every $f \in \mathcal{F}$ and any $h \in [H]$, there exists $\hat{\xi}_{h,f}$ such that

$$\max_{\pi' \in \Psi_{1:H}^{\mathrm{cov}}} \mathbb{E}^{\pi'} \left[ \left( \phi_h^\star(x,a)^\top \xi_{h,f}^\star - \hat{\phi}_h(x,a)^\top \hat{\xi}_{h,f} \right)^2 \right] \leq \frac{600d^{\frac{3}{2}}}{n} \log \frac{3dnHT|\Phi|}{\delta}. \tag{36}$$

Eq. (36) matches Eq. (99) with a different error bound on the right hand. From Lemma G.1, we have $\Psi_h^{\mathrm{cov}}$ is a $\left( \frac{1}{8Ad}, \varepsilon \right)$-policy cover for layer $h$, following the rest of the proof in Lemma G.2, for every $\pi$ and every $f \in \mathcal{F}, h \in [H]$, we have

$$\left| \mathbb{E}^\pi \left[ \phi_h^\star(x,a)^\top \xi_{h,f}^\star - \hat{\phi}_h(x,a)^\top \hat{\xi}_{h,f} \right] \right| \leq \frac{75d^{\frac{5}{4}}}{\sqrt{n}} \sqrt{|\mathcal{A}| \log \frac{3dnHT|\Phi|}{\delta}} + 9d^{\frac{5}{2}} \epsilon. \tag{37}$$

The $d^{\frac{5}{2}}$ in the second term of Eq. (37) improves the $d^{\frac{7}{2}}$ term in Eq. (98) because $\hat{\xi}_{h,f} \in \mathbb{B}_d(\sqrt{d})$ rather than $\mathbb{B}_d(d^{\frac{3}{2}})$. Putting the choice $n = 11250d^{\frac{5}{2}}|\mathcal{A}|T^{\frac{2}{3}} \log \frac{3dnHT|\Phi|}{\delta}$ and $\epsilon = \frac{d^{\frac{5}{2}}}{18}T^{-\frac{1}{3}}$ into Eq. (37), we have for every $\pi$ and every $f \in \mathcal{F}, h \in [H]$, we have

$$\left| \mathbb{E}^\pi \left[ \phi_h^\star(x,a)^\top \xi_{h,f}^\star - \hat{\phi}_h(x,a)^\top \hat{\xi}_{h,f} \right] \right| \leq d^5 T^{-\frac{1}{3}} \tag{38}$$

Utilizing above results, for every $\pi$, every $f \in \mathcal{F}$ and any $h \in [H-1]$, we have

$$\left| \sum_x (\hat{d}_{h+1}^\pi(x) - d_{h+1}^\pi(x)) f(x) \right|$$

$$= \left| \sum_{x,a} \hat{d}_h^\pi(x) \pi_h(a \mid x) \hat{\phi}_h(x,a)^\top \hat{\xi}_{h,f} - \sum_{x,a} d_h^\pi(x) \pi(a \mid x) \phi_h^\star(x,a)^\top \xi_{h,f}^\star \right| \quad \text{(Eq. (31))}$$

$$\leq \left| \sum_{x,a} d_h^\pi(x) \pi_h(a \mid x) \left( \phi_h^\star(x,a)^\top \xi_{h,f}^\star - \hat{\phi}_h(x,a)^\top \hat{\xi}_{h,f} \right) \right|$$

$$+ \left| \sum_x \left( \hat{d}_h^\pi(x) - d_h^\pi(x) \right) \underbrace{\sum_a \pi_h(a|x) \hat{\phi}_h(x,a)^\top \hat{\xi}_{h,f}}_{\in \mathcal{F}} \right|$$

$$\leq d^5 T^{-\frac{1}{3}} + \left| \sum_x \left( \hat{d}_h^\pi(x) - d_h^\pi(x) \right) f'(x) \right| \quad \text{(Eq. (38))}$$

$\square$

where in the last step, we define $f'(x) = \sum_a \pi_h(a|x) \hat{\phi}_h(x,a)^\top \hat{\xi}_{h,f} \in \mathcal{F}$. This allow us to use recursion to finish the proof.

### D.3 Regret Analysis

We begin by proving Lemma D.3, showing that the policy class $\Pi_{\mathrm{lin}}^{\mathrm{cov}}(\epsilon')$ suffices to approximate all policies with small error.

**Lemma D.3.** *For any policy $\pi$, there exists a policy $\pi' \in \Pi_{\mathrm{lin}}^{\mathrm{cov}}(\epsilon')$ such that*

$$\sum_{t=1}^T V_1^{\pi'}(x_1; \ell^t) - \sum_{t=1}^T V_1^\pi(x_1; \ell^t) \leq H\epsilon'$$

**Proof.** Let $\theta_h^{t,\pi} \in \mathbb{B}_d(H\sqrt{d})$ be such that

$$\forall (x,a) \in \mathcal{X} \times \mathcal{A}, \quad Q_h^\pi(x,a; \ell^t) = \phi_h^\star(x,a)^\top \theta_h^{\pi,t}.$$

Such a $\theta_h^{\pi,t}$ is guaranteed to exist by the low-rank MDP structure (Assumption 2.1) and Assumption 2.2. For every $h \in [H]$, since $\sum_{t=1}^T \theta_h^{\pi,t} \in \mathbb{B}_d(\sqrt{d}HT)$, we define $\theta_h' \in \mathcal{C}\left(\mathbb{B}_d\left(\sqrt{d}HT\right), \epsilon'\right)$ such that $\|\theta_h' - \sum_{t=1}^T \theta_h^{\pi,t}\|_2 \le \epsilon'$, and let $\pi_h(a \mid x) = \mathbb{I}\{a = \mathrm{argmin}_{a\in\mathcal{A}} \phi_h^\star(x,a)^\top \theta_h'\}$ for every $h \in [H]$. We have $\pi' \in \Pi_{\mathrm{lin}}^{\mathrm{cov}}(\epsilon')$. From Lemma I.7, we have

$$\sum_{t=1}^T V_1^{\pi'}(x_1; \ell^t) - \sum_{t=1}^T V_1^\pi(x_1; \ell^t)$$

$$= \sum_{t=1}^T \sum_{h=1}^H \mathbb{E}_{x \sim d_h^{\pi'}}\left[\sum_{a\in\mathcal{A}}(\pi_h'(a|x) - \pi_h(a|x)) Q_h^\pi(x,a;\ell^t)\right]$$

$$= \sum_{h=1}^H \mathbb{E}_{x \sim d_h^{\pi'}}\left[\sum_{a\in\mathcal{A}}(\pi'(a|x) - \pi(a|x))\phi_h^\star(x,a)^\top \sum_{t=1}^T \theta_h^{\pi,t}\right]$$

$$= \underbrace{\sum_{h=1}^H \mathbb{E}_{x \sim d_h^{\pi'}}\left[\sum_{a\in\mathcal{A}}(\pi_h'(a|x) - \pi_h(a|x))\phi_h^\star(x,a)^\top \theta_h'\right]}_{\le 0} + \sum_{h=1}^H \mathbb{E}_{x \sim d_h^{\pi'}}\left[\sum_{a\in\mathcal{A}}(\pi_h'(a|x) - \pi_h(a|x))\phi_h^\star(x,a)^\top \left(\sum_{t=1}^T \theta_h^{\pi,t} - \theta_h'\right)\right]$$

$$\le H\epsilon'$$

where the last inequality comes from the fact that $\pi_h(a \mid x) = \mathbb{I}\{a = \mathrm{argmin}_{a\in\mathcal{A}} \phi_h^\star(x,a)^\top \theta_h'\}$ and $\|\theta_h' - \sum_{t=1}^T \theta_h^{\pi,t}\|_2 \le \epsilon'$ for every $h \in [H]$. $\qquad\square$

From Lemma D.3, for any policy $\pi_1^\star$, there exists a policy $\pi_0^\star \in \Pi_{\mathrm{lin}}^{\mathrm{cov}}(\frac{1}{T})$ such that

$$\mathrm{Reg}_T(\pi_1^\star) = \mathbb{E}\left[\sum_{t=1}^T V_1^{\boldsymbol{\pi}^t}(x_1; \ell^t) - \sum_{t=1}^T V_1^{\pi_1^\star}(x_1; \ell^t)\right]$$

$$\le \mathbb{E}\left[\sum_{t=1}^T V_1^{\boldsymbol{\pi}^t}(x_1; \ell^t) - \sum_{t=1}^T V_1^{\pi_0^\star}(x_1; \ell^t)\right] + 1 \tag{39}$$

Our candidate policy space $\Pi'$ has a $\beta$ mixture of the random policy. For any policy $\pi_0^\star \in \Pi_{\mathrm{lin}}^{\mathrm{cov}}(\frac{1}{T})$, define policy $\pi^\star$ such that for any state $x \in \mathcal{X}$, we have $\pi^\star(\cdot \mid x) = (1 - \beta)\pi_0^\star(\cdot \mid x) + \frac{\beta}{|\mathcal{A}|}$. We have $\pi^\star \in \Pi'$. Define

$$\widehat{V}_1^\pi(x_1; \ell) = \sum_{h=1}^H \sum_{x_h \in \mathcal{X}} \sum_{a_h \in \mathcal{A}} \hat{d}_h^\pi(x_h)\pi_h(a_h \mid x_h) \cdot \ell_h^t(x_h, a_h)$$

Utilizing Eq. (39) and the decomposition in Eq. (21), together with the fact that $|\Psi_h^{\mathrm{cov}}| \le d$ from Lemma G.1, we have for any policy $\pi_1^\star$,

$$\mathrm{Reg}_T(\pi_1^\star)$$

$$\le \underbrace{\mathbb{E}\left[\sum_{t=1}^T \sum_\pi p^t(\pi)\left(V_1^\pi(\boldsymbol{x}_1; \ell^t) - \widehat{V}_1^\pi(\boldsymbol{x}_1; \ell^t)\right)\right]}_{\textbf{Bias1}} + \underbrace{\mathbb{E}\left[\sum_{t=1}^T \widehat{V}_1^{\pi^*}(\boldsymbol{x}_1; \ell^t) - V_1^{\pi^*}(\boldsymbol{x}_1; \ell^t)\right]}_{\textbf{Bias2}}$$

$$+ \underbrace{\mathbb{E}\left[\sum_{t=1}^T \sum_\pi p^t(\pi)\widehat{V}_1^\pi(\boldsymbol{x}_1; \ell^t) - \widehat{V}_1^{\pi^*}(\boldsymbol{x}_1; \ell^t)\right]}_{\textbf{EXP}} + \underbrace{\mathbb{E}\left[\sum_{t=1}^T \sum_\pi \left(\rho^t(\pi) - p^t(\pi)\right)V_1^\pi(\boldsymbol{x}_1; \ell^t)\right]}_{\textbf{Error1}}$$

$$+ \underbrace{\mathbb{E}\left[\sum_{t=1}^T V_1^{\pi^*}(\boldsymbol{x}_1; \ell^t) - V_1^{\pi_0^\star}(\boldsymbol{x}_1; \ell^t)\right]}_{\textbf{Error2}} + \widetilde{O}\left(\epsilon^{-2}|\mathcal{A}|d^{13}H^6 \log(|\Phi|/\delta))\right) + dHn \tag{40}$$

Following Lemma C.2, we have

$$\textbf{\textit{Error1}} + \textbf{\textit{Error2}} \le H\gamma T + 2H^2\beta T \le 3H^2 T^{\frac{2}{3}}. \tag{41}$$

**Lemma D.4.**

$$\textbf{\textit{Bias1}} + \textbf{\textit{Bias2}} \le 2d^5 H^2 T^{\frac{2}{3}}$$

**Proof.** For any policy $\pi$ and any $t \in [T]$,

$$\left| V_1^\pi(\boldsymbol{x}_1; \ell^t) - \widehat{V}_1^\pi(\boldsymbol{x}_1; \ell^t) \right| = \sum_{h=1}^H \sum_{x_h \in \mathcal{X}} \left| d_h^\pi(x_h) - \hat{d}_h^\pi(x_h) \right| \sum_{a_h \in \mathcal{A}} \pi_h(a_h \mid x_h) \cdot \ell_h^t(x_h, a_h)$$

$$\le d^5 H^2 T^{-\frac{1}{3}} \qquad\qquad\qquad \text{(Lemma D.2)}$$

Thus,

$$\textbf{Bias1} \le d^5 H^2 T^{\frac{2}{3}} \qquad \text{and} \qquad \textbf{Bias2} \le d^5 H^2 T^{\frac{2}{3}}$$

$$\square$$

We now prove a modle-free counterpart of Lemma C.4 in Lemma D.5.

**Lemma D.5.** *For any episode $t \in [T]$, for any policy $\pi$ we have*

$$\widehat{V}_1^\pi(x_1; \ell^t) \le \mathbb{E}_{\boldsymbol{\pi}^t \sim \rho^t} \mathbb{E}^{\boldsymbol{\pi}^t} \left[ \hat{\ell}^t(\pi; \boldsymbol{\pi}^t, \boldsymbol{x}_{1:H}, \boldsymbol{a}_{1:H}) \right] + d^{\frac{11}{2}} H T^{-\frac{1}{3}} \sum_{h=2}^H \left\| \hat{\phi}_{h-1}(\pi) \right\|_{(\Sigma_{h-1}^t)^{-1}},$$

$$\widehat{V}_1^\pi(x_1; \ell^t) \ge \mathbb{E}_{\boldsymbol{\pi}^t \sim \rho^t} \mathbb{E}^{\boldsymbol{\pi}^t} \left[ \hat{\ell}^t(\pi; \boldsymbol{\pi}^t, \boldsymbol{x}_{1:H}, \boldsymbol{a}_{1:H}) \right] - d^{\frac{11}{2}} H T^{-\frac{1}{3}} \sum_{h=2}^H \left\| \hat{\phi}_{h-1}(\pi) \right\|_{(\Sigma_{h-1}^t)^{-1}}.$$

**Proof.**

We now prove the first inequality:

$$\widehat{V}_1^\pi(x_1; \ell^t)$$

$$= \sum_{h=1}^H \sum_{x_h \in \mathcal{X}} \sum_{a_h \in \mathcal{A}} \hat{d}_h^\pi(x_h) \pi_h(a_h \mid x_h) \cdot \ell_h^t(x_h, a_h),$$

$$= \underbrace{\sum_{a_1 \in \mathcal{A}} \pi_1(a_1 \mid x_1) \cdot \ell_1^t(x_1, a_1)}_{\textbf{First}} + \underbrace{\sum_{h=2}^H \sum_{x_h \in \mathcal{X}} \sum_{a_h \in \mathcal{A}} \hat{d}_h^\pi(x_h) \pi_h(a_h \mid x_h) \phi_h(x_h, a_h)^\top g_h^t}_{\textbf{Remain}} \qquad (42)$$

Through importance sampling, we have

$$\textbf{First} = \mathbb{E}_{\boldsymbol{\pi}^t \sim \rho^t} \mathbb{E}^{\boldsymbol{\pi}^t} \left[ \frac{\pi_1(\boldsymbol{a}_1 \mid \boldsymbol{x}_1)}{\pi_1^t(\boldsymbol{a}_1 \mid \boldsymbol{x}_1)} \cdot \ell_1^t(\boldsymbol{x}_1, \boldsymbol{a}_1) \right].$$

We now bound the remaining term in (42) Since $\Sigma_{h-1}^t = \mathbb{E}_{\boldsymbol{\pi}^t \sim \rho^t} \left[ \hat{\phi}_{h-1}(\boldsymbol{\pi}^t) \hat{\phi}_{h-1}(\boldsymbol{\pi}^t)^\top \right]$, we have

**Remain**

$$= \sum_{h=2}^H \sum_{x_h \in \mathcal{X}} \hat{d}_h^\pi(x_h) \underbrace{\sum_{a_h \in \mathcal{A}} \pi_h(a_h \mid x_h) \phi_h(x_h, a_h)^\top g_h^t}_{:= f(x_h)},$$

$$= \sum_{h=2}^H \sum_{x_{h-1} \in \mathcal{X}_{h-1}} \sum_{a_{h-1}} \hat{d}_{h-1}^\pi(x) \pi_{h-1}(a_{h-1} \mid x_{h-1}) \hat{\phi}(x_{h-1}, a_{h-1})^\top \hat{\xi}_{h-1,f} \qquad \text{(by Eq. (31))}$$

$$= \sum_{h=2}^H \hat{\phi}_{h-1}(\pi)^\top \hat{\xi}_{h-1,f}$$

$$= \sum_{h=2}^H \hat{\phi}_{h-1}(\pi)^\top (\Sigma_{h-1}^t)^{-1} \mathbb{E}_{\boldsymbol{\pi}^t \sim \rho^t} [\hat{\phi}_{h-1}(\boldsymbol{\pi}^t) \hat{\phi}_{h-1}(\boldsymbol{\pi}^t)^\top] \hat{\xi}_{h-1,f}$$

$$= \mathbb{E}_{\boldsymbol{\pi}^t \sim \rho^t}\left[\sum_{h=2}^{H} \hat{\phi}_{h-1}(\pi)^\top (\Sigma_{h-1}^t)^{-1} \hat{\phi}_{h-1}(\boldsymbol{\pi}^t) \hat{\phi}_{h-1}(\boldsymbol{\pi}^t)^\top \hat{\xi}_{h-1,f}\right]$$

$$= \mathbb{E}_{\boldsymbol{\pi}^t \sim \rho^t}\left[\sum_{h=2}^{H} \hat{\phi}_{h-1}(\pi)^\top (\Sigma_{h-1}^t)^{-1} \hat{\phi}_{h-1}(\boldsymbol{\pi}^t) \sum_{x_{h-1} \in \mathcal{X}_{h-1}} \sum_{a_{h-1}} \hat{d}_{h-1}^{\boldsymbol{\pi}^t}(x) \pi_{h-1}(a_{h-1}|x_{h-1}) \hat{\phi}(x_{h-1}, a_{h-1})^\top \hat{\xi}_{h-1,f}\right]$$

$$= \mathbb{E}_{\boldsymbol{\pi}^t \sim \rho^t}\left[\sum_{h=2}^{H} \hat{\phi}_{h-1}(\pi)^\top (\Sigma_{h-1}^t)^{-1} \hat{\phi}_{h-1}(\boldsymbol{\pi}^t) \sum_{x_h \in \mathcal{X}_h} \hat{d}_h^{\boldsymbol{\pi}^t}(x_h) f(x_h)\right]$$

$$= \mathbb{E}_{\boldsymbol{\pi}^t \sim \rho^t}\left[\sum_{h=2}^{H} \hat{\phi}_{h-1}(\pi)^\top (\Sigma_{h-1}^t)^{-1} \hat{\phi}_{h-1}(\boldsymbol{\pi}^t) \sum_{x_h \in \mathcal{X}_h} \sum_{a_h} \hat{d}_h^{\boldsymbol{\pi}^t}(x_h) \pi_h(a_h|x_h) \phi_h(x_h, a_h)^\top g_h^t\right]$$

$$= \mathbb{E}_{\boldsymbol{\pi}^t \sim \rho^t}\left[\sum_{h=2}^{H} \hat{\phi}_{h-1}(\pi)^\top (\Sigma_{h-1}^t)^{-1} \hat{\phi}_{h-1}(\boldsymbol{\pi}^t) \sum_{x_h \in \mathcal{X}_h} \sum_{a_h} d_h^{\boldsymbol{\pi}^t}(x_h) \pi_h(a_h|x_h) \phi_h(x_h, a_h)^\top g_h^t\right]$$

$$+ \mathbb{E}_{\boldsymbol{\pi}^t \sim \rho^t}\left[\sum_{h=2}^{H} \hat{\phi}_{h-1}(\pi)^\top (\Sigma_{h-1}^t)^{-1} \hat{\phi}_{h-1}(\boldsymbol{\pi}^t) \sum_{x_h \in \mathcal{X}_h} \sum_{a_h} \left(\hat{d}_h^{\boldsymbol{\pi}^t}(x_h) - d_h^{\boldsymbol{\pi}^t}(x_h)\right) \pi_h(a_h|x_h) \phi_h(x_h, a_h)^\top g_h^t\right]$$

$$= \mathbb{E}_{\boldsymbol{\pi}^t \sim \rho^t}\left[\sum_{h=2}^{H} \sum_{x_h \in \mathcal{X}} \sum_{a_h \in \mathcal{A}} \hat{\phi}_{h-1}(\pi)^\top \left(\Sigma_{h-1}^t\right)^{-1} \hat{\phi}_{h-1}(\boldsymbol{\pi}^t) d_h^{\boldsymbol{\pi}^t}(x_h) \pi_h(a_h \mid x_h) \ell_h^t(x_h, a_h)\right]$$

$$+ d^5 H T^{-\frac{1}{3}} \mathbb{E}_{\boldsymbol{\pi}^t \sim \rho^t}\left[\sum_{h=2}^{H} \sum_{x_h \in \mathcal{X}} \sum_{a_h \in \mathcal{A}} \left|\hat{\phi}_{h-1}(\pi)^\top \left(\Sigma_{h-1}^t\right)^{-1} \hat{\phi}_{h-1}(\boldsymbol{\pi}^t)\right|\right], \qquad \text{(by Lemma D.2)}$$

$$\leq \mathbb{E}_{\boldsymbol{\pi}^t \sim \rho^t}\left[\sum_{h=2}^{H} \sum_{x_h \in \mathcal{X}} \sum_{a_h \in \mathcal{A}} \hat{\phi}_{h-1}(\pi)^\top \left(\Sigma_{h-1}^t\right)^{-1} \hat{\phi}_{h-1}(\boldsymbol{\pi}^t) d_h^{\boldsymbol{\pi}^t}(x_h, a_h) \frac{\pi_h(a_h \mid x_h)}{\boldsymbol{\pi}_h^t(a_h \mid x_h)} \ell_h^t(x_h, a_h)\right]$$

$$+ d^5 H T^{-\frac{1}{3}} \sum_{h=2}^{H} \left\|\hat{\phi}_{h-1}(\pi)\right\|_{(\Sigma_{h-1}^t)^{-1}} \mathbb{E}_{\boldsymbol{\pi}^t \sim \rho^t}\left[\left\|\hat{\phi}_{h-1}(\boldsymbol{\pi}^t)\right\|_{(\Sigma_{h-1}^t)^{-1}}\right], \qquad (43)$$

$$\leq \mathbb{E}_{\boldsymbol{\pi}^t \sim \rho^t} \mathbb{E}^{\boldsymbol{\pi}^t}\left[\sum_{h=2}^{H} \hat{\phi}_{h-1}(\pi)^\top \left(\Sigma_{h-1}^t\right)^{-1} \hat{\phi}_{h-1}(\boldsymbol{\pi}^t) \frac{\pi_h(\boldsymbol{a}_h^t \mid \boldsymbol{x}_h^t)}{\boldsymbol{\pi}_h^t(\boldsymbol{a}_h^t \mid \boldsymbol{x}_h^t)} \ell_h^t(\boldsymbol{x}_h^t, \boldsymbol{a}_h^t)\right]$$

$$+ d^{\frac{11}{2}} H T^{-\frac{1}{3}} \sum_{h=2}^{H} \left\|\hat{\phi}_{h-1}(\pi)\right\|_{(\Sigma_{h-1}^t)^{-1}},$$

Adding up **First** and **Remain**, and using the defintion of $\hat{\ell}$ in (22) implies the first inequality of the lemma.

The second inequality follows the same procedure except for applying Cauchy-Schwarz inequality in the opposite direction in Eq. (43). $\qquad \square$

Given Lemma D.5, we could follow the same procedure in Lemma C.5 except replacing the factor of bonus $b^t(\pi)$ from $\sqrt{d} H \epsilon$ to $d^{\frac{11}{2}} H T^{-\frac{1}{3}}$. This leads to the following changes. Firstly, by a similar argument as Eq. (27), we have

$$\textbf{EXP} = \textbf{FTRL} + 4 d^6 H^2 T^{\frac{2}{3}}$$

where

$$\textbf{FTRL} := \sum_{t=1}^{T} \sum_{\pi} p^t(\pi) \cdot \left(\mathbb{E}_{\boldsymbol{\pi}^t \sim \rho^t} \mathbb{E}^{\boldsymbol{\pi}^t}\left[\hat{\ell}^t(\pi; \boldsymbol{\pi}^t, \boldsymbol{x}_{1:H}, \boldsymbol{a}_{1:H})\right] - b^t(\pi)\right)$$

$$- \sum_{t=1}^{T} \left(\mathbb{E}_{\boldsymbol{\pi}^t \sim \rho^t} \mathbb{E}^{\boldsymbol{\pi}^t}\left[\hat{\ell}^t(\pi^\star; \boldsymbol{\pi}^t, \boldsymbol{x}_{1:H}, \boldsymbol{a}_{1:H})\right] - b^t(\pi^\star)\right).$$

Secondly, now we have

$$\left|b^t(\pi)\right| = \left|d^{\frac{11}{2}} H T^{-\frac{1}{3}} \sum_{h=2}^{H} \left\|\hat{\phi}_{h-1}(\pi)\right\|_{(\Sigma_{h-1}^t)^{-1}}\right| \leq d^6 H^2 \frac{T^{-\frac{1}{3}}}{\sqrt{\gamma}} = d^6 H^2 T^{-\frac{1}{6}}. \qquad (\gamma = T^{-\frac{1}{3}})$$

To ensure $\eta\left|\hat{\ell}^t(\pi;\pi^t,x_{1:H},a_{1:H})-b^t(\pi)\right|\le 1$, it suffices to set $\eta\le\left(\frac{2|\mathcal{A}|Hd}{\beta\gamma}+d^6H^2T^{-\frac{1}{6}}\right)^{-1}=$ $\left(2|\mathcal{A}|HdT^{\frac{2}{3}}+d^6H^2T^{-\frac{1}{6}}\right)^{-1}$ from $\beta=\gamma=T^{-\frac{1}{3}}$. Thus our choice $\eta=(4Hd|\mathcal{A}|)^{-1}T^{-\frac{2}{3}}$ satisfies the condition if we assume $T\ge d^6H^2$. Moreover, now we have

$$
\begin{aligned}
\textbf{Stability-2} &= 2\eta\mathbb{E}\left[\sum_{t=1}^T\sum_{\pi\in\Pi'}p^t(\pi)b^t(\pi)^2\right],\\
&= 2\eta d^{11}H^3T^{-\frac{2}{3}}\mathbb{E}\left[\sum_{t=1}^T\sum_{h=2}^H\sum_{\pi\in\Pi'}p^t(\pi)\left\|\hat{\phi}_{h-1}(\pi)\right\|^2_{(\Sigma_{h-1}^t)^{-1}}\right],\\
&\le 4\eta d^{11}H^3T^{-\frac{2}{3}}\mathbb{E}\left[\sum_{t=1}^T\sum_{h=1}^{H-1}\sum_{\pi\in\Pi'}\rho^t(\pi)\left\|\hat{\phi}_{h-1}(\pi)\right\|^2_{(\Sigma_{h-1}^t)^{-1}}\right],\\
&\le 4\eta d^{11}H^4T^{\frac{1}{3}}\\
&\le d^{10}H^3T^{-\frac{1}{3}}\\
&\le d^4HT^{\frac{2}{3}}. \hspace{5cm} \text{(Assume } T\ge d^6H^2)
\end{aligned}
$$

Putting these two changes back to the proof into Lemma C.5, given $\eta=(4Hd|\mathcal{A}|)^{-1}T^{-\frac{2}{3}}$ we have

$$
\begin{aligned}
\textbf{EXP} &= \frac{\log|\Pi_{\text{lin}}^{\text{cov}}(\frac{1}{T})|}{\eta}+\frac{6d\eta H^2|\mathcal{A}|T}{\beta}+4\eta d^2H^4\epsilon^2T+d^4HT^{\frac{2}{3}}+4d^6H^2T^{\frac{2}{3}}\\
&= \widetilde{\mathcal{O}}\left(d^6|\mathcal{A}|H^2T^{\frac{2}{3}}\log(|\Phi|)\right) \hspace{4cm} (44)
\end{aligned}
$$

Putting Eq. (41), Lemma D.4, Eq. (44) into Eq. (40) together with $\epsilon=18^{-1}d^{\frac{5}{2}}T^{-\frac{1}{3}}$, we have

$$
\text{Reg}_T\le\widetilde{O}\left(d^8H^6|\mathcal{A}|T^{\frac{2}{3}}\log(|\Phi|))\right)
$$

# E    Proof of Theorem 4.1 (Model-Free, Banfit Feedback)

We start by introducing some notation. We let $\mathcal{I}^{(k)}$ denote the rounds in the $k$-th epoch:

$$\mathcal{I}^{(k)} \coloneqq \{T_0 + (k-1)\cdot N_{\mathrm{reg}} + 1, \ldots, T_0 + k \cdot N_{\mathrm{reg}}\}, \tag{45}$$

where $T_0$ is as in Line 1 of Algorithm 3. Throughout the analysis, we condition on the event

$$\mathcal{E} \coloneqq \mathcal{E}^{\mathrm{cov}}, \tag{46}$$

where $\mathcal{E}^{\mathrm{cov}}$ is as in Lemma G.1. Further, for any $k \in [K]$, let $\rho^{(k)}$ be the distribution of the random policy:

$$\mathbb{I}\{\boldsymbol{\zeta} = 0\} \cdot \widehat{\pi}^{(k)} + \mathbb{I}\{\boldsymbol{\zeta} = 1\} \cdot \boldsymbol{\pi} \circ_{\boldsymbol{h}} \pi_{\mathrm{unif}} \circ_{\boldsymbol{h}+1} \widehat{\pi}^{(k)}, \tag{47}$$

with $\boldsymbol{\zeta} \sim \mathrm{Ber}(\nu)$, $\boldsymbol{h} \sim \mathtt{unif}([H])$, and $\boldsymbol{\pi} \sim \mathtt{unif}(\Psi_{\boldsymbol{h}}^{\mathrm{cov}})$.

We start our analysis by applying the performance difference lemma.

**Applying the performance difference lemma.**    For any $k \in [K]$, $t \in \mathcal{I}^{(k)}$, and $\rho^{(k)}$ as just defined, we have

$$
\begin{aligned}
&\mathbb{E}_{\boldsymbol{\pi}\sim\rho^{(k)}}\mathbb{E}\left[V_1^{\boldsymbol{\pi}}(x_1;\ell^t)\right] - V_1^{\pi^\star}(x_1;\ell^t) \\
&= (1-\nu)\cdot\left(V_1^{\widehat{\pi}^{(k)}}(x_1;\ell^t) - V_1^{\pi^\star}(x_1;\ell^t)\right) \\
&\quad + \frac{\nu}{Hd}\sum_{h\in[H]}\sum_{\pi\in\Psi_h^{\mathrm{cov}}}\left(V_1^{\pi\circ_h\pi_{\mathrm{unif}}\circ_{h+1}\widehat{\pi}^{(k)}}(x_1;\ell^t) - V_1^{\pi^\star}(x_1;\ell^t)\right), \\
&\leq (1-\nu)\cdot\sum_{h=1}^{H}\mathbb{E}^{\pi^\star}\left[\sum_{a\in\mathcal{A}}\left(\widehat{\pi}_h^{(k)}(a\mid\boldsymbol{x}_h) - \pi_h^\star(a\mid\boldsymbol{x}_h)\right)\cdot Q_h^{\widehat{\pi}^{(k)}}(\boldsymbol{x}_h,a;\ell^t)\right] + H\nu, \\
&= (1-\nu)\cdot\sum_{h=1}^{H}\mathbb{E}^{\pi^\star}\left[\sum_{a\in\mathcal{A}}\left(\widehat{\pi}_h^{(k)}(a\mid\boldsymbol{x}_h) - \pi_h^\star(a\mid\boldsymbol{x}_h)\right)\cdot\widehat{Q}_h^{(k)}(\boldsymbol{x}_h,a)\right] + H\nu \\
&\quad + (1-\nu)\cdot\sum_{h=1}^{H}\mathbb{E}^{\pi^\star}\left[\sum_{a\in\mathcal{A}}\widehat{\pi}_h^{(k)}(a\mid\boldsymbol{x}_h)\cdot\left(Q_h^{\widehat{\pi}^{(k)}}(\boldsymbol{x}_h,a;\ell^t) - \widehat{Q}_h^{(k)}(\boldsymbol{x}_h,a)\right)\right] \\
&\quad + (1-\nu)\cdot\sum_{h=1}^{H}\mathbb{E}^{\pi^\star}\left[\sum_{a\in\mathcal{A}}\pi_h^\star(a\mid\boldsymbol{x}_h)\cdot\left(\widehat{Q}_h^{(k)}(\boldsymbol{x}_h,a) - Q_h^{\widehat{\pi}^{(k)}}(\boldsymbol{x}_h,a;\ell^t)\right)\right].
\end{aligned}
$$

Thus, by the triangle inequality, we have

$$
\begin{aligned}
&\sum_{t\in\mathcal{I}^{(k)}}\left(\mathbb{E}_{\boldsymbol{\pi}\sim\rho^{(k)}}\mathbb{E}\left[V_1^{\boldsymbol{\pi}}(x_1;\ell^t)\right] - V_1^{\pi^\star}(x_1;\ell^t)\right) \\
&\leq (1-\nu)\cdot\sum_{t\in\mathcal{I}^{(k)}}\sum_{h=1}^{H}\mathbb{E}^{\pi^\star}\left[\sum_{a\in\mathcal{A}}\left(\widehat{\pi}_h^{(k)}(a\mid\boldsymbol{x}_h) - \pi_h^\star(a\mid\boldsymbol{x}_h)\right)\cdot\widehat{Q}_h^{(k)}(\boldsymbol{x}_h,a)\right] + HN_{\mathrm{reg}}\nu, \\
&\quad + 2(1-\nu)\cdot\sum_{h=1}^{H}\max_{\pi'\in\Pi}\left|\sum_{t\in\mathcal{I}^{(k)}}\mathbb{E}^{\pi^\star}\left[\sum_{a\in\mathcal{A}}\pi'_h(a\mid\boldsymbol{x}_h)\cdot\left(Q_h^{\widehat{\pi}^{(k)}}(\boldsymbol{x}_h,a) - \widehat{Q}_h^{(k)}(\boldsymbol{x}_h,a)\right)\right]\right|. \tag{48}
\end{aligned}
$$

We start by bound the first term in the right-hand side of (48) (the regret term).

## E.1    Bounding the Regret Term

Fix $h \in [H]$ and define $\tilde{\pi}_h^\star(\cdot\mid x) \coloneqq (1-\gamma/T)\cdot\pi_h^\star(\cdot\mid x) + \gamma\pi_{\mathrm{unif}}(\cdot\mid x)/T$ for all $x$, where $\gamma$ is as in Algorithm 3. We have that $|\widehat{Q}_h^{(k)}(x,a)| = |\hat{\phi}_h^{(k)}(x,a)^\top\hat{\theta}_h^{(k)}| \leq H\sqrt{d}$ (since $\hat{\theta}_h^{(k)} \in \mathbb{B}_d(H\sqrt{d})$ and $\|\hat{\phi}_h^k(x,a)\| \leq 1$). By applying Lemma I.5, we have that for any $\eta \leq \frac{1}{H\sqrt{d}}$ and $x \in \mathcal{X}$:

$$
\begin{aligned}
&\sum_{k\in[K]}\sum_{t\in\mathcal{I}^{(k)}}\sum_{a\in\mathcal{A}}\left(\widehat{\pi}_h^{(k)}(a\mid x) - \pi_h^\star(a\mid x)\right)\cdot\widehat{Q}_h^{(k)}(x,a) \\
&\leq \sum_{k\in[K]}\sum_{t\in\mathcal{I}^{(k)}}\sum_{a\in\mathcal{A}}\left(\widehat{\pi}_h^{(k)}(a\mid x) - \tilde{\pi}_h^\star(a\mid x)\right)\cdot\widehat{Q}_h^{(k)}(x,a) + H\gamma,
\end{aligned}
$$

$$\leq \frac{N_{\text{reg}} \log(T/\gamma)}{\eta} + \eta \sum_{k=1}^{K} \sum_{t \in \mathcal{I}^{(k)}} \sum_{a \in \mathcal{A}} \widehat{\pi}_h^{(k)}(a \mid x) \cdot \widehat{Q}_h^{(k)}(x,a)^2 + H\gamma,$$

$$\leq \frac{N_{\text{reg}} \log(T/\gamma)}{\eta} + H^2 d\eta T + H\gamma, \tag{49}$$

### E.2 Bounding the Bias Term

Fix epoch $k \in [K]$, round $t \in \mathcal{I}^{(k)}$, layer $h \in [2 .. H]$, and $\pi' \in \Pi$. Further, let

$$\mathcal{X}_{h,\varepsilon} \coloneqq \left\{ x \in \mathcal{X} : \max_{\pi \in \Pi} d_h^{\pi}(x) \geq \varepsilon \cdot \|\mu_h^{\star}(x)\| \right\} \tag{50}$$

be the set of $\varepsilon$-*reachable* states. With this notation, we now bound the bias term in (48); we have

$$\left| \sum_{t \in \mathcal{I}^{(k)}} \mathbb{E}^{\pi^{\star}} \left[ \sum_{a \in \mathcal{A}} \pi_h'(a \mid \boldsymbol{x}_h) \cdot \left( Q_h^{\widehat{\pi}^{(k)}}(\boldsymbol{x}_h, a; \ell^t) - \widehat{Q}_h^{(k)}(\boldsymbol{x}_h, a) \right) \right] \right|$$

$$\leq \left| \sum_{t \in \mathcal{I}^{(k)}} \mathbb{E}^{\pi^{\star}} \left[ \mathbb{I}\{\boldsymbol{x}_h \notin \mathcal{X}_{h,\varepsilon}\} \sum_{a \in \mathcal{A}} \pi_h'(a \mid \boldsymbol{x}_h) \cdot \left( Q_h^{\widehat{\pi}^{(k)}}(\boldsymbol{x}_h, a; \ell^t) - \widehat{Q}_h^{(k)}(\boldsymbol{x}_h, a) \right) \right] \right|$$

$$+ \left| \sum_{t \in \mathcal{I}^{(k)}} \mathbb{E}^{\pi^{\star}} \left[ \mathbb{I}\{\boldsymbol{x}_h \in \mathcal{X}_{h,\varepsilon}\} \sum_{a \in \mathcal{A}} \pi_h'(a \mid \boldsymbol{x}_h) \cdot \left( Q_h^{\widehat{\pi}^{(k)}}(\boldsymbol{x}_h, a; \ell^t) - \widehat{Q}_h^{(k)}(\boldsymbol{x}_h, a) \right) \right] \right|,$$

and so by Lemma I.1,

$$\leq \left| \sum_{t \in \mathcal{I}^{(k)}} \mathbb{E}^{\pi^{\star}} \left[ \mathbb{I}\{\boldsymbol{x}_h \in \mathcal{X}_{h,\varepsilon}\} \sum_{a \in \mathcal{A}} \pi_h'(a \mid \boldsymbol{x}_h) \cdot \left( Q_h^{\widehat{\pi}^{(k)}}(\boldsymbol{x}_h, a; \ell^t) - \widehat{Q}_h^{(k)}(\boldsymbol{x}_h, a) \right) \right] \right| + 2N_{\text{reg}} H d^2 \varepsilon,$$

$$= N_{\text{reg}} \cdot \left| \mathbb{E}^{\pi^{\star}} \left[ \mathbb{I}\{\boldsymbol{x}_h \in \mathcal{X}_{h,\varepsilon}\} \sum_{a \in \mathcal{A}} \pi_h'(a \mid \boldsymbol{x}_h) \cdot \left( \frac{1}{N_{\text{reg}}} \sum_{t \in \mathcal{I}^{(k)}} Q_h^{\widehat{\pi}^{(k)}}(\boldsymbol{x}_h, a; \ell^t) - \widehat{Q}_h^{(k)}(\boldsymbol{x}_h, a) \right) \right] \right| + 2N_{\text{reg}} H d^2 \varepsilon.$$

Thus, by letting

$$\overline{Q}_h^{\widehat{\pi}^{(k)}}(\cdot, \cdot) \coloneqq \frac{1}{N_{\text{reg}}} \sum_{t \in \mathcal{I}^{(k)}} Q_h^{\widehat{\pi}^{(k)}}(\cdot, \cdot; \ell^t), \tag{51}$$

and using Jensen's inequality (twice), we get

$$\left| \sum_{t \in \mathcal{I}^{(k)}} \mathbb{E}^{\pi^{\star}} \left[ \sum_{a \in \mathcal{A}} \pi_h'(a \mid \boldsymbol{x}_h) \cdot \left( Q_h^{\widehat{\pi}^{(k)}}(\boldsymbol{x}_h, a; \ell^t) - \widehat{Q}_h^{(k)}(\boldsymbol{x}_h, a) \right) \right] \right|$$

$$\leq N_{\text{reg}} \cdot \mathbb{E}^{\pi^{\star}} \left[ \mathbb{I}\{\boldsymbol{x}_h \in \mathcal{X}_{h,\varepsilon}\} \sum_{a \in \mathcal{A}} \pi_h'(a \mid \boldsymbol{x}_h) \cdot \left| \overline{Q}_h^{\widehat{\pi}^{(k)}}(\boldsymbol{x}_h, a) - \widehat{Q}_h^{(k)}(\boldsymbol{x}_h, a) \right| \right] + 2N_{\text{reg}} H d^2 \varepsilon,$$

$$\leq N_{\text{reg}} \cdot \sqrt{\mathbb{E}^{\pi^{\star}} \left[ \mathbb{I}\{\boldsymbol{x}_h \in \mathcal{X}_{h,\varepsilon}\} \sum_{a \in \mathcal{A}} \pi_h'(a \mid \boldsymbol{x}_h) \cdot \left( \overline{Q}_h^{\widehat{\pi}^{(k)}}(\boldsymbol{x}_h, a) - \widehat{Q}_h^{(k)}(\boldsymbol{x}_h, a) \right)^2 \right]} + 2N_{\text{reg}} H d^2 \varepsilon,$$

$$\leq N_{\text{reg}} \cdot \sqrt{\mathbb{E}^{\pi^{\star}} \left[ \mathbb{I}\{\boldsymbol{x}_h \in \mathcal{X}_{h,\varepsilon}\} \cdot \max_{a \in \mathcal{A}} \left( \overline{Q}_h^{\widehat{\pi}^{(k)}}(\boldsymbol{x}_h, a) - \widehat{Q}_h^{(k)}(\boldsymbol{x}_h, a) \right)^2 \right]} + 2N_{\text{reg}} H d^2 \varepsilon,$$

and now by the fact that $\Psi_h^{\text{cov}}$ is an $(\frac{1}{8Ad}, \varepsilon)$ policy cover for layer $h$ (see Lemma G.1 and Definition 4.1):

$$\leq N_{\text{reg}} \cdot \sqrt{8Ad \max_{\pi \in \Psi_h^{\text{cov}}} \mathbb{E}^{\pi} \left[ \mathbb{I}\{\boldsymbol{x}_h \in \mathcal{X}_{h,\varepsilon}\} \cdot \max_{a \in \mathcal{A}} \left( \overline{Q}_h^{\widehat{\pi}^{(k)}}(\boldsymbol{x}_h, a) - \widehat{Q}_h^{(k)}(\boldsymbol{x}_h, a) \right)^2 \right]} + 2N_{\text{reg}} H d^2 \varepsilon,$$

$$\leq N_{\text{reg}} \cdot \sqrt{8A^2 d \sum_{\pi \in \Psi_h^{\text{cov}}} \mathbb{E}^{\pi \circ_h \pi_{\text{unif}}} \left[ \left( \overline{Q}_h^{\widehat{\pi}^{(k)}}(\boldsymbol{x}_h, \boldsymbol{a}_h) - \widehat{Q}_h^{(k)}(\boldsymbol{x}_h, \boldsymbol{a}_h) \right)^2 \right]} + 2N_{\text{reg}} H d^2 \varepsilon. \tag{52}$$

Next, we bound the regression error term in the right-hand side of (52).

### E.3 Bounding the Regression Error

**Lemma E.1.** *Let $\delta \in (0,1)$ and $\nu \in (0, 1/4)$ be given. There is an event $\mathcal{E}^{\mathrm{reg}}$ of probability at least $1 - 2\delta$ under which*

$$\sum_{\pi \in \Psi_h} \mathbb{E}^{\pi \circ_h \pi_{\mathrm{unif}}} \left[ \left( \hat{\phi}_h^{(k)}(\boldsymbol{x}_h, \boldsymbol{a}_h)^\top \hat{\theta}_h^{(k)} - \overline{Q}_h^{\widehat{\pi}^{(k)}}(\boldsymbol{x}_h, \boldsymbol{a}_h) \right)^2 \right] \le \varepsilon_{\mathrm{reg}}^2 := \frac{40 H^3 d \log(2|\mathcal{F}|/\delta)}{\nu N_{\mathrm{reg}}}. \tag{53}$$

**Proof.** Fix $\delta \in (0,1)$, $h \in [H]$ and $k \in [K]$, and let $(\boldsymbol{x}_h^t, \boldsymbol{a}_h^t, \boldsymbol{\zeta}^t, \boldsymbol{h}^t)$ be as in Algorithm 3. With this, define

$$\boldsymbol{I}_h^t := \mathbb{I}\{\boldsymbol{\zeta}^t = 0 \text{ or } \boldsymbol{h}^t \le h\}, \tag{54}$$

and note that $(\boldsymbol{x}_h^t, \boldsymbol{a}_h^t, \boldsymbol{I}_h^t)_{t \in \mathcal{I}^{(k)}}$ are identically and independently distributed. Further, for $t \in \mathcal{I}^{(k)}$ and $\pi \in \Pi$, let $\theta_h^{t, \pi} \in \mathbb{B}_d(H\sqrt{d})$ be such that

$$\forall (x, a) \in \mathcal{X} \times \mathcal{A}, \quad Q_h^\pi(x, a; \ell^t) = \phi_h^\star(x, a)^\top \theta_h^{\pi, t}.$$

Such a $\theta_h^{\pi, t}$ is guaranteed to exist by the low-rank MDP structure (Assumption 2.1) and Assumption 2.2. With this, note that for $\overline{Q}_h^{\widehat{\pi}^{(k)}}$ as in (51), we have

$$\overline{Q}_h^{\widehat{\pi}^{(k)}}(x, a) = \phi_h^\star(x, a)^\top \theta_h^{(k)}, \quad \text{where} \quad \theta_h^{(k)} := \frac{1}{N_{\mathrm{reg}}} \sum_{t \in \mathcal{I}^{(k)}} \theta_h^{\widehat{\pi}^{(k)}, t}. \tag{55}$$

For the rest of this proof, we let $\mathcal{F}$ be the function class

$$\mathcal{F} := \{ f : (x, a) \mapsto \phi_h(x, a)^\top \theta \mid \theta \in \mathcal{C} \cup \{\theta_h^{(k)}\}, \phi \in \Phi \},$$

where $\mathcal{C}$ is a minimal $(N_{\mathrm{reg}})^{-1}$-cover of $\mathbb{B}(H\sqrt{d})$ in $\|\cdot\|$ distance. Further, for $\tau \in \mathcal{I}^{(k)}$ we let

$$\boldsymbol{z}_h^\tau := \sum_{l=h}^H \ell_l^\tau(\boldsymbol{x}_l^\tau, \boldsymbol{a}_l^\tau);$$

$$\varepsilon_h^\tau := \sum_{l=h}^H \ell_l^\tau(\boldsymbol{x}_l^\tau, \boldsymbol{a}_l^\tau) - \frac{1}{N_{\mathrm{reg}}} \sum_{t \in \mathcal{I}^{(k)}} Q_h^{\widehat{\pi}^{(k)}}(\boldsymbol{x}_h^\tau, \boldsymbol{a}_h^\tau; \ell^t); \tag{56}$$

and

$$\widehat{L}(f) := \sum_{t \in \mathcal{I}^{(k)}} \boldsymbol{I}_h^t \cdot (f(\boldsymbol{x}_h^t, \boldsymbol{a}_h^t) - \boldsymbol{z}_h^t)^2, \tag{57}$$

for $f \in \mathcal{F}$. Finally, let $f_\star(x, a) := \phi_h^\star(x, a)^\top \theta_h^{(k)}$, where $\theta_h^{(k)}$ is as in (55) and $\hat{f}(x, a) := \widehat{Q}_h^{(k)}(x, a)$. With this, note that $f_\star$ and $\hat{f}$ satisfy $f_\star(\boldsymbol{x}_h^t, \boldsymbol{a}_h^t) = \boldsymbol{z}_h^t - \varepsilon_h^t$ and $\hat{f} \in \mathrm{argmin}_{f \in \mathcal{F}} \widehat{L}(f)$.

Now, since $\hat{f} \in \mathrm{argmin}_{f \in \mathcal{F}} \widehat{L}(f)$, we have

$$0 \ge \widehat{L}(\hat{f}) - \widehat{L}(f_\star) = \nabla \widehat{L}(f_\star)[\hat{f} - f_\star] + \|\hat{f} - f_\star\|^2, \tag{58}$$

where $\nabla$ denotes directional derivative and

$$\|\hat{f} - f_\star\|^2 := \sum_{t \in \mathcal{I}^{(k)}} \boldsymbol{I}_h^t \cdot (\hat{f}(\boldsymbol{x}_h^t, \boldsymbol{a}_h^t) - f_\star(\boldsymbol{x}_h^t, \boldsymbol{a}_h^t))^2.$$

Rearranging (58) and using that $f_\star(\boldsymbol{x}_h^t, \boldsymbol{a}_h^t) = \boldsymbol{z}_h^t - \varepsilon_h^t$, we get that

$$\|\hat{f} - f_\star\|^2 \le -\nabla \widehat{L}(f_\star)[\hat{f} - f_\star],$$
$$= 2 \sum_{t \in \mathcal{I}^{(k)}} \boldsymbol{I}_h^t \cdot (\boldsymbol{z}_h^t - f_\star(\boldsymbol{x}_h^t, \boldsymbol{a}_h^t))(\hat{f}(\boldsymbol{x}_h^t, \boldsymbol{a}_h^t) - f_\star(\boldsymbol{x}_h^t, \boldsymbol{a}_h^t)),$$
$$= 2 \sum_{t \in \mathcal{I}^{(k)}} \boldsymbol{I}_h^t \cdot \varepsilon_h^t \cdot (\hat{f}(\boldsymbol{x}_h^t, \boldsymbol{a}_h^t) - f_\star(\boldsymbol{x}_h^t, \boldsymbol{a}_h^t)). \tag{59}$$

We now bound the right-hand side of (59). For any $h \in [H]$, $k \in [K]$ and any $\hat{f}, f_\star \in \mathcal{F}$, we apply Lemma I.2 with

- $\mathfrak{F}^i = \sigma(\{(\boldsymbol{x}_h^j, \boldsymbol{a}_h^j, \boldsymbol{\varepsilon}_h^j, \boldsymbol{\zeta}^j) : j \le t_i\})$ where $t_i \coloneqq (k-1) \cdot N_{\text{reg}} + i$;

- The random variable $\boldsymbol{w}^i$ set as the difference
$$\boldsymbol{w}^i = \mathbb{I}\{\boldsymbol{\zeta}^{t_i} = 0 \text{ or } \boldsymbol{h}^{t_i} \le h\} \cdot \boldsymbol{\varepsilon}_h^{t_i} \cdot (\hat{f}(\boldsymbol{x}_h^{t_i}, \boldsymbol{a}_h^{t_i}) - f_\star(\boldsymbol{x}_h^{t_i}, \boldsymbol{a}_h^{t_i}))$$
$$- \mathbb{E}\left[\mathbb{I}\{\boldsymbol{\zeta}^{t_i} = 0 \text{ or } \boldsymbol{h}^{t_i} \le h\} \cdot \boldsymbol{\varepsilon}_h^{t_i} \cdot (\hat{f}(\boldsymbol{x}_h^{t_i}, \boldsymbol{a}_h^{t_i}) - f_\star(\boldsymbol{x}_h^{t_i}, \boldsymbol{a}_h^{t_i})) \mid \mathfrak{F}^{i-1}\right]$$
where $t_i \coloneqq (k-1) \cdot N_{\text{reg}} + i$;

- $n = N_{\text{reg}} = |\mathcal{I}^{(k)}|$.

- $R = 4H^2$; and

- $\lambda = 1/(16H^2)$;

to get that there is an event $\mathcal{E}$ of porbability at least $1 - \delta$ under which

$$\sum_{t \in \mathcal{I}^{(k)}} \boldsymbol{I}_h^t \cdot \boldsymbol{\varepsilon}_h^t \cdot (\hat{f}(\boldsymbol{x}_h^t, \boldsymbol{a}_h^t) - f_\star(\boldsymbol{x}_h^t, \boldsymbol{a}_h^t))$$
$$\le \sum_{t \in \mathcal{I}^{(k)}} \mathbb{E}_t[\boldsymbol{I}_h^t \cdot \boldsymbol{\varepsilon}_h^t \cdot (\hat{f}(\boldsymbol{x}_h^t, \boldsymbol{a}_h^t) - f_\star(\boldsymbol{x}_h^t, \boldsymbol{a}_h^t))]$$
$$+ \frac{1}{8H^2} \sum_{t \in \mathcal{I}^{(k)}} \mathbb{E}_t[\boldsymbol{I}_h^t \cdot (\boldsymbol{\varepsilon}_h^t)^2 \cdot (\hat{f}(\boldsymbol{x}_h^t, \boldsymbol{a}_h^t) - f_\star(\boldsymbol{x}_h^t, \boldsymbol{a}_h^t))^2] + 16H^2 \log(|\mathcal{F}|/\delta),$$
$$\le \sum_{t \in \mathcal{I}^{(k)}} \mathbb{E}_t[\boldsymbol{I}_h^t \cdot \boldsymbol{\varepsilon}_h^t \cdot (\hat{f}(\boldsymbol{x}_h^t, \boldsymbol{a}_h^t) - f_\star(\boldsymbol{x}_h^t, \boldsymbol{a}_h^t))]$$
$$+ \frac{1}{4} \sum_{t \in \mathcal{I}^{(k)}} \mathbb{E}_t[\boldsymbol{I}_h^t \cdot (\hat{f}(\boldsymbol{x}_h^t, \boldsymbol{a}_h^t) - f_\star(\boldsymbol{x}_h^t, \boldsymbol{a}_h^t))^2] + 16H^2 \log(|\mathcal{F}|/\delta), \tag{60}$$

where $\mathbb{E}_t[\cdot]$ is defined as $\mathbb{E}\left[\cdot \mid \mathfrak{F}^{t-1}\right]$ and the last step uses that $|\boldsymbol{\varepsilon}_h^t| \le 2H$, for all $t \in \mathcal{I}^{(k)}$. For the rest of the proof, we condition on $\mathcal{E}$ and to simplify notation let

$$\boldsymbol{\Delta}_h^t \coloneqq \hat{f}(\boldsymbol{x}_h^t, \boldsymbol{a}_h^t) - f_\star(\boldsymbol{x}_h^t, \boldsymbol{a}_h^t). \tag{61}$$

Using the expression of $\boldsymbol{\varepsilon}_h^t$ in (56), we have

$$\sum_{t \in \mathcal{I}^{(k)}} \mathbb{E}_t[\boldsymbol{I}_h^t \cdot \boldsymbol{\varepsilon}_h^t \cdot \boldsymbol{\Delta}_h^t]$$
$$= \sum_{t \in \mathcal{I}^{(k)}} \mathbb{E}_t\left[\boldsymbol{I}_h^t \cdot \left(\sum_{l=h}^{H} \ell_l^t(\boldsymbol{x}_l^t, \boldsymbol{a}_l^t) - \frac{1}{N_{\text{reg}}} \sum_{\tau \in \mathcal{I}^{(k)}} Q_h^{\widehat{\pi}^{(k)}}(\boldsymbol{x}_h^t, \boldsymbol{a}_h^t; \ell^\tau)\right) \cdot \boldsymbol{\Delta}_h^t\right]. \tag{62}$$

Now, since $\boldsymbol{I}_h^t = \mathbb{I}\{\boldsymbol{\zeta}^t = 0 \text{ or } \boldsymbol{h}^t \le h\}$, we have

$$\mathbb{E}_t\left[\sum_{l=h}^{H} \ell_l^t(\boldsymbol{x}_l^t, \boldsymbol{a}_l^t) \mid \boldsymbol{x}_h^t, \boldsymbol{a}_h^t, \boldsymbol{I}_h^t = 1\right] = Q_h^{\widehat{\pi}^{(k)}}(\boldsymbol{x}_h^t, \boldsymbol{a}_h^t; \ell^t).$$

Plugging this into (62) and using the law of total expectation, we have

$$\sum_{t \in \mathcal{I}^{(k)}} \mathbb{E}_t[\boldsymbol{I}_h^t \cdot \boldsymbol{\varepsilon}_h^t \cdot \boldsymbol{\Delta}_h^t]$$
$$= \sum_{t \in \mathcal{I}^{(k)}} \mathbb{E}_t\left[\boldsymbol{I}_h^t \cdot \left(Q_h^{\widehat{\pi}^{(k)}}(\boldsymbol{x}_h^t, \boldsymbol{a}_h^t; \ell^t) - \frac{1}{N_{\text{reg}}} \sum_{\tau \in \mathcal{I}^{(k)}} Q_h^{\widehat{\pi}^{(k)}}(\boldsymbol{x}_h^t, \boldsymbol{a}_h^t; \ell^\tau)\right) \cdot \boldsymbol{\Delta}_h^t\right],$$
$$= \sum_{t \in \mathcal{I}^{(k)}} \mathbb{E}_t\left[\boldsymbol{I}_h^t \cdot Q_h^{\widehat{\pi}^{(k)}}(\boldsymbol{x}_h^t, \boldsymbol{a}_h^t; \ell^t) \cdot \boldsymbol{\Delta}_h^t\right] - \frac{1}{N_{\text{reg}}} \sum_{\tau \in \mathcal{I}^{(k)}} \sum_{t \in \mathcal{I}^{(k)}} \mathbb{E}_t\left[\boldsymbol{I}_h^t \cdot Q_h^{\widehat{\pi}^{(k)}}(\boldsymbol{x}_h^t, \boldsymbol{a}_h^t; \ell^\tau) \cdot \boldsymbol{\Delta}_h^t\right]. \tag{63}$$

On the other hand, since $(\boldsymbol{x}_h^t, \boldsymbol{a}_h^t, \boldsymbol{I}_h^t)_{t \in \mathcal{I}^{(k)}}$ are i.i.d. and $\ell^t$ is chosen by an oblivious adversary, we have

$$\forall t \in \mathcal{I}^{(k)}, \quad \mathbb{E}_t\left[\boldsymbol{I}_h^t \cdot Q_h^{\widehat{\pi}^{(k)}}(\boldsymbol{x}_h^t, \boldsymbol{a}_h^t; \ell^t) \cdot \boldsymbol{\Delta}_h^t\right] = \frac{1}{N_{\text{reg}}} \sum_{\tau \in \mathcal{I}^{(k)}} \mathbb{E}_\tau\left[\boldsymbol{I}_h^\tau \cdot Q_h^{\widehat{\pi}^{(k)}}(\boldsymbol{x}_h^\tau, \boldsymbol{a}_h^\tau; \ell^t) \cdot \boldsymbol{\Delta}_h^\tau\right].$$

Plugging this into (63) shows that

$$\sum_{t \in \mathcal{I}^{(k)}} \mathbb{E}_t[\boldsymbol{I}_h^t \cdot \varepsilon_h^t \cdot \boldsymbol{\Delta}_h^t] = 0. \tag{64}$$

Combining this with (60) and (59), we get that

$$\sum_{t \in \mathcal{I}^{(k)}} \boldsymbol{I}_h^t \cdot (\hat{f}(\boldsymbol{x}_h^t, \boldsymbol{a}_h^t) - f_\star(\boldsymbol{x}_h^t, \boldsymbol{a}_h^t))^2 \leq \frac{1}{4} \sum_{t \in \mathcal{I}^{(k)}} \mathbb{E}_t[\boldsymbol{I}_h^t \cdot (\hat{f}(\boldsymbol{x}_h^t, \boldsymbol{a}_h^t) - f_\star(\boldsymbol{x}_h^t, \boldsymbol{a}_h^t))^2]$$
$$+ 16 H^2 \log(|\mathcal{F}|/\delta). \tag{65}$$

Now, since $(\boldsymbol{x}_h^t, \boldsymbol{a}_h^t, \boldsymbol{I}_h^t)_{t \in \mathcal{I}^{(k)}}$ are i.i.d., we have by Lemma I.3 that there is an event $\mathcal{E}'$ of probability at least $1 - \delta$ under which we have

$$\sum_{t \in \mathcal{I}^{(k)}} \mathbb{E}_t\left[\boldsymbol{I}_h^t \cdot (\hat{f}(\boldsymbol{x}_h^t, \boldsymbol{a}_h^t) - f_\star(\boldsymbol{x}_h^t, \boldsymbol{a}_h^t))^2\right] \leq 2 \sum_{t \in \mathcal{I}^{(k)}} \boldsymbol{I}_h^t \cdot (\hat{f}(\boldsymbol{x}_h^t, \boldsymbol{a}_h^t) - f_\star(\boldsymbol{x}_h^t, \boldsymbol{a}_h^t))^2$$
$$+ 8 H^2 \log(2|\mathcal{F}|/\delta). \tag{66}$$

Combining this with (65) and rearranging, we get that under $\mathcal{E}^{\mathrm{reg}} := \mathcal{E} \cap \mathcal{E}'$:

$$\frac{1}{N_{\mathrm{reg}}} \sum_{t \in \mathcal{I}^{(k)}} \mathbb{E}_t\left[\boldsymbol{I}_h^t \cdot (\hat{f}(\boldsymbol{x}_h^t, \boldsymbol{a}_h^t) - f_\star(\boldsymbol{x}_h^t, \boldsymbol{a}_h^t))^2\right] \leq \frac{40 H^2 \log(2|\mathcal{F}|/\delta)}{N_{\mathrm{reg}}}. \tag{67}$$

On the other hand, we have that for all $t \in \mathcal{I}^{(k)}$:

$$\mathbb{E}_t\left[\boldsymbol{I}_h^t \cdot (\hat{f}(\boldsymbol{x}_h^t, \boldsymbol{a}_h^t) - f_\star(\boldsymbol{x}_h^t, \boldsymbol{a}_h^t))^2\right] \geq \mathbb{E}_t\left[\mathbb{I}\{\boldsymbol{\zeta}^t = 1, \boldsymbol{h}^t = h\} \cdot (\hat{f}(\boldsymbol{x}_h^t, \boldsymbol{a}_h^t) - f_\star(\boldsymbol{x}_h^t, \boldsymbol{a}_h^t))^2\right],$$
$$= \frac{\nu}{Hd} \sum_{\pi \in \Psi_h} \mathbb{E}^{\pi \circ_h \pi_{\mathrm{unif}}}\left[(\hat{f}(\boldsymbol{x}_h^t, \boldsymbol{a}_h^t) - f_\star(\boldsymbol{x}_h^t, \boldsymbol{a}_h^t))^2\right]. \tag{68}$$

Plugging this into (67) and using the expressions of $\hat{f}$ and $f_\star$, we get

$$\sum_{\pi \in \Psi_h} \mathbb{E}^{\pi \circ_h \pi_{\mathrm{unif}}}\left[\left((\hat{\phi}_h^{(k)}(\boldsymbol{x}_h, \boldsymbol{a}_h)^\top \hat{\theta}_h^{(k)} - \overline{Q}_h^{\hat{\pi}^{(k)}}(\boldsymbol{x}_h, \boldsymbol{a}_h)\right)^2\right] \leq \frac{40 H^3 d \log(2|\mathcal{F}|/\delta)}{\nu N_{\mathrm{reg}}}.$$

$\square$

### E.4  Putting It All Together

By combining (48), (52), (53), and (49), we have that under the event $\mathcal{E} := \mathcal{E}^{\mathrm{cov}} \cap \mathcal{E}^{\mathrm{reg}}$ (where $\mathcal{E}^{\mathrm{cov}}$ and $\mathcal{E}^{\mathrm{reg}}$ are as in Lemma G.1 and Lemma E.1, respectively):

$$\mathrm{Reg}_T = H T_0 + \sum_{k \in [K]} \sum_{t \in \mathcal{I}^{(k)}} \left(\mathbb{E}_{\boldsymbol{\pi} \sim \rho^{(k)}} \mathbb{E}\left[V_1^{\boldsymbol{\pi}}(x_1; \ell^t)\right] - V_1^{\pi^\star}(x_1; \ell^t)\right),$$

$$\leq H T_0 + H T \nu + T \cdot \sqrt{8 A^2 d \cdot \varepsilon_{\mathrm{reg}}^2} + \frac{N_{\mathrm{reg}} \log(T/\gamma)}{\eta} + H^2 d \eta T + H \gamma. \tag{69}$$

Thus, plugging in the expression of $T_0$ from Algorithm 3, and ignoring polynormal factors $d, A, H, \log(|\Phi|\varepsilon^{-1}\delta^{-1})$, we get that

$$\mathrm{Reg}_T \prec \frac{1}{\varepsilon^2} + T\varepsilon + T\nu + N_{\mathrm{reg}} \cdot \sqrt{8 A^2 d \cdot \varepsilon_{\mathrm{reg}}^2} + 2 N_{\mathrm{reg}} H d\varepsilon + \frac{N_{\mathrm{reg}}}{\eta} + \eta T,$$

$$\prec T^{2/3} + \nu T + T \cdot \sqrt{8 A^2 d \cdot \varepsilon_{\mathrm{reg}}^2} + \sqrt{T N_{\mathrm{reg}}}, \quad \text{(by setting } \varepsilon = T^{-1/3} \text{ and } \eta = (N_{\mathrm{reg}}/T)^{1/2})$$

$$= T^{2/3} + \nu T + T \cdot \sqrt{\frac{1}{\nu N_{\mathrm{reg}}}} + \sqrt{T N_{\mathrm{reg}}}, \quad \text{(used the expression of } \varepsilon_{\mathrm{reg}}^2 \text{ in (53))}$$

$$\prec T^{2/3} + \nu T + \sqrt{T} \cdot \left(\frac{T}{\nu}\right)^{1/4}, \quad \text{(by setting } N_{\mathrm{reg}} = (T/\nu)^{1/2})$$

$$\prec T^{4/5}, \tag{70}$$

where the last step follows by setting $\nu = T^{-1/5}$.

# F   Proof of Theorem 4.2 (Model-Free, Bandit Feedback, Adaptive Adversary)

We let $\mathcal{I}^{(k)}$ denote the rounds in the $k$th epoch:

$$\mathcal{I}^{(k)} \coloneqq \{T_0 + (k-1) \cdot N_{\mathrm{reg}} + 1, \ \ldots, \ T_0 + k \cdot N_{\mathrm{reg}}\}, \tag{71}$$

where $T_0$ is as in Line 8 of Algorithm 4. Throughout the analysis, we condition on the event

$$\mathcal{E} \coloneqq \mathcal{E}^{\mathrm{cov}} \cap \mathcal{E}^{\mathrm{rep+span}} \cap \mathcal{E}^{\mathrm{freed}}, \tag{72}$$

where $\mathcal{E}^{\mathrm{cov}}$, $\mathcal{E}^{\mathrm{rep+span}}$, and $\mathcal{E}^{\mathrm{freed}}$ are as in Lemma G.1, Corollary F.1, and Lemma F.3, respectively.

We start our analysis by applying the performance difference lemma.

**Applying the performance difference lemma.**   For any $k \in [K]$, $t \in \mathcal{I}^{(k)}$, and let $\rho^{(k)}$ be the distribution of the random policy:

$$\mathbb{I}\{\boldsymbol{\zeta}^t = 0\} \cdot \widehat{\pi}^{(k)} + \mathbb{I}\{\boldsymbol{\zeta}^t = 1\} \cdot \boldsymbol{\pi}^t \circ_{\boldsymbol{h}^t+1} \widehat{\pi}^{(k)}, \tag{73}$$

with $\boldsymbol{\zeta}^t \sim \mathrm{Ber}(\nu)$, $\boldsymbol{h}^t \sim \mathrm{unif}([H])$, and $\boldsymbol{\pi}^t \sim \mathrm{unif}(\Psi_{\boldsymbol{h}^t}^{\mathrm{span}})$.

$$\mathbb{E}_{\boldsymbol{\pi} \sim \rho^{(k)}} \mathbb{E}\left[V_1^{\boldsymbol{\pi}}(x_1; \boldsymbol{\mathcal{H}}^{t-1})\right] - V_1^{\pi^\star}(x_1; \boldsymbol{\mathcal{H}}^{t-1})$$

$$= (1-\nu) \cdot \left(V_1^{\widehat{\pi}^{(k)}}(x_1; \boldsymbol{\mathcal{H}}^{t-1}) - V_1^{\pi^\star}(x_1; \boldsymbol{\mathcal{H}}^{t-1})\right)$$

$$+ \frac{\nu}{Hd} \sum_{h \in [H]} \sum_{\pi \in \Psi_h^{\mathrm{cov}}} \left(V_1^{\pi \circ_{h+1} \widehat{\pi}^{(k)}}(x_1; \boldsymbol{\mathcal{H}}^{t-1}) - V_1^{\pi^\star}(\boldsymbol{x}_1; \boldsymbol{\mathcal{H}}^{t-1})\right),$$

$$= (1-\nu) \cdot \sum_{h=1}^{H} \mathbb{E}^{\pi^\star}\left[\sum_{a \in \mathcal{A}} \left(\widehat{\pi}_h^{(k)}(a \mid \boldsymbol{x}_h) - \pi_h^\star(a \mid \boldsymbol{x}_h)\right) \cdot Q_h^{\widehat{\pi}^{(k)}}(\boldsymbol{x}_h, a; \boldsymbol{\mathcal{H}}^{t-1}) \mid \boldsymbol{\mathcal{H}}^{t-1}\right] + HN_{\mathrm{reg}}\nu,$$

$$= (1-\nu) \cdot \sum_{h=1}^{H} \mathbb{E}^{\pi^\star}\left[\sum_{a \in \mathcal{A}} \left(\widehat{\pi}_h^{(k)}(a \mid \boldsymbol{x}_h) - \pi_h^\star(a \mid \boldsymbol{x}_h)\right) \cdot \widehat{Q}_h^{(k)}(\boldsymbol{x}_h, a) \mid \boldsymbol{\mathcal{H}}^{t-1}\right] + HN_{\mathrm{reg}}\nu$$

$$+ (1-\nu) \cdot \sum_{h=1}^{H} \mathbb{E}^{\pi^\star}\left[\sum_{a \in \mathcal{A}} \widehat{\pi}_h^{(k)}(a \mid \boldsymbol{x}_h) \cdot \left(Q_h^{\widehat{\pi}^{(k)}}(\boldsymbol{x}_h, a; \boldsymbol{\mathcal{H}}^{t-1}) - \widehat{Q}_h^{(k)}(\boldsymbol{x}_h, a)\right) \mid \boldsymbol{\mathcal{H}}^{t-1}\right]$$

$$+ (1-\nu) \cdot \sum_{h=1}^{H} \mathbb{E}^{\pi^\star}\left[\sum_{a \in \mathcal{A}} \pi_h^\star(a \mid \boldsymbol{x}_h) \cdot \left(\widehat{Q}_h^{(k)}(\boldsymbol{x}_h, a) - Q_h^{\widehat{\pi}^{(k)}}(\boldsymbol{x}_h, a; \boldsymbol{\mathcal{H}}^{t-1})\right) \mid \boldsymbol{\mathcal{H}}^{t-1}\right].$$

Thus, by the triangle inequality, we have

$$\sum_{t \in \mathcal{I}^{(k)}} \left(\mathbb{E}_{\boldsymbol{\pi} \sim \rho^{(k)}} \mathbb{E}\left[V_1^{\boldsymbol{\pi}}(x_1; \boldsymbol{\mathcal{H}}^{t-1})\right] - V_1^{\pi^\star}(x_1; \boldsymbol{\mathcal{H}}^{t-1})\right)$$

$$\leq (1-\nu) \cdot \sum_{t \in \mathcal{I}^{(k)}} \sum_{h=1}^{H} \mathbb{E}^{\pi^\star}\left[\sum_{a \in \mathcal{A}} \left(\widehat{\pi}_h^{(k)}(a \mid \boldsymbol{x}_h) - \pi_h^\star(a \mid \boldsymbol{x}_h)\right) \cdot \widehat{Q}_h^{(k)}(\boldsymbol{x}_h, a) \mid \boldsymbol{\mathcal{H}}^{t-1}\right] + HN_{\mathrm{reg}}\nu$$

$$+ 2(1-\nu) \cdot \sum_{h=1}^{H} \max_{\pi' \in \Pi} \left|\sum_{t \in \mathcal{I}^{(k)}} \mathbb{E}^{\pi^\star}\left[\sum_{a \in \mathcal{A}} \pi_h'(a \mid \boldsymbol{x}_h) \cdot \left(Q_h^{\widehat{\pi}^{(k)}}(\boldsymbol{x}_h, a; \boldsymbol{\mathcal{H}}^{t-1}) - \widehat{Q}_h^{(k)}(\boldsymbol{x}_h, a)\right) \mid \boldsymbol{\mathcal{H}}^{t-1}\right]\right|. \tag{74}$$

We start by bounding the first term on the right-hand side of (74) (the regret term).

## F.1   Bounding the Regret Term

Fix $h \in [H]$ and define $\tilde{\pi}_h^\star(\cdot \mid x) \coloneqq (1 - \gamma/T) \cdot \pi_h^\star(\cdot \mid x) + \gamma \pi_{\mathrm{unif}}(\cdot \mid x)/T$ for all $x$, where $\gamma$ is as in Algorithm 4. We have that $|\widehat{Q}_h^{(k)}(x, a)| = |\bar{\phi}_h^{\mathrm{rep}}(x, a)^\top \hat{\theta}_h^{(k)}| \leq 8Hd^2$ (since $\hat{\theta}_h^{(k)} \in \mathbb{B}_{2d}(4Hd^2)$ and $\|\bar{\phi}_h^{\mathrm{rep}}(x, a)\| \leq \|\phi_h^{\mathrm{loss}}(x, a)\| + \|\phi_h^{\mathrm{rep}}(x, a)\| \leq 2$). By applying I.5, we have that for any $\eta \leq \frac{1}{8Hd^2}$ and $x \in \mathcal{X}$:

$$\sum_{k \in [K]} \sum_{t \in \mathcal{I}^{(k)}} \sum_{a \in \mathcal{A}} \left(\widehat{\pi}_h^{(k)}(a \mid x) - \pi_h^\star(a \mid x)\right) \cdot \widehat{Q}_h^{(k)}(x, a)$$

$$\leq \sum_{k \in [K]} \sum_{t \in \mathcal{I}^{(k)}} \sum_{a \in \mathcal{A}} \left(\widehat{\pi}_h^{(k)}(a \mid x) - \tilde{\pi}_h^\star(a \mid x)\right) \cdot \widehat{Q}_h^{(k)}(x, a) + H\gamma,$$

$$\leq \frac{N_{\text{reg}}\log(T/\gamma)}{\eta} + \eta \sum_{k=1}^{K} \sum_{t\in\mathcal{I}^{(k)}} \sum_{a\in\mathcal{A}} \widehat{\pi}_h^{(k)}(a\mid x)\cdot\widehat{Q}_h^{(k)}(x,a)^2 + H\gamma,$$

$$\leq \frac{N_{\text{reg}}\log(T/\gamma)}{\eta} + 64H^2 d^4 \eta T + H\gamma, \tag{75}$$

### F.2 Bounding the Bias Term

To bound the bias term (second term in (74)), we make use of the following result.

**Lemma F.1.** *For $t\in[T]$, $h\in[H]$, and $\pi\in\Pi$, there exists $\bar{\theta}_h^{t,\pi}\in\mathbb{B}_{2d}(H\sqrt{d})$ such that for all $(x,a)\in\mathcal{X}\times\mathcal{A}$ and history $\mathcal{H}^{t-1} = (x_{1:H}^{1:t-1}, a_{1:H}^{1:t-1})$,*

$$Q_h^{\pi}(x,a;\mathcal{H}^{t-1}) = \bar{\phi}_h^{\star}(x,a)^{\top}\bar{\theta}_h^{t,\pi}, \quad \text{where} \quad \bar{\phi}_h^{\star} := [\phi^{\text{loss}}, \phi^{\star}] \in \mathbb{R}^{2d}. \tag{76}$$

**Proof.** Fix $t\in[T]$, $h\in[H]$, and $\pi\in\Pi$. By the low-rank MDP structure and the normalizing assumption on $\mu_h^{\star}$ in Assumption 2.1, there exists $w_{h+1}^{t,\pi}\in\mathbb{B}_d((H-s)\sqrt{d})$ such that for all $(x,a)\in\mathcal{X}\times\mathcal{A}$ and history $\mathcal{H}^{t-1} = (x_{1:H}^{1:t-1}, a_{1:H}^{1:t-1})$,

$$\mathbb{E}^{\pi}\left[\sum_{s=h+1}^{H}\ell_s(\boldsymbol{x}_s,\boldsymbol{a}_s;\mathcal{H}^{t-1})\mid \boldsymbol{x}_h = x, \boldsymbol{a}_h = a\right] = \phi_h^{\star}(x,a)^{\top}w_{h+1}^{t,\pi}. \tag{77}$$

Now, with $g_h^t$ as in Assumption 4.2, (15) and (77) implies that $\bar{\theta}_h^{t,\pi} := [g_h^t, w_{h+1}^{t,\pi}]\in\mathbb{R}^{2d}$ (where $[\cdot,\cdot]$ denotes the vertical stacking of vectors) satisfies the desired property. $\square$

Fix epoch $k\in[K]$, round $t\in\mathcal{I}^{(k)}$, layer $h\in[2..H]$, and $\pi'\in\Pi$. By Lemma F.1, there exists $\bar{\theta}_h^{t,\widehat{\pi}^{(k)}}\in\mathbb{B}_{2d}(Hd^{1/2})$ such that for all $(x,a)\in\mathcal{X}\times\mathcal{A}$,

$$Q_{h+1}^{\widehat{\pi}^{(k)}}(x,a;\mathcal{H}^{t-1}) = \bar{\phi}_{h+1}^{\star}(x,a)^{\top}\bar{\theta}_{h+1}^{t,\widehat{\pi}^{(k)}}.$$

With this, let $\boldsymbol{w}_{h+1}^{t,\widehat{\pi}^{(k)}}\in\mathbb{B}_d(3Hd^2)$ be as in Corollary F.1 with $f(x) := \frac{1}{Hd^{1/2}}\max_{a\in\mathcal{A}}\bar{\phi}_{h+1}^{\star}(x,a)^{\top}\bar{\theta}_{h+1}^{t,\widehat{\pi}^{(k)}}$; note that this function belongs to the function class $\mathcal{F}_{h+1}$ in Algorithm 4. With this, we define

$$\bar{\boldsymbol{\vartheta}}_h^{t,\widehat{\pi}^{(k)}} := \left[\boldsymbol{g}_h^t, \boldsymbol{w}_{h+1}^{t,\widehat{\pi}^{(k)}}\right]\in\mathbb{B}_{2d}(4Hd^2), \tag{78}$$

where $\boldsymbol{g}_h^t\in\mathbb{B}_d(1)$ is such that $\ell_h(x,a;\mathcal{H}^{t-1}) = \phi_h^{\text{loss}}(x,a)^{\top}\boldsymbol{g}_h^t$.

With this notation, we now bound the bias term in Eq. (74): we have

$$\left|\sum_{t\in\mathcal{I}^{(k)}}\mathbb{E}^{\pi^{\star}}\left[\sum_{a\in\mathcal{A}}\pi_h'(a\mid\boldsymbol{x}_h)\cdot\left(Q_h^{\widehat{\pi}^{(k)}}(\boldsymbol{x}_h,a;\mathcal{H}^{t-1}) - \widehat{Q}_h^{(k)}(\boldsymbol{x}_h,a)\right)\mid\mathcal{H}^{t-1}\right]\right|$$

$$\leq \left|\sum_{t\in\mathcal{I}^{(k)}}\mathbb{E}^{\pi^{\star}}\left[\sum_{a\in\mathcal{A}}\pi_h'(a\mid\boldsymbol{x}_h)\cdot\left(\bar{\phi}_h^{\text{rep}}(\boldsymbol{x}_h,a)^{\top}\bar{\boldsymbol{\vartheta}}_h^{t,\widehat{\pi}^{(k)}} - \widehat{Q}_h^{(k)}(\boldsymbol{x}_h,a)\right)\mid\mathcal{H}^{t-1}\right]\right|$$

$$+ \left|\sum_{t\in\mathcal{I}^{(k)}}\mathbb{E}^{\pi^{\star}}\left[\sum_{a\in\mathcal{A}}\pi_h'(a\mid\boldsymbol{x}_h)\cdot\left(Q_h^{\widehat{\pi}^{(k)}}(\boldsymbol{x}_h,a;\mathcal{H}^{t-1}) - \bar{\phi}_h^{\text{rep}}(\boldsymbol{x}_h,a)^{\top}\bar{\boldsymbol{\vartheta}}_h^{t,\widehat{\pi}^{(k)}}\right)\mid\mathcal{H}^{t-1}\right]\right|,$$

and by Corollary F.1 (in particular (92))

$$\leq \left|\sum_{t\in\mathcal{I}^{(k)}}\mathbb{E}^{\pi^{\star}}\left[\sum_{a\in\mathcal{A}}\pi_h'(a\mid\boldsymbol{x}_h)\cdot\left(\bar{\phi}_h^{\text{rep}}(\boldsymbol{x}_h,a)^{\top}\bar{\boldsymbol{\vartheta}}_h^{t,\widehat{\pi}^{(k)}} - \bar{\phi}_h^{\text{rep}}(\boldsymbol{x}_h,a)^{\top}\hat{\theta}_h^{(k)}\right)\mid\mathcal{H}^{t-1}\right]\right| + N_{\text{reg}}\cdot\varepsilon_{\text{rep}},$$

and by Corollary F.1 again (in particular (93)) and the triangle inequality

$$\leq 2\sum_{\pi\in\Psi_h^{\text{span}}}\left|\sum_{t\in\mathcal{I}^{(k)}}\mathbb{E}^{\pi}\left[\bar{\phi}_h^{\text{rep}}(\boldsymbol{x}_h,\boldsymbol{a}_h)^{\top}\bar{\boldsymbol{\vartheta}}_h^{t,\widehat{\pi}^{(k)}} - \bar{\phi}_h^{\text{rep}}(\boldsymbol{x}_h,\boldsymbol{a}_h)^{\top}\hat{\theta}_h^{(k)}\mid\mathcal{H}^{t-1}\right]\right| + N_{\text{reg}}\cdot\varepsilon_{\text{span}} + N_{\text{reg}}\cdot\varepsilon_{\text{rep}},$$

$$\leq 2\sum_{\pi\in\Psi_h^{\text{span}}}\left|\sum_{t\in\mathcal{I}^{(k)}}\mathbb{E}^{\pi}\left[Q_h^{\widehat{\pi}^{(k)}}(\boldsymbol{x}_h,a;\mathcal{H}^{t-1}) - \bar{\phi}_h^{\text{rep}}(\boldsymbol{x}_h,\boldsymbol{a}_h)^{\top}\hat{\theta}_h^{(k)}\mid\mathcal{H}^{t-1}\right]\right| + N_{\text{reg}}\cdot\varepsilon_{\text{span}} + N_{\text{reg}}\cdot\varepsilon_{\text{rep}}$$

$$+ 2 \sum_{\pi \in \Psi_h^{\mathrm{span}}} \left| \sum_{t \in \mathcal{I}^{(k)}} \mathbb{E}^\pi \left[ \bar{\phi}_h^{\mathrm{rep}}(\boldsymbol{x}_h, \boldsymbol{a}_h)^\top \bar{\boldsymbol{\vartheta}}_h^{t, \widehat{\pi}^{(k)}} - Q_h^{\widehat{\pi}^{(k)}}(\boldsymbol{x}_h, a; \mathcal{H}^{t-1}) \mid \mathcal{H}^{t-1} \right] \right|, \quad \text{(triangle inequality)}$$

$$\leq 2 \sum_{\pi \in \Psi_h^{\mathrm{span}}} \left| \sum_{t \in \mathcal{I}^{(k)}} \mathbb{E}^\pi \left[ Q_h^{\widehat{\pi}^{(k)}}(\boldsymbol{x}_h, a; \mathcal{H}^{t-1}) - \bar{\phi}_h^{\mathrm{rep}}(\boldsymbol{x}_h, \boldsymbol{a}_h)^\top \hat{\theta}_h^{(k)} \mid \mathcal{H}^{t-1} \right] \right| + N_{\mathrm{reg}} \cdot \varepsilon_{\mathrm{span}} + 3 d N_{\mathrm{reg}} \cdot \varepsilon_{\mathrm{rep}},$$

$$\tag{79}$$

where the last inequality follows by Corollary F.1 (in particular (92)).

Next, we bound the estimation error term in the right-hand side of (79).

### F.3  Bound the Regression Error

By Lemma F.3 (Freedman's inequality), we have that for all $\pi \in \Psi_h^{\mathrm{span}}$ and $\bar{\theta} \in \mathbb{B}_{2d}(4Hd^2)$:

$$N_{\mathrm{reg}} \cdot \varepsilon_{\mathrm{freed}} \geq \left| \sum_{t \in \mathcal{I}^{(k)}} \mathbb{I}\{\boldsymbol{h}^t = h, \boldsymbol{\pi}^t = \pi, \boldsymbol{\zeta}^t = 1\} \cdot \left( \bar{\phi}_h^{\mathrm{rep}}(\boldsymbol{x}_h^t, \boldsymbol{a}_h^t)^\top \bar{\theta} - \sum_{s=h}^H \boldsymbol{\ell}_s^t \right) \right.$$

$$\left. - \sum_{t \in \mathcal{I}^{(k)}} \mathbb{E}\left[ \mathbb{I}\{\boldsymbol{h}^t = h, \boldsymbol{\pi}^t = \pi, \boldsymbol{\zeta}^t = 1\} \cdot \left( \bar{\phi}_h^{\mathrm{rep}}(\boldsymbol{x}_h^t, \boldsymbol{a}_h^t)^\top \bar{\theta} - \sum_{s=h}^H \boldsymbol{\ell}_s^t \right) \mid \mathcal{H}^{t-1} \right] \right|. \tag{80}$$

On the other hand, we have

$$\sum_{t \in \mathcal{I}^{(k)}} \mathbb{E}\left[ \mathbb{I}\{\boldsymbol{h}^t = h, \boldsymbol{\pi}^t = \pi, \boldsymbol{\zeta}^t = 1\} \cdot \left( \bar{\phi}_h^{\mathrm{rep}}(\boldsymbol{x}_h^t, \boldsymbol{a}_h^t)^\top \bar{\theta} - \sum_{s=h}^H \boldsymbol{\ell}_s^t \right) \mid \mathcal{H}^{t-1} \right]$$

$$= \frac{\nu}{Hd} \sum_{t \in \mathcal{I}^{(k)}} \mathbb{E}^{\pi \circ_{h+1} \widehat{\pi}^{(k)}} \left[ \bar{\phi}_h^{\mathrm{rep}}(\boldsymbol{x}_h, \boldsymbol{a}_h)^\top \bar{\theta} - \sum_{s=h}^H \boldsymbol{\ell}_s^t \mid \mathcal{H}^{t-1} \right],$$

$$= \frac{\nu}{Hd} \sum_{t \in \mathcal{I}^{(k)}} \mathbb{E}^\pi \left[ \bar{\phi}_h^{\mathrm{rep}}(\boldsymbol{x}_h, \boldsymbol{a}_h)^\top \bar{\theta} - Q_h^{\widehat{\pi}^{(k)}}(\boldsymbol{x}_h, a; \mathcal{H}^{t-1}) \mid \mathcal{H}^{t-1} \right]. \tag{81}$$

Now, by Corollary F.1 (in particular (92)) and the triangle inequality, we have

$$N_{\mathrm{reg}} \cdot \varepsilon_{\mathrm{rep}} \geq \left| \sum_{t \in \mathcal{I}^{(k)}} \mathbb{E}^\pi \left[ \bar{\phi}_h^{\mathrm{rep}}(\boldsymbol{x}_h, \boldsymbol{a}_h)^\top \bar{\theta} - Q_h^{\widehat{\pi}^{(k)}}(\boldsymbol{x}_h, a; \mathcal{H}^{t-1}) \mid \mathcal{H}^{t-1} \right] \right.$$

$$\left. - \sum_{t \in \mathcal{I}^{(k)}} \mathbb{E}^\pi \left[ \bar{\phi}_h^{\mathrm{rep}}(\boldsymbol{x}_h, \boldsymbol{a}_h)^\top \bar{\theta} - \bar{\phi}_h^{\mathrm{rep}}(\boldsymbol{x}_h, \boldsymbol{a}_h)^\top \bar{\boldsymbol{\vartheta}}_h^{t, \widehat{\pi}^{(k)}} \mid \mathcal{H}^{t-1} \right] \right|,$$

$$= \left| \sum_{t \in \mathcal{I}^{(k)}} \mathbb{E}^\pi \left[ \bar{\phi}_h^{\mathrm{rep}}(\boldsymbol{x}_h, \boldsymbol{a}_h)^\top \bar{\theta} - Q_h^{\widehat{\pi}^{(k)}}(\boldsymbol{x}_h, a; \mathcal{H}^{t-1}) \mid \mathcal{H}^{t-1} \right] \right.$$

$$\left. - \mathbb{E}^\pi \left[ \bar{\phi}_h^{\mathrm{rep}}(\boldsymbol{x}_h, \boldsymbol{a}_h)^\top \right] \left( N_{\mathrm{reg}} \cdot \bar{\theta} - \sum_{t \in \mathcal{I}^{(k)}} \bar{\boldsymbol{\vartheta}}_h^{t, \widehat{\pi}^{(k)}} \right) \right|. \tag{82}$$

Thus, by combining (80), (81), and (82), we have

$$\sum_{\pi \in \Psi_h^{\mathrm{span}}} \left| \sum_{t \in \mathcal{I}^{(k)}} \mathbb{I}\{\boldsymbol{h}^t = h, \boldsymbol{\pi}^t = \pi, \boldsymbol{\zeta}^t = 1\} \cdot \left( \bar{\phi}_h^{\mathrm{rep}}(\boldsymbol{x}_h^t, \boldsymbol{a}_h^t)^\top \bar{\theta} - \sum_{s=h}^H \boldsymbol{\ell}_s^t \right) \right|$$

$$\leq d N_{\mathrm{reg}} \cdot \varepsilon_{\mathrm{freed}} + \frac{\nu N_{\mathrm{reg}} \varepsilon_{\mathrm{rep}}}{H} + \frac{\nu}{Hd} \sum_{\pi \in \Psi_h^{\mathrm{span}}} \left| \sum_{t \in \mathcal{I}^{(k)}} \mathbb{E}^\pi \left[ \bar{\phi}_h^{\mathrm{rep}}(\boldsymbol{x}_h, \boldsymbol{a}_h)^\top \right] \left( N_{\mathrm{reg}} \cdot \bar{\theta} - \sum_{t \in \mathcal{I}^{(k)}} \bar{\boldsymbol{\vartheta}}_h^{t, \widehat{\pi}^{(k)}} \right) \right|.$$

Using that $\hat{\theta}_h^{(k)}$ is the minimizer over $\mathbb{B}_{2d}(4Hd^2)$ of the left-hand side, and the right-hand side evaluated to $d N_{\mathrm{reg}} \cdot \varepsilon_{\mathrm{freed}} + \frac{\nu N_{\mathrm{reg}} \varepsilon_{\mathrm{rep}}}{H}$ with $\bar{\theta} = \frac{1}{N_{\mathrm{reg}}} \sum_{t \in \mathcal{I}^{(k)}} \bar{\boldsymbol{\vartheta}}_h^{t, \widehat{\pi}^{(k)}} \in \mathbb{B}_{2d}(4Hd^2)$, we have that

$$\sum_{\pi \in \Psi_h^{\mathrm{span}}} \left| \sum_{t \in \mathcal{I}^{(k)}} \mathbb{I}\{\boldsymbol{h}^t = h, \boldsymbol{\pi}^t = \pi, \boldsymbol{\zeta}^t = 1\} \cdot \left( \bar{\phi}_h^{\mathrm{rep}}(\boldsymbol{x}_h^t, \boldsymbol{a}_h^t)^\top \hat{\theta}_h^{(k)} - \sum_{s=h}^H \boldsymbol{\ell}_s^t \right) \right| \leq d N_{\mathrm{reg}} \cdot \varepsilon_{\mathrm{freed}} + \frac{\nu N_{\mathrm{reg}} \varepsilon_{\mathrm{rep}}}{H}.$$

Now, combining this with (80) and (81) with $\bar{\theta} = \hat{\theta}_k^{(k)}$, we get that

$$
\sum_{\pi \in \Psi_h^{\mathrm{span}}} \left| \sum_{t \in \mathcal{I}^{(k)}} \mathbb{E}^\pi \left[ \bar{\phi}_h^{\mathrm{rep}}(\boldsymbol{x}_h, \boldsymbol{a}_h)^\top \hat{\theta}_h^{(k)} - Q_h^{\hat{\pi}^{(k)}}(\boldsymbol{x}_h, a; \boldsymbol{\mathcal{H}}^{t-1}) \mid \boldsymbol{\mathcal{H}}^{t-1} \right] \right|
$$

$$
\leq \frac{Hd^2 N_{\mathrm{reg}} \varepsilon_{\mathrm{freed}}}{\nu} + \frac{Hd}{\nu} \sum_{\pi \in \Psi_h^{\mathrm{span}}} \left| \sum_{t \in \mathcal{I}^{(k)}} \mathbb{I}\{\boldsymbol{h}^t = h, \boldsymbol{\pi}^t = \pi, \boldsymbol{\zeta}^t = 1\} \cdot \left( \bar{\phi}_h^{\mathrm{rep}}(\boldsymbol{x}_h^t, \boldsymbol{a}_h^t)^\top \hat{\theta}_h^{(k)} - \sum_{s=h}^H \boldsymbol{\ell}_s^t \right) \right|,
$$

$$
\leq \frac{2Hd^2 N_{\mathrm{reg}} \varepsilon_{\mathrm{freed}}}{\nu} + d N_{\mathrm{reg}} \varepsilon_{\mathrm{rep}}. \tag{83}
$$

## F.4 Putting It All Together

By combining (74), (79), (83), and (75), we have that under the event $\mathcal{E}$ in (72):

$$
\mathrm{Reg}_T = HT_0 + \sum_{k \in [K]} \sum_{t \in \mathcal{I}^{(k)}} \left( \mathbb{E}_{\boldsymbol{\pi} \sim \rho^{(k)}} \mathbb{E} \left[ V_1^{\boldsymbol{\pi}}(x_1; \boldsymbol{\mathcal{H}}^{t-1}) \right] - V_1^{\pi^*}(x_1; \boldsymbol{\mathcal{H}}^{t-1}) \right),
$$

$$
\leq HT_0 + HT\nu + T \cdot \varepsilon_{\mathrm{span}} + 3dT \cdot \varepsilon_{\mathrm{rep}} + \frac{2Hd^2 T \varepsilon_{\mathrm{freed}}}{\nu}
$$

$$
+ dT\varepsilon_{\mathrm{rep}} + \frac{N_{\mathrm{reg}} \log(T/\gamma)}{\eta} + 64H^2 d^4 \eta T + H\gamma. \tag{84}
$$

Thus, plugging in the expression of $T_0$, $\varepsilon_{\mathrm{freed}}$, and $(\varepsilon_{\mathrm{span}}, \varepsilon_{\mathrm{rep}})$ from Algorithm 4, Lemma F.3, and Corollary F.1, respectively, and ignoring polynormal factors $d, A, H, \log(|\Phi| \varepsilon^{-1} \delta^{-1})$, we get that

$$
\mathrm{Reg}_T \prec \frac{1}{\varepsilon^2} + T\varepsilon + T\nu + \frac{T}{\nu} \sqrt{\frac{\nu}{N_{\mathrm{reg}}}} + \frac{N_{\mathrm{reg}}}{\eta} + \eta T,
$$

$$
\prec T^{2/3} + T N_{\mathrm{reg}}^{-1/3} + \sqrt{T N_{\mathrm{reg}}}, \quad (\text{by setting } \varepsilon = T^{-1/3}, \eta = (N_{\mathrm{reg}}/T)^{1/2}, \nu = N_{\mathrm{reg}}^{-1/3}) \tag{85}
$$

$$
\prec T^{4/5} \quad (N_{\mathrm{reg}} = T^{3/5}), \tag{86}
$$

where the last step follows by setting $N_{\mathrm{reg}} = T^{2/3}$.

## F.5 Spanner Guarantee

---

**Algorithm 7** Spanner: Computing an Approximate Spanner.

---

**Require:** Layer $h$, feature classes $\Phi$, policy covers $\Psi_{1:H}$, feature map $\bar{\phi} : \mathcal{X} \times \mathcal{A} \to \mathbb{R}^{2d}$, # of episodes $n$.

1: Define $\mathcal{G} = \{g : (x, a) \mapsto \phi(x, a)^\top w \mid \phi \in \Phi, w \in \mathbb{B}_d(2\sqrt{d})\}$.
2: For $\theta \in \mathbb{R}^{2d}$ and $(x, a) \in \mathcal{X} \times \mathcal{A}$, define

$$
r_t(x, a; \theta) \coloneqq \begin{cases} \bar{\phi}_h(x, a)^\top \theta, & \text{for } t = h, \\ 0, & \text{otherwise.} \end{cases} \tag{87}
$$

3: Set $\mathcal{G}_h = \{(x, a) \mapsto \bar{\phi}_h(x, a)^\top \theta : \theta \in \mathbb{B}_{2d}(1)\}$, and for $t \in [h - 1]$, set $\mathcal{G}_t = \mathcal{G}$.
4: For each $t \in [h]$, set $P_t = \mathrm{unif}(\Psi_t)$.
5: For $\theta \in \mathbb{R}^{2d}$, define $\mathrm{LinOpt}(\theta) = \mathrm{PSDP}(h, r_{1:h}(\cdot, \cdot; \theta), \mathcal{G}_{1:h}, P_{1:h}, n) \in \Pi$.   // PSDP as in Mhammedi et al. (2023).
6: For $\theta \in \mathbb{R}^{2d}$ and $\pi \in \Pi$, define $\mathrm{LinEst}(\pi) = \mathrm{EstVec}(h, \bar{\phi}_h, \pi, n)$. // EstVec as in Mhammedi et al. (2023).
7: Set $\pi_{1:2d} = \mathrm{RobustSpanner}\left(\mathrm{LinOpt}(\cdot), \mathrm{LinEst}(\cdot), 2, \sqrt{\frac{Ad^2 \log(ndH|\Phi|/\delta)}{\alpha n}}\right)$.    // RobustSpanner as in Mhammedi et al. (2023).
8: **Return:** Policy cover $\{\pi_1, \dots, \pi_{2d}\}$.

---

**Lemma F.2** (Spanner Guarantee). *Let $\varepsilon, \alpha, \delta \in (0, 1)$, $h \in [H]$, $n \geq 1$, and $\bar{\phi}_h : \mathcal{X} \times \mathcal{A} \to \mathbb{R}^{2d}$ be given. Suppose that Assumption 2.1 and Assumption 4.1 hold, and let $\Psi_{1:h}$ be such that for all $s \in [h]$, $\Psi_s$ is an $(\alpha, \varepsilon)$-policy cover for layer $s$ with $|\Psi_s| = 2d$. Then, the output $\Psi_h^{\mathrm{span}} =$*

Spanner$(h, \Phi, \Psi_{1:h}, \bar{\phi}_h, n)$ (Algorithm 7) is such that $|\Psi_h^{\mathrm{span}}| = 2d$ and, with probability at least $1 - \delta$, for all $\pi' \in \Pi$, there exist $\{\beta_\pi \in [-2, 2] : \pi \in \Psi_h^{\mathrm{span}}\}$ such that

$$\left\| \mathbb{E}^{\pi'}[\bar{\phi}_h(\boldsymbol{x}_h, \boldsymbol{a}_h)] - \sum_{\pi \in \Psi_h^{\mathrm{span}}} \beta_\pi \cdot \mathbb{E}^\pi[\bar{\phi}_h(\boldsymbol{x}_h, \boldsymbol{a}_h)] \right\| \leq \varepsilon_{\mathrm{span}}(n, \alpha, \delta), \tag{88}$$

where

$$\varepsilon_{\mathrm{span}}(n, \alpha, \delta) \coloneqq cH^2 d \sqrt{\frac{dA \cdot (d\log(2n\sqrt{d}H) + \log(n|\Phi|/\delta))}{\alpha n}} + H^2 d^{5/2} \varepsilon. \tag{89}$$

where $c > 0$ is a large enough absolute constant. Furthermore, the number of episodes $T_{\mathrm{span}}(n)$ used by the call to Spanner is at most $\widetilde{O}(H^2 d^2 n)$.

**Proof.** To derive the desired bound, we will use the generic guarantee of RobustSpanner from (Mhammedi et al., 2023, Proposition E.1). To invoke this result, we first need to derive guarantees for the optimization and estimation subroutines LinOpt and LinEst withn the Spanner algorithm (Algorithm 7). In particular, we need to show that there is some $\varepsilon' \in (0, 1)$ such that (with high probability) for any $\bar{\theta} \in \mathbb{R}^{2d} \setminus \{0\}$ and $\pi \in \Pi$, the outputs $\hat{\pi}_{\bar{\theta}} \coloneqq \mathrm{LinOpt}(\bar{\theta}/\|\bar{\theta}\|)$ and $\hat{\phi}^\pi \coloneqq \mathrm{LinEst}(\pi)$ satisfy

$$\sup_{\pi \in \Pi} \bar{\theta}^\top \mathbb{E}^\pi[\bar{\phi}_h(\boldsymbol{x}_h, \boldsymbol{a}_h)] \leq \bar{\theta}^\top \mathbb{E}^{\hat{\pi}_{\bar{\theta}}}[\bar{\phi}_h(\boldsymbol{x}_h, \boldsymbol{a}_h)] + \varepsilon' \cdot \|\bar{\theta}\| \quad \text{and} \quad \|\hat{\phi}^\pi - \mathbb{E}^\pi[\bar{\phi}_h(\boldsymbol{x}_h, \boldsymbol{a}_h)]\| \leq \varepsilon'. \tag{90}$$

With this, we can apply (Mhammedi et al., 2023, Proposition E.1) to get that the output

$$\pi_{1:2d} = \mathrm{RobustSpanner}(\mathrm{LinOpt}(\cdot), \mathrm{LinEst}(\cdot), 2, \varepsilon)$$

for $\varepsilon \leq 2\varepsilon'$ is such that for all $\pi \in \Pi$, there exist $\beta_1, \ldots, \beta_d \in [-2, 2]$ satisfying

$$\left\| \mathbb{E}^\pi[\bar{\phi}_h(\boldsymbol{x}_h, \boldsymbol{a}_h)] - \sum_{i=1}^d \beta_i \cdot \mathbb{E}^{\pi_i}[\bar{\phi}_h(\boldsymbol{x}_h, \boldsymbol{a}_h)] \right\| \leq 6d\varepsilon'. \tag{91}$$

Since LinOpt is based on PSDP as in Line 5 of Algorithm 7 and $\Psi_1, \ldots, \Psi_h$ are $(\alpha, \varepsilon)$-policy covers for layers 1 to $h$, respectively, (Mhammedi et al., 2023, Corollary H.1) implies that there is an event $\mathcal{E}^{\mathrm{PSDP}}$ of probability at least $1 - \delta/2$ under which for any $\bar{\theta} \in \mathbb{R}^d \setminus \{0\}$, the output $\hat{\pi}_{\bar{\theta}} = \mathrm{LinOpt}(\bar{\theta})$ satisfies

$$\sup_{\pi \in \Pi} \bar{\theta}^\top \mathbb{E}^\pi[\bar{\phi}_h(\boldsymbol{x}_h, \boldsymbol{a}_h)] \leq \bar{\theta}^\top \mathbb{E}^{\hat{\pi}_{\bar{\theta}}}[\bar{\phi}_h(\boldsymbol{x}_h, \boldsymbol{a}_h)]$$

$$+ \|\bar{\theta}\| \cdot \left( cH^2 \sqrt{\frac{dA \cdot (d\log(2n\sqrt{d}H) + \log(n|\Phi|/\delta))}{\alpha n}} + H^2 d^{3/2} \varepsilon \right),$$

for a large enough absolute constant $c > 0$. On the other hand, since LinEst is based on EstVec as in Line 6, (Mhammedi et al., 2023, Lemma G.3) implies that there is an event $\mathcal{E}^{\mathrm{EstVec}}$ of probability at least $1 - \delta/2$ under which for all $\pi \in \Pi$, the output $\hat{\phi}^\pi \coloneqq \mathrm{LinEst}(\pi)$ satisfies

$$\|\hat{\phi}^\pi - \mathbb{E}^\pi[\bar{\phi}_h(\boldsymbol{x}_h, \boldsymbol{a}_h)]\| \leq c \cdot \sqrt{\frac{\log(2/\delta)}{n}},$$

for a large enough absolute constant $c > 0$. Therefore, under $\mathcal{E}^{\mathrm{PSDP}} \cap \mathcal{E}^{\mathrm{EstVec}}$, LinOpt and LinEst satisfy (90) with

$$\varepsilon' \coloneqq cH^2 \sqrt{\frac{dA \cdot (d\log(2n\sqrt{d}H) + \log(n|\Phi|/\delta))}{\alpha n}} + H^2 d^{3/2} \varepsilon.$$

Therefore, by (Mhammedi et al., 2023, Proposition) and the fact that $d\sqrt{\frac{A\log(ndH|\Phi|/\delta)}{\alpha n}} \leq \varepsilon'$, the output

$$\pi_{1:2d} = \mathrm{RobustSpanner}\left( \mathrm{LinOpt}(\cdot), \mathrm{LinEst}(\cdot), 2, d\sqrt{\frac{A\log(ndH|\Phi|/\delta)}{\alpha n}} \right)$$

is such that for all $\pi \in \Pi$, there exist $\beta_1, \ldots, \beta_{2d} \in [-2, 2]$ satisfying

$$\left\| \mathbb{E}^\pi[\bar{\phi}_h(\boldsymbol{x}_h, \boldsymbol{a}_h)] - \sum_{i=1}^{2d} \beta_i \cdot \mathbb{E}^{\pi_i}[\bar{\phi}_h(\boldsymbol{x}_h, \boldsymbol{a}_h)] \right\| \leq \varepsilon_{\mathrm{span}}(n, \alpha, \delta),$$

where $\varepsilon_{\mathrm{span}}(n, \alpha, \delta)$ is as in (89).

**Bounding the number of episodes** By (Mhammedi et al., 2023, Proposition E.1), `RobustSpanner` calls `LinOpt` and `LinEst` as most $\widetilde{O}(d^2)$ times. Each call to `LinOpt` [resp. `LinEst`] requires $H^2 n$ episodes. This implies the desired bound on the number of iterations. $\qquad\square$

## F.6 Representation + Spanner

**Corollary F.1.** *Let* $\varepsilon, \delta \in (0,1)$, $\phi_{1:H}^{\mathrm{rep}}$, *and* $\Psi_h^{\mathrm{span}}$ *be as in Algorithm 4. Then, for all* $h \in [H]$, $\phi_h^{\mathrm{rep}} \in \Phi$ *and* $|\Psi_h^{\mathrm{span}}| = 2d$ *and there is an event* $\mathcal{E}^{\mathrm{rep+span}}$ *of probability* $1 - 3\delta/4$ *under which for all* $h \in [H]$:

- *For* $f \in \mathcal{F}_{h+1}$, *with* $\mathcal{F}_{h+1}$ *as in* (97), *there exists* $w_{h+1}^f \in \mathbb{B}_d(3d^{3/2})$ *such that:*

$$\forall \pi \in \Pi, \quad \left| \mathbb{E}^\pi \left[ \phi_h^{\mathrm{rep}}(\boldsymbol{x}_h, \boldsymbol{a}_h)^\top w_{h+1}^f - \mathbb{E}[f(\boldsymbol{x}_{h+1}) \mid \boldsymbol{x}_h, \boldsymbol{a}_h] \right] \right| \leq \varepsilon_{\mathrm{rep}} \coloneqq 10 d^{7/2} \varepsilon; \quad (92)$$

- *For all* $\pi' \in \Pi$, *there exist* $\{\beta_\pi \in [-2,2] : \pi \in \Psi_h^{\mathrm{span}}\}$ *such that for* $\bar{\phi}_h^{\mathrm{rep}} \coloneqq [\phi_h^{\mathrm{loss}}, \phi_h^{\mathrm{rep}}]$

$$\left\| \mathbb{E}^{\pi'} [\bar{\phi}_h^{\mathrm{rep}}(\boldsymbol{x}_h, \boldsymbol{a}_h)] - \sum_{\pi \in \Psi_h^{\mathrm{span}}} \beta_\pi \cdot \mathbb{E}^\pi [\bar{\phi}_h^{\mathrm{rep}}(\boldsymbol{x}_h, \boldsymbol{a}_h)] \right\| \leq \varepsilon_{\mathrm{span}} \coloneqq 2H^2 d^{5/2} \varepsilon. \quad (93)$$

**Proof.** From Algorithm 4, we have

$$\phi_h^{\mathrm{rep}} = \mathtt{RepLearn}(h, \mathcal{F}_{h+1}, \Phi, \mathtt{unif}(\Psi_h^{\mathrm{cov}}), T_{\mathrm{rep}}) \quad \text{and} \quad \Psi_h^{\mathrm{span}} = \mathtt{Spanner}(h, \Phi, \Psi_{1:h}^{\mathrm{cov}}, \bar{\phi}_h^{\mathrm{rep}}, T_{\mathrm{span}}),$$

where

$$\Psi_{1:H}^{\mathrm{cov}} = \mathtt{VoX}(\Phi, \varepsilon, \delta/4), \qquad T_{\mathrm{rep}} \coloneqq \frac{AH \log(|\Phi|/\delta)}{\alpha \varepsilon^2}, \qquad T_{\mathrm{span}} = \frac{A \log(dH|\Phi|\varepsilon^{-1}\delta^{-1})}{\alpha \varepsilon^2}, \quad (94)$$

and $\alpha \coloneqq \frac{1}{8Ad}$. By Lemma G.1, there is an event $\mathcal{E}^{\mathrm{cov}}$ of probability at least $1 - \delta/4$ under which, for all $h \in [H]$, $\Psi_h^{\mathrm{cov}}$ is an $(\alpha, \varepsilon)$-policy cover for layer $h$ with $|\Psi_h^{\mathrm{cov}}| = d$. In what follows, we condition on $\mathcal{E}^{\mathrm{cov}}$. By Lemma G.2 and Lemma F.2, there are events $\mathcal{E}^{\mathrm{rep}}$ and $\mathcal{E}^{\mathrm{span}}$ of probability at least $1 - \delta/4$ each such that under $\mathcal{E}^{\mathrm{rep}} \cap \mathcal{E}^{\mathrm{span}}$ (92) and (93) hold; this follows from (98) and (88) and the choices of $T_{\mathrm{rep}}$ and $T_{\mathrm{span}}$ in (94). Finally, by the union bound, we have $\mathbb{P}[\mathcal{E}^{\mathrm{cov}} \cap \mathcal{E}^{\mathrm{rep}} \cap \mathcal{E}^{\mathrm{span}}] \geq 1 - \delta$ which completes the proof. $\qquad\square$

## F.7 Martingal Concentration

**Lemma F.3.** *Let* $K$, $N_{\mathrm{reg}}$, $\bar{\phi}_{1:H}^{\mathrm{rep}}$, *and* $\mathcal{I}^{(k)}$ *be as in Algorithm 4 for* $k \in [K]$. *There is an event* $\mathcal{E}^{\mathrm{freed}}$ *of probability at least* $1 - \delta/4$ *under which for all* $\bar{\theta} \in \mathbb{B}_{2d}(4Hd^2)$, $h \in [H]$, $k \in [K]$, *and* $\pi \in \Psi_h^{\mathrm{span}}$:

$$\frac{1}{N_{\mathrm{reg}}} \left| \sum_{t \in \mathcal{I}^{(k)}} \mathbb{I}\{\boldsymbol{h}^t = h, \boldsymbol{\pi}^t = \pi, \boldsymbol{\zeta}^t = 1\} \cdot \left( \bar{\phi}_h^{\mathrm{rep}}(\boldsymbol{x}_h^t, \boldsymbol{a}_h^t)^\top \bar{\theta} - \sum_{s=h}^H \boldsymbol{\ell}_s^t \right) \right.$$

$$\left. - \sum_{t \in \mathcal{I}^{(k)}} \mathbb{E} \left[ \mathbb{I}\{\boldsymbol{h}^t = h, \boldsymbol{\pi}^t = \pi, \boldsymbol{\zeta}^t = 1\} \cdot \left( \bar{\phi}_h^{\mathrm{rep}}(\boldsymbol{x}_h^t, \boldsymbol{a}_h^t)^\top \bar{\theta} - \sum_{s=h}^H \boldsymbol{\ell}_s^t \right) \mid \mathcal{H}^{t-1} \right] \right| \leq \varepsilon_{\mathrm{freed}} \coloneqq 4Hd^2 \sqrt{\frac{\nu \log(dKHN_{\mathrm{reg}}/\delta)}{N_{\mathrm{reg}}}},$$

*where the random variables* $\boldsymbol{h}^t, \boldsymbol{\zeta}^t, \boldsymbol{\pi}^t$, *and* $\mathcal{H}^{t-1}$ *are as in Algorithm 7.*

**Proof.** Fix $\bar{\theta} \in \mathbb{B}_{2d}(4d^2)$, $h \in [H]$, $k \in [K]$, and $\pi \in \Psi_h^{\mathrm{span}}$. We apply Lemma I.2 (Freedman's inequality) with

- $R = 4Hd^2$;

- $n = N_{\mathrm{reg}}$;

- The random variable $\boldsymbol{w}^i$ set as the difference

$$\boldsymbol{w}^i \coloneqq \mathbb{I}\{\boldsymbol{h}^{t_i} = h, \boldsymbol{\pi}^{t_i} = \pi, \boldsymbol{\zeta}^{t_i} = 1\} \cdot \left( \bar{\phi}_h^{\mathrm{rep}}(\boldsymbol{x}_h^{t_i}, \boldsymbol{a}_h^{t_i})^\top \bar{\theta} - \sum_{s=h}^H \boldsymbol{\ell}_s^{t_i} \right)$$

$$- \mathbb{E} \left[ \mathbb{I}\{\boldsymbol{h}^{t_i} = h, \boldsymbol{\pi}^{t_i} = \pi, \boldsymbol{\zeta}^{t_i} = 1\} \cdot \left( \bar{\phi}_h^{\mathrm{rep}}(\boldsymbol{x}_h^{t_i}, \boldsymbol{a}_h^{t_i})^\top \bar{\theta} - \sum_{s=h}^H \boldsymbol{\ell}_s^{t_i} \right) \mid \mathcal{H}^{t-1} \right], \quad (95)$$

where $t_i \coloneqq (k-1) \cdot N_{\mathrm{reg}} + i$;

- The filtration $\mathfrak{F}^i$ set as the $\sigma$-algebra $\sigma(\mathcal{H}^{t_i-1})$;

- The variance term $V_n$ has the following upper bound

$$V_n = \sum_{i=1}^{N_{\mathrm{reg}}} \mathbb{E}\left[ (\boldsymbol{w}^i)^2 \mid \mathfrak{F}^{i-1} \right] \le \sum_{i=1}^{N_{\mathrm{reg}}} \mathbb{E}\left[ \mathbb{I}\{\boldsymbol{h}^{t_i} = h, \boldsymbol{\pi}^{t_i} = \pi, \boldsymbol{\zeta}^{t_i} = 1\} \cdot \left( \bar{\phi}_h^{\mathrm{rep}}(\boldsymbol{x}_h^{t_i}, \boldsymbol{a}_h^{t_i})^\top \bar{\theta} - \sum_{s=h}^{H} \boldsymbol{\ell}_s^{t_i} \right)^2 \mid \mathfrak{F}^{i-1} \right]$$

$$\le 8 H d^3 \nu N_{\mathrm{reg}};$$

- $\lambda = H^{-1} \left( \frac{d^2 \nu N_{\mathrm{reg}}}{\log(K H N_{\mathrm{reg}}/\delta)} \right)^{-1/2}$;

to get that there is an event $\mathcal{E}_{h,k,\pi}^{\mathrm{freed}}(\bar{\theta})$ of probability at last $1 - (N_{\mathrm{reg}})^{-d} H^{-1} K^{-1} d^{-1} \delta/8$ under which

$$\frac{1}{N_{\mathrm{reg}}} \left| \sum_{t \in \mathcal{I}^{(k)}} \mathbb{I}\{\boldsymbol{h}^t = h, \boldsymbol{\pi}^t = \pi, \boldsymbol{\zeta}^t = 1\} \cdot \left( \bar{\phi}_h^{\mathrm{rep}}(\boldsymbol{x}_h^t, \boldsymbol{a}_h^t)^\top \bar{\theta} - \sum_{s=h}^{H} \boldsymbol{\ell}_s^t \right) \right.$$

$$\left. - \sum_{t \in \mathcal{I}^{(k)}} \mathbb{E}\left[ \mathbb{I}\{\boldsymbol{h}^t = h, \boldsymbol{\pi}^t = \pi, \boldsymbol{\zeta}^t = 1\} \cdot \left( \bar{\phi}_h^{\mathrm{rep}}(\boldsymbol{x}_h^t, \boldsymbol{a}_h^t)^\top \bar{\theta} - \sum_{s=h}^{H} \boldsymbol{\ell}_s^t \right) \mid \mathcal{H}^{t-1} \right] \right|$$

$$\le 4 H d^2 \sqrt{\frac{\nu \log(d K H N_{\mathrm{reg}}/\delta)}{N_{\mathrm{reg}}}}. \tag{96}$$

Let $\mathcal{C}$ be a minimal $(d H N_{\mathrm{reg}})^{-1}$-cover of $\mathbb{B}_{2d}(4 H d^2)$ with respect to the $\|\cdot\|$ distance. Under the event

$$\mathcal{E}^{\mathrm{freed}} \coloneqq \bigcap_{h \in [H], k \in [K], \pi \in \Psi_h^{\mathrm{span}}, \bar{\theta} \in \mathbb{B}_{2d}(4 H d^2)} \mathcal{E}_{h,k,\pi}^{\mathrm{freed}}(\bar{\theta}),$$

Eq. (96) holds for all $h \in [H], k \in [K]$, and $\bar{\theta} \in \mathbb{B}_{2d}(4 H d^2)$ up to an additive $O(1/N_{\mathrm{reg}})$ error. By the union bound, we have $\mathbb{P}[\mathcal{E}^{\mathrm{freed}}] \ge 1 - \delta/4$ which completes the proof. $\qquad\square$

# G Policy Cover and Representation Learning Algorithms

In this section, we present guarantees for VoX, RepLearn, and RobustSpanner which we need in the analysis of our oracle efficient algorithm. The results are based on (Mhammedi et al., 2023).

## G.1 Policy Cover

The following result is a restatement of (Mhammedi et al., 2023, Theorem 12).

**Lemma G.1** (VoX Guarantee). *Let $\varepsilon, \delta \in (0, 1)$ be given. Suppose Assumption 2.1 and Assumption 4.1 hold. Then, there is an event $\mathcal{E}^{\mathrm{cov}}$ of probability at least $1 - \delta$ under which the output $\Psi_{1:H}^{\mathrm{cov}} = \mathrm{VoX}(\Phi, \varepsilon, \delta)$ is such that for all $h \in [H]$:*

- *$\Psi_h^{\mathrm{cov}}$ is a $\left(\frac{1}{8Ad}, \varepsilon\right)$-policy cover for layer $h$;*

- *$|\Psi_h^{\mathrm{cov}}| \leq d$.*

*Furthermore, the number of episodes $T_{\mathrm{cov}}(\varepsilon)$ used by the call to $\mathrm{VoX}$ is bounded by $\widetilde{O}(Ad^{13}H^6 \log(\Phi/\delta))/\varepsilon^2$.*

## G.2 Representation Learning

**Lemma G.2** (Representation Learning Guarantee). *Let $\varepsilon, \alpha, \delta \in (0, 1)$, $h \in [H - 1]$, and $n \geq 1$ be given and define the function class*

$$\mathcal{F}_{h+1} \coloneqq \left\{ f : (x, a) \mapsto \max_{a \in \mathcal{A}} \bar{\phi}_{h+1}(x, a)^\top \bar{\theta} \mid \bar{\phi}_{h+1} = [\phi_{h+1}^{\mathrm{loss}}, \phi_{h+1}], \phi \in \Phi, \bar{\theta} \in \mathbb{B}_{2d}(1) \right\}. \tag{97}$$

*Further, let $\Psi$ be an $(\alpha, \varepsilon)$-policy cover for layer $h$ with $|\Psi| = d$, and suppose Assumption 2.1 and Assumption 4.1 hold. Then, with probability at least $1 - \delta$, the output $\bar{\phi}_h^{\mathrm{rep}} = \mathrm{RepLearn}(h, \mathcal{F}_{h+1}, \Phi, \mathrm{unif}(\Psi), n)$ is such that for all $f \in \mathcal{F}_{h+1}$ there exists $w_{h+1}^f \in \mathbb{B}_d(3d^{3/2})$ such that:*

$$\forall \pi \in \Pi, \quad \left| \mathbb{E}^\pi \left[ \phi_h^{\mathrm{rep}}(\boldsymbol{x}_h, \boldsymbol{a}_h)^\top w_{h+1}^f - \mathbb{E}[f(\boldsymbol{x}_{h+1}) \mid \boldsymbol{x}_h, \boldsymbol{a}_h] \right] \right| \leq c \cdot \sqrt{\frac{AHd^5 \log(|\Phi|/\delta)}{\alpha n}} + 9d^{7/2}\varepsilon, \tag{98}$$

*where $c > 0$ is a large enough absolute constant. Furthermore, the number of episodes $T_{\mathrm{rep}}(\varepsilon)$ used by the call to $\mathrm{RepLearn}$ is equal to $n$.*

**Proof.** By (Mhammedi et al., 2023, Theorem F.1) and the assumption that $|\Psi| = d$, there is an event $\mathcal{E}$ of probability at least $1 - \delta/2$ under which $\phi_h^{\mathrm{rep}}$ satisfies:

$$\sup_{f \in \mathcal{F}_{h+1}} \inf_{w \in \mathbb{B}_d(3d^{3/2})} \max_{\pi' \in \Psi} \mathbb{E}^{\pi'} \left[ (\phi_h^{\mathrm{rep}}(\boldsymbol{x}_h, \boldsymbol{a}_h)^\top w - \mathbb{E}[f(\boldsymbol{x}_{h+1}) \mid \boldsymbol{x}_h, \boldsymbol{a}_h])^2 \right] \leq c \cdot \frac{Ad^5 \log(|\Phi|/\delta)}{n}, \tag{99}$$

where $c$ is a large enough absolute constant. We use this to show (98). In what follows, we condition on $\mathcal{E}$. Fix $\pi \in \Pi$ and $f \in \mathcal{G}$ and let $w_{h+1}^f$ be the vector $w \in \mathbb{B}_d(3d^{3/2})$ achieving the infimum in (99) for the given choice of $f$. Let $\mathcal{X}_{h,\varepsilon}$ be the set of $\varepsilon$-reachable states at layer $h$ as defined in (50). With this this, we have for all $h \in [H]$,

$$\left| \mathbb{E}^\pi \left[ \phi_h^{\mathrm{rep}}(\boldsymbol{x}_h, \boldsymbol{a}_h)^\top w_{h+1}^f - \mathbb{E}[f(\boldsymbol{x}_{h+1}) \mid \boldsymbol{x}_h, \boldsymbol{a}_h] \right] \right|$$

$$\leq \left| \mathbb{E}^\pi \left[ \mathbb{I}\{\boldsymbol{x}_h \notin \mathcal{X}_{h,\varepsilon}\} \cdot (\phi_h^{\mathrm{rep}}(\boldsymbol{x}_h, \boldsymbol{a}_h)^\top w_{h+1}^f - \mathbb{E}[f(\boldsymbol{x}_{h+1}) \mid \boldsymbol{x}_h, \boldsymbol{a}_h]) \right] \right|$$

$$+ \left| \mathbb{E}^\pi \left[ \mathbb{I}\{\boldsymbol{x}_h \in \mathcal{X}_{h,\varepsilon}\} \cdot (\phi_h^{\mathrm{rep}}(\boldsymbol{x}_h, \boldsymbol{a}_h)^\top w_{h+1}^f - \mathbb{E}[f(\boldsymbol{x}_{h+1}) \mid \boldsymbol{x}_h, \boldsymbol{a}_h])^2 \right] \right|,$$

$$\leq \left| \mathbb{E}^\pi \left[ \mathbb{I}\{\boldsymbol{x}_h \in \mathcal{X}_{h,\varepsilon}\} \cdot (\phi_h^{\mathrm{rep}}(\boldsymbol{x}_h, \boldsymbol{a}_h)^\top w_{h+1}^f - \mathbb{E}[f(\boldsymbol{x}_{h+1}) \mid \boldsymbol{x}_h, \boldsymbol{a}_h]) \right] \right| + 9d^{7/2}\varepsilon, \tag{100}$$

where the last inequality follows by Lemma I.1.

We now bound the first term on the right-hand side of (100). By Jensen's inequality, we have

$$\left| \mathbb{E}^\pi \left[ \mathbb{I}\{\boldsymbol{x}_h \in \mathcal{X}_{h,\varepsilon}\} \cdot (\phi_h^{\mathrm{rep}}(\boldsymbol{x}_h, \boldsymbol{a}_h)^\top w_{h+1}^f - \mathbb{E}[f(\boldsymbol{x}_{h+1}) \mid \boldsymbol{x}_h, \boldsymbol{a}_h]) \right] \right|$$

$$\leq \sqrt{\mathbb{E}^\pi\left[\mathbb{I}\{\boldsymbol{x}_h \in \mathcal{X}_{h,\varepsilon}\} \cdot (\phi_h^{\mathrm{rep}}(\boldsymbol{x}_h, \boldsymbol{a}_h)^\top w_{h+1}^f - \mathbb{E}[f(\boldsymbol{x}_{h+1}) \mid \boldsymbol{x}_h, \boldsymbol{a}_h])^2\right]},$$

and so using that $\Psi$ is a $(\alpha, \varepsilon')$-policy cover (see Definition 4.1), we have

$$\leq \sqrt{\alpha^{-1} \cdot \max_{\pi' \in \Psi} \mathbb{E}^{\pi'}\left[(\phi_h^{\mathrm{rep}}(\boldsymbol{x}_h, \boldsymbol{a}_h)^\top w_{h+1}^f - \mathbb{E}[f(\boldsymbol{x}_{h+1}) \mid \boldsymbol{x}_h, \boldsymbol{a}_h])^2\right]},$$

$$\leq \sqrt{c \cdot \frac{AHd^5 \log(|\Phi|/\delta)}{\alpha n}},$$

where the last step follows by (99). Combining this with (100) yields (98). $\qquad\square$

# H  Lower Bound for Bandit feedback with Unstructured Losses

In full-information, one does not require any structure on the losses. We show that this is not the case for the bandit case via a lower bound depending polynomially on the number of states. This lower bound implies that the low-rank transition structure with unstructured losses does not give any significant improvements over the tabular setting.

**Theorem H.1.** *There exists a low-rank MDP with $S$ states, $A$ actions and sufficiently large time step $T$ with unstructured losses such that any agent suffers at least regret of $\Omega(\sqrt{SAT})$.*

**Proof.** We assume $4S < \sqrt{T}$. The construction is an $H = 1$ (i.e. contextual bandit) MDP with uniform initial distribution over states. Each state is a copy of an $A$-armed bandit problem with Bernoulli losses with mean $\frac{1}{2}$, and one randomly chosen optimal arm with mean $\frac{1}{2} - \Delta$. Following the standard lower bound construction for bandits (Lattimore and Szepesvári, 2020), there exists $\Delta = \Theta(1/\sqrt{TAS})$ such that the the regret of playing any individual bandit problem for $N \leq 2T/S$ rounds is lower bounded by $\Omega(N\Delta)$. Let denote $N(s)$ the number of time the agent receives the initial state $s$, then any agent suffers a regret lower bound in our MDP of

$$\Omega\left(\sum_{s=1}^{S} \mathbb{E}[\min\{N(s), 2T/S\}]\right).$$

$N(s)$ is the sum of $T$ Bernoulli random variables with mean $1/S$. We have by Hoeffding's inequality

$$\mathbb{P}[N(s) > T/S + x] \leq \exp(-2x^2/T).$$

This allows to upper bound the tail

$$
\begin{aligned}
\mathbb{E}[N(s)\mathbb{I}(N(s) > 2T/S)] &\leq \int_{T/S}^{\infty} x(2x/T \exp(-2x^2/T))\, dx \\
&\leq \int_{T/S}^{\infty} 4 \exp(x/\sqrt{T} - 2x^2/T)\, dx \qquad (\tfrac{1}{2}(x/\sqrt{T})^2 < \exp(x/\sqrt{T})) \\
&\leq \int_{T/S}^{\infty} 4 \exp(-x/\sqrt{T})\, dx \qquad\qquad (x \geq T/S > 4\sqrt{T}) \\
&= 4\sqrt{T} \exp(-\sqrt{T}/S) \leq 4\sqrt{T} \exp(-4) \leq T/(2S).
\end{aligned}
$$

Hence $\mathbb{E}[\min\{N(s), 2T/S\}] = T/S - \mathbb{E}[N(s)\mathbb{I}(N(s) > 2T/S)] \geq T/(2S)$ and the regret in the MDP is lower bounded by $\Omega\left(\sqrt{TSA}\right)$. $\qquad\square$

# I  Helper Results

**Lemma I.1.** *Let $\varepsilon, B > 0$ and $h \in [2 .. H]$ be given. For any function $f : \mathcal{X} \to [-B, B]$ and $\pi \in \Pi$, we have*

$$\mathbb{E}^{\pi}[\mathbb{I}\{\boldsymbol{x}_h \notin \mathcal{X}_{h,\varepsilon}\} \cdot f(\boldsymbol{x}_h)] \leq B\sqrt{d}\varepsilon, \tag{101}$$

*where*

$$\mathcal{X}_{h,\varepsilon} := \left\{x \in \mathcal{X} : \max_{\pi \in \Pi} d_h^{\pi}(x) \geq \varepsilon \cdot \|\mu_h^{\star}(x)\|\right\}, \tag{102}$$

*denotes the set of states that are $\varepsilon$-reachable at layer $h$.*

**Proof.** Fix $f : \mathcal{X} \to [-B, B]$ and $\pi \in \Pi$. Using the definition of $\mathcal{X}_{h,\varepsilon}$ in (102), we have that $x \notin \mathcal{X}_{h,\varepsilon}$ only if $d_h^{\pi}(x) < \varepsilon\|\mu_h^{\star}(x)\|$. Using this, we have

$$
\begin{aligned}
\mathbb{E}^{\pi}[\mathbb{I}\{\boldsymbol{x}_h \notin \mathcal{X}_{h,\varepsilon}\} \cdot f(\boldsymbol{x}_h)] &\leq \sum_{x \in \mathcal{X}} \mathbb{I}\{x \notin \mathcal{X}_{h,\varepsilon}\} \cdot d_h^{\pi}(x) \cdot f(x), \\
&\leq B\varepsilon \sum_{x \in \mathcal{X}} \|\mu_h^{\star}(x)\|, \\
&\leq Bd^{3/2}\varepsilon, \tag{103}
\end{aligned}
$$

where the last step follows by the normalizing assumption on $\mu^{\star}$ (see Assumption 2.1) and (Mhammedi et al., 2023, Lemma I.3). $\qquad\square$

### I.1 Martingale Concentration and Regression Results

**Lemma I.2.** *Let $R > 0$ be given and let $\boldsymbol{w}^1, \dots \boldsymbol{w}^n$ be a sequence of real-valued random variables adapted to filtration $\mathfrak{F}^1, \cdots, \mathfrak{F}^n$. Assume that for all $i \in [n]$, $\boldsymbol{w}^i \le R$ and $\mathbb{E}[\boldsymbol{w}^i \mid \mathfrak{F}^{i-1}] = 0$. Define $\boldsymbol{S}_n := \sum_{i=1}^n \boldsymbol{w}^i$ and $V_n := \sum_{i=1}^n \mathbb{E}[(\boldsymbol{w}^i)^2 \mid \mathfrak{F}^{i-1}]$. Then, for any $\delta \in (0,1)$ and $\lambda \in [0, 1/R]$, with probability at least $1 - \delta$,*

$$\boldsymbol{S}_n \le \lambda V_n + \ln(1/\delta)/\lambda. \tag{104}$$

We now state two helpful results from Mhammedi et al. (2024b) without a proof.

**Lemma I.3.** *Let $B > 0$ and $n \in \mathbb{N}$ be given. abstract set. Further, let $\mathcal{Q} \subseteq \{g : \mathcal{X} \times \mathcal{A} \to [0, B]\}$ be a finite function class and $(\boldsymbol{x}^1, \boldsymbol{a}^1, \boldsymbol{\varepsilon}^1), \dots, (\boldsymbol{x}^n, \boldsymbol{a}^n, \boldsymbol{\varepsilon}^n)$ be a sequence of i.i.d. random variables in $\mathcal{X} \times \mathcal{A} \times \mathbb{R}$. Then, for any $\delta \in (0,1)$, with probability at least $1 - \delta$, we have*

$$\forall g \in \mathcal{Q}, \quad \frac{1}{2}\|g\|^2 - 2B^2 \log(2|\mathcal{Q}|/\delta) \le \|g\|_n^2 \le 2\|g\|^2 + 2B^2 \log(2|\mathcal{Q}|/\delta), \tag{105}$$

*where $\|g\|^2 := \sum_{i \in [n]} \mathbb{E}[g(\boldsymbol{x}^i, \boldsymbol{a}^i)^2 \mid \mathfrak{F}^{i-1}]$ and $\|g\|_n^2 := \sum_{i=1}^n g(\boldsymbol{x}^i, \boldsymbol{a}^i)^2$.*

**Lemma I.4** (Generic regression guarantee). *Let $B > 0$, $n \in \mathbb{N}$, and $f_\star : \mathcal{X} \times \mathcal{A} \to [0, B]$ be given. Further, let $\mathcal{F} \subseteq \{f : \mathcal{X} \times \mathcal{A} \to [0, B]\}$ be a finite function class and $(\boldsymbol{x}^1, \boldsymbol{a}^1, \boldsymbol{\varepsilon}^1), \dots, (\boldsymbol{x}^n, \boldsymbol{a}^n, \boldsymbol{\varepsilon}^n)$ be a sequence of i.i.d. random variables in $\mathcal{X} \times \mathcal{A} \times \mathbb{R}$. Suppose that*

- *$f_\star \in \mathcal{F}$;*
- *$\boldsymbol{z}^i = f_\star(\boldsymbol{x}^i, \boldsymbol{a}^i) + \boldsymbol{\varepsilon}^i + \boldsymbol{b}^i$, for all $i \in [n]$;*
- *$\boldsymbol{b}^1, \dots, \boldsymbol{b}^n \in \mathbb{R}$ (not necessarily i.i.d.);*
- *$\boldsymbol{\varepsilon}^i \in [-B, B]$, for all $i \in [n]$; and*
- *$\mathbb{E}[\boldsymbol{\varepsilon}^i \mid \boldsymbol{x}^i, \boldsymbol{a}^i] = 0$.*

*Then, for $\hat{f} \in \operatorname{argmin}_{f \in \mathcal{F}} \sum_{i=1}^n (f(\boldsymbol{x}^i, \boldsymbol{a}^i) - \boldsymbol{z}^i)^2$ and any $\delta \in (0,1)$, with probability at least $1 - \delta/2$,*

$$\|\hat{f} - f_\star\|_n^2 \le 8B^2 \log(2|\mathcal{F}|/\delta) + 8 \sum_{i=1}^n (\boldsymbol{b}^i)^2, \tag{106}$$

*where $\|\hat{f} - f_\star\|_n^2 := \sum_{i=1}^n (\hat{f}(\boldsymbol{x}^i, \boldsymbol{a}^i) - f^\star(\boldsymbol{x}^i, \boldsymbol{a}^i))^2$.*

**Proof.** Fix $\delta \in (0,1)$ and let $\widehat{L}_n(f) := \sum_{i=1}^n (f(\boldsymbol{x}^i, \boldsymbol{a}^i) - \boldsymbol{z}^i)^2$, for $f \in \mathcal{F}$, and note that since $\hat{f} \in \operatorname{argmin}_{f \in \mathcal{F}} \widehat{L}_n(f)$, we have

$$0 \ge \widehat{L}_n(\hat{f}) - \widehat{L}_n(f_\star) = \nabla \widehat{L}_n(f_\star)[\hat{f} - f_\star] + \|\hat{f} - f_\star\|_n^2, \tag{107}$$

where $\nabla$ denotes directional derivative. Rearranging, we get that

$$\|\hat{f} - f_\star\|_n^2 \le -2\nabla \widehat{L}_n(f_\star)[\hat{f} - f_\star] - \|\hat{f} - f_\star\|_n^2,$$

$$= 4 \sum_{i=1}^n (\boldsymbol{z}^i - f_\star(\boldsymbol{x}^i, \boldsymbol{a}^i))(\hat{f}(\boldsymbol{x}^i, \boldsymbol{a}^i) - f_\star(\boldsymbol{x}^i, \boldsymbol{a}^i)) - \|\hat{f} - f_\star\|_n^2,$$

$$= 4 \sum_{i=1}^n (\boldsymbol{\varepsilon}^i + \boldsymbol{b}^i)(\hat{f}(\boldsymbol{x}^i, \boldsymbol{a}^i) - f_\star(\boldsymbol{x}^i, \boldsymbol{a}^i)) - \|\hat{f} - f_\star\|_n^2,$$

$$= 4 \sum_{i=1}^n \boldsymbol{\varepsilon}^i \cdot (\hat{f}(\boldsymbol{x}^i, \boldsymbol{a}^i) - f_\star(\boldsymbol{x}^i, \boldsymbol{a}^i)) - \|\hat{f} - f_\star\|_n^2 + 4 \sum_{i=1}^n \boldsymbol{b}^i \cdot (\hat{f}(\boldsymbol{x}^i, \boldsymbol{a}^i) - f_\star(\boldsymbol{x}^i, \boldsymbol{a}^i)),$$

$$\tag{108}$$

$$\le 4 \sum_{i=1}^n \boldsymbol{\varepsilon}^i \cdot (\hat{f}(\boldsymbol{x}^i, \boldsymbol{a}^i) - f_\star(\boldsymbol{x}^i, \boldsymbol{a}^i)) - \|\hat{f} - f_\star\|_n^2 + 4 \sum_{i=1}^n (\boldsymbol{b}^i)^2 + \frac{1}{2} \sum_{i=1}^n (\hat{f}(\boldsymbol{x}^i, \boldsymbol{a}^i) - f_\star(\boldsymbol{x}^i, \boldsymbol{a}^i))^2,$$

$$= 4 \sum_{i=1}^n \boldsymbol{\varepsilon}^i \cdot (\hat{f}(\boldsymbol{x}^i, \boldsymbol{a}^i) - f_\star(\boldsymbol{x}^i, \boldsymbol{a}^i)) - \|\hat{f} - f_\star\|_n^2 + 4 \sum_{i=1}^n (\boldsymbol{b}^i)^2 + \frac{1}{2} \|\hat{f} - f_\star\|_n^2. \tag{109}$$

Thus, rearranging, we get

$$\|\hat{f} - f_\star\|_n^2 \le 8 \sum_{i=1}^n \boldsymbol{\varepsilon}^i \cdot (\hat{f}(\boldsymbol{x}^i, \boldsymbol{a}^i) - f_\star(\boldsymbol{x}^i, \boldsymbol{a}^i)) - 2\|\hat{f} - f_\star\|_n^2 + 8 \sum_{i=1}^n (\boldsymbol{b}^i)^2. \tag{110}$$

We now bound the first term on the right-hand side of (110). For this, we apply Lemma I.2 with $\boldsymbol{w}^i = \boldsymbol{\varepsilon}^i \cdot (\hat{f}(\boldsymbol{x}^i, \boldsymbol{a}^i) - f_\star(\boldsymbol{x}^i, \boldsymbol{a}^i))$, $R = B^2$, $\lambda = 1/(8B^2)$, and $\mathfrak{F}^i = \varnothing$, and use

1. the union bound over $f \in \mathcal{F}$; and

2. the facts that $\mathbb{E}[\boldsymbol{\varepsilon}^i \mid \boldsymbol{x}^i, \boldsymbol{a}^i] = 0$,

to get that with probability at least $1 - \delta/2$,

$$\sum_{i=1}^n \boldsymbol{\varepsilon}^i \cdot (\hat{f}(\boldsymbol{x}^i, \boldsymbol{a}^i) - f_\star(\boldsymbol{x}^i, \boldsymbol{a}^i)) \le \frac{1}{4} \|\hat{f} - f_\star\|_n^2 + B^2 \log(2|\mathcal{F}|/\delta). \tag{111}$$

Combining this with (110), we get that with probability at least $1 - \delta/2$,

$$\|\hat{f} - f_\star\|_n^2 \le 8B^2 \log(2|\mathcal{F}|/\delta) + 8 \sum_{i=1}^n (\boldsymbol{b}^i)^2. \tag{112}$$

This completes the proof. $\qquad\square$

## I.2 Online Learning

The following is the standard guarantee of exponential weights (e.g. Lemma F.4 of Sherman et al. (2023b)).

**Lemma I.5** (Exponential Weights). *Given a sequence of loss functions $\{g^t\}_{t=1}^T$ over a decision set $\Pi$, $\{p^t\}_{t=1}^T$ is a distribution sequence with $p^t \in \Delta(\Pi)$, $\forall t \in [T]$ such that*

$$p^{t+1}(\pi) \propto \exp\left(-\eta \sum_{t=1}^T g^t(\pi)\right).$$

*If $p^1$ is a uniform distribution over $|\Pi|$ and $\eta g^t(\pi) \ge -1$ for all $t \in [T]$ and $\pi \in \Pi$. Then*

$$\max_{p \in \Delta(\Pi)} \left\{ \sum_{t=1}^T \langle g^t, p^t - p \rangle \right\} \le \frac{\log(|\Pi|)}{\eta} + \eta \sum_{t=1}^T \sum_{\pi \in \Pi} p^t(\pi) g^t(\pi)^2$$

## I.3 Reinforcement Learning

The following is standard simulation lemma which is first proposed by Abbeel and Ng (2005).

**Lemma I.6** (Simulation Lemma). *For two finite-horizon MDPs $\widehat{M} = \{\mathcal{X}, \mathcal{A}, \ell, \{\widehat{P}_h\}_{h=1}^H\}$ and $M = \{\mathcal{X}, \mathcal{A}, \ell, \{P_h\}_{h=1}^H\}$ with horizon $H$ and $\|\ell\|_\infty \le 1$. Let the corresponding value function be $\widehat{V}_h^\pi(x; \ell)$ and $V_h^\pi(x; \ell)$ for step $h \in [H]$. For any policy $\pi : \mathcal{X} \to \Delta(\mathcal{A})$, we have*

$$\left| \widehat{V}_1^\pi(x_1; \ell) - V_1^\pi(x_1; \ell) \right| \le H \sum_{h=1}^H \mathbb{E}_{x, a \sim d_h^\pi} \left[ \left\| \widehat{P}_h(\cdot \mid x, a) - P_h(\cdot \mid x, a) \right\|_1 \right].$$

The following is the standard performance difference lemma which is first proposed by Kakade and Langford (2002).

**Lemma I.7** (Performance Difference Lemma). *For a finite-horizon MDPs $M = \{\mathcal{X}, \mathcal{A}, \ell, \{P_h\}_{h=1}^H\}$ starting at $x_1$, and two policies $\pi, \pi' : \mathcal{X} \to \Delta(\mathcal{A})$, we have*

$$V_1^{\pi'}(x_1; \ell) - V_1^\pi(x_1; \ell) = \sum_{h=1}^H \mathbb{E}_{x \sim d_h^\pi} \left[ \sum_{a \in \mathcal{A}} (\pi_h'(a|x) - \pi_h(a|x)) Q_h^{\pi'}(x, a; \ell) \right]$$

