# OpenReview forum: "Beating Adversarial Low-Rank MDPs with Unknown Transition and Bandit Feedback"
_NeurIPS.cc/2024/Conference — NeurIPS 2024 poster_

### Official Review · Reviewer_3DuN · 2024-06-17

**Soundness:** 3
**Presentation:** 2
**Contribution:** 3
**Rating:** 6
**Confidence:** 3

**Summary:**

This paper studies online learning in low-rank MDPs with adversarial losses (having linear structure), and derives a set of regret guarantees:

1. A model-based and inefficient algorithm ensures $T^{2/3}$ regret, assuming the loss feature vector is unknown. The algorithm has a two-stage design: the first stage is an initial reward-free exploration stage from [1] to ensure low estimation error for the second (exploitation) stage.

2. A model-free and oracle-efficient algorithm ensures $T^{5/6}$ regret, assuming the loss feature vector is unknown. This is thanks to recent advances in exploration for low-rank MDP by [2].

3. A model-free and oracle-efficient algorithm ensures $T^{5/6}$ regret against adaptive adversaries, assuming the loss feature vector is known, by leveraging a representation learning algorithm from [2].

**Strengths:**

By adopting recent progress in representation learning and low-rank MDP from the literature, the paper provides a set of results for low-rank MDPs with adversarial (linear) losses under bandit feedback. This is the first set of results for the more challenging bandit feedback.

**Weaknesses:**

I overall appreciate the contributions from this work, in which the authors take advantage of SOTA techniques in the literature, but also with some original adaptations, to give a complete picture on regret bounds in a new setup. However, I think the author could do a better job in presentation and delivering insights, which may make readers benefit more from reading this paper. Below are some questions that I think should be made clear(er) in the paper (and I don’t get an answer after reading the paper):

1. Maybe I missed it somewhere, but what is exactly the “oracle” that is referred to throughout this paper? Is it hidden in the VoX algorithm?

2. Even with linear loss struction, the regret still has to be $\Omega(\sqrt{A})$, is this a common result in the low-rank MDP literature, or unique due to the adversarial losses (and bandit feedback)?

3. In Thm. 4.1, is $\delta$ the failure probability for the regret guarantee? In other words, is Thm. 4.1 stating a high-probability regret bound? If so, why is the way of stating a high-prob. bound in Thm. 4.1 different from that in Thm. 5.1? If not, then the use of notation $\delta$ is not consistent in the paper.

4. What’s the difficulty of obtaining regret bounds against adaptive adversaries, without knowing the loss feature map? In section 5, the sentence “Given the difficulty of this setting, we make the additional assumption” fails to yield a smooth transition, as I don’t get the connection between the difficulty (it’s not even clear what it is) and the need for the assumption.

5. Does the difficulty lie in obtaining high-prob regrets against oblivious adversaries? Is it true that with the help of the known loss feature, one can establish high-prob. regret bound, which easily implies regret bounds against adaptive adversaries?

6. If algorithm 4 is mostly from other papers and there’s no novel modification, the authors may consider moving it to appendix to save some space in the main body for more informative messages.

7. In line 10 of algorithm 1, “Define $\sum_h^t = \sum$”, anything missing before the $=$?

8. Sketched proofs in the main body for deriving regret bounds would be very helpful, but I fully understand it due to the page limit.

9. If a reference has been accepted to a conference, the authors may want to cite the conference version rather than arXiv, unless there's a significant update.

I will consider increasing my rating to 6 and supporting acceptance if the authors can address these questions so that I can have a clearer understanding of the results.

**Questions:**

Questions here are mainly for curiosity and will not affect my recommendation/rating much.

1. Typically, such two-stage design (initial exploration for warm-up + standard online learning) may not give rate-optimal guarantee. In adversarial linear MDPs, [3] performs estimation on-the-fly to get the first rate-optimal regret. I was wondering what would be the difficulty of doing similar things here (and enjoying better regrets). Could the authors point out some other places where the sub-optimality comes from?

2. In Thm. 4.1, the regret scales with the size of the function class. I think it makes some sense but any similar results in the literature?

References

[1] Cheng, Yuan, Ruiquan Huang, Yingbin Liang, and Jing Yang. "Improved Sample Complexity for Reward-free Reinforcement Learning under Low-rank MDPs." In The Eleventh International Conference on Learning Representations. 2022.

[2] Mhammedi, Zakaria, Adam Block, Dylan J. Foster, and Alexander Rakhlin. "Efficient Model-Free Exploration in Low-Rank MDPs." arXiv preprint arXiv:2307.03997 (2023).

[3] Liu, Haolin, Chen-Yu Wei, and Julian Zimmert. "Towards Optimal Regret in Adversarial Linear MDPs with Bandit Feedback." In The Twelfth International Conference on Learning Representations. 2023.

**Limitations:**

In the checklist, it says "The paper discuss the limitations of the work in the discussion section", but I didn't find a section named "discussion".

====== update after author-reviewer discussion ======

During the initial rebuttal and the discussion phase, the authors addressed my clarification question, so as of now I increase my score from 5 to 6

---

> ### Author Rebuttal · Authors · 2024-08-06
>
> Thank you for your support and valuable feedback. We will adopt your writing suggestions in our future version.
>
> Your questions about $poly(A)$ dependence in regrets and the difficulty of adaptive adversaries are explained in **Q1** and **Q3** of the global response. We answer your other questions below.
>
> **Q1**: *What is exactly the “oracle” that is referred to throughout this paper? Is it hidden in the VoX algorithm?*
>
>
> **A**: Yes, the oracle is hidden in VoX. A call to VoX requires a polynomial number of calls to a min-max optimization Oracle over a class of value functions to learn a good representation; the Oracle is required by the RepLearn subroutine within VoX. We note that this Oracle is by now standard and has been used in many other works such as  [Mhammedi et al., 2023;  Zhang et al.,2022b].
>
> **Q2**: *In Thm. 4.1, is $\delta$ the failure probability for the regret guarantee? In other words, is Thm. 4.1 stating a high-probability regret bound?*
>
> **A**: Thank you for pointing it out. This is just a typo in the statement of the theorem. The result in Theorem 4.1 is in expectation and $\delta$ should be replaced by $poly(1/T)$.
>
> **Q3**: *Does the difficulty of adaptive adversaries lie in obtaining high-prob regrets against oblivious adversaries? Is it true that with the help of the known loss feature, one can establish high-prob. regret bound, which easily implies regret bounds against adaptive adversaries?*
>
> **A**:   The situation is a little more subtle than this. Even with a known feature loss, Algorithm 2 would still fail against an adaptive adversary because the estimation of the average $Q$-functions within an epoch can no longer be reduced to a standard least-squares regression problem with i.i.d.~data; see the answer to **Q3** in the general response. Instead, the algorithm in Section 5 aims at estimating the average $Q$-functions, not in a least-squares sense, but in expectation over roll-ins using policies in $\Psi_{\texttt{span}}$; the ``in-expectation'' estimation task is in a sense easier. Here, $\Psi_{\texttt{span}}$ is a set of policies with good coverage over state-action pairs. More specifically, $\Psi_{\texttt{span}}$ is required to be such that $\\{ \mathbb{E}^{\pi}[\phi^{\texttt{loss}}(x_h, a_h)]   |  \pi \in \Psi_{\texttt{span}} \\}$ essentially covers all directions in $\mathbb{R}^d$. For this reason, we need $\phi_{\texttt{loss}}$ to compute $\Psi_\texttt{span}$.
>
>
> **Q4**: *In line 10 of algorithm 1, Define $\Sigma_h^t = \sum$, anything missing before the $=$ ?*
>
> **A**: $\Sigma_h^t$ is the covariance matrix of time $t$ and step $h$ whose definition is on the right-hand side of ``$=$''. The $\sum$ on the right-hand side is the summation notation.
>
> **Q5**: *In adversarial linear MDPs, [3] performs estimation on-the-fly to get the first rate-optimal regret. I was wondering what would be the difficulty of doing similar things here (and enjoying better regrets). Could the authors point out some other places where the sub-optimality comes from?*
>
> **A**: Indeed, the suboptimal regret of Algorithm 1 is due to the two-stage design. The online occupancy estimation technique in [3] relies crucially on the transition feature being known. It is unclear how to extend this to the low-rank setting with unknown features. Getting the optimal regret is still a challenging open problem.
>
> **Q6**: *If a reference has been accepted to a conference, the authors may want to cite the conference version rather than arXiv, unless there's a significant update.*
>
> **A**:  Thank you for the suggestion. For the case of [Mhammedi et al., 2023] in particular, the paper has significant updates on arXiv compared to their camera-ready NeurIPS version (removes the reachability assumption with different analysis). Thus, we cite both versions in our paper. We will change other references to the conference version.
>
> **Q7**:  *In Thm. 4.1, the regret scales with the size of the function class. I think it makes some sense but any similar results in the literature?*
>
> **A**:  For low-rank MDP, the $\log(|\Phi|)$ in Theorem 4.1 matches the best-known result in the stochastic setting [Mhammedi et al., 2023; Zhang et al., 2022b]. Most papers on low-rank MDP have worse dependence $\log(|\Phi||\Upsilon|)$ under the model-based assumption (e.g. [Agarwal et al., 2020; Uehara et al., 2022]).
>
>
> Moreover, the logarithmic dependence on the size of the function class is very common in RL with general function approximation. For example, in the general decision estimation coefficient (DEC) framework, such dependence appears in both model-based (Theorem 3.3 in [Foster et al., 2021]) and model-free (Theorem 2.1/Corollary 2.1 in [Foster et al., 2023]) algorithms.
>
> **References**:
>
> [Mhammedi et al., 2023]  Mhammedi, Z., Block, A., Foster, D. J., and Rakhlin, A. (2023). Efficient model-free exploration in low-rank mdps. arXiv preprint arXiv:2307.03997
>
> [Zhang et al., 2022b] Zhang, X., Song, Y., Uehara, M., Wang, M., Agarwal, A., and Sun, W. (2022b). Efficient reinforcement learning in block mdps: A model-free representation learning approach. ICML.
>
>
> [Agarwal et al., 2020] Agarwal, A., Kakade, S., Krishnamurthy, A., and Sun, W. (2020). Flambe: Structural complexity and representation learning of low rank mdps. NeurIPS.
>
>
> [Uehara et al., 2022] Uehara, M., Zhang, X., and Sun, W. (2022). Representation learning for online and offline rl in low-rank mdps. ICLR.
>
> [Foster et al., 2021] Foster, D. J., Kakade, S. M., Qian, J.,  Rakhlin, A. (2021). The statistical complexity of interactive decision making. arXiv preprint arXiv:2112.13487.
>
> [Foster et al., 2023]  Foster, D. J., Golowich, N., Qian, J., Rakhlin, A., Sekhari, A. (2023). Model-free reinforcement learning with the decision-estimation coefficient. NIPS.

---

> > ### Comment · Reviewer_3DuN · 2024-08-07
> > **Regarding Q2 and Thm. 4.1**
> >
> > I thank the authors for the response. How Thm 4.1 should be corrected is still unclear to me from the response. Could the authors state the corrected Thm 4.1 in this thread? Also, what is the role of $\\delta$ (confidence parameter) here? What happens if we make it very small (close to 0) or large (close to 1)?

---

> ### Author Response · Authors · 2024-08-09
> **Statement of Theorem 4.1**
>
> Thank you for your comments. The statement of Theorem 4.1 should read as follows:
>
> **Theorem 4.1.** *Let $\delta\in(0,\frac{1}{T})$ be given and suppose Assumption 2.1. and Assumption 2.2. hold. Then, for $T =\text{poly}(A,H,d, \log(|\Phi|))$ sufficiently large, Algorithm 2 guarantees $Reg_T \leq \text{poly}(A,H,d, \log(|\Phi|T)) \cdot T^{5/6}$ against an oblivious adversary, where $Reg_T :=E_{\rho^{1:T}}[\sum_{t=1}^T E_{\pi^t\sim \rho^t}[V^{\pi^t}(x_1;\ell^t)] -\sum_{t=1}^T V^{\pi}(x_1;\ell^t) ]$.*
>
>
> Note that $Reg_T$ in this statement is a deterministic quantity; there is no randomness. The $\delta$ in the submission was just a typo.
>
> We now explain why there is no $\delta$ in the statement above even though the analysis of Theorem 4.1 (in particular Lemma G.4 and G.5) have a $\delta$.  The current analysis of Algorithm 2 implies that under the setting of Theorem 4.1, we have that for any $\delta\in(0,1)$, there is an event $\mathcal{E}(\delta)$ with probability at least $1-\delta$ under which
> $\widehat{Reg}\_T \leq \text{poly}(A,H,d, \log(|\Phi|/\delta)) \cdot T^{5/6}$, where $\widehat{Reg}\_{T} := \sum\_{t=1}^T E_{\pi^t \sim \rho^t}[V^{\pi^t}(x\_1;\ell^t)] -\sum\_{t=1}^T V^{\pi}(x\_1;\ell^t);$ Note that the randomness here comes from $\rho^{1:T}$.
>
> Now, since $Reg_T = E_{\rho^{1:T}}[\widehat{Reg}_T]$ and $\widehat{Reg}_T \leq H T$, we have
>
> $Reg\_T =P[\mathcal{E}(1/T) ] E\_{\rho^{1:T}}\left[\widehat{Reg}\_T\mid \mathcal{E}(1/T) \right]+ P[\mathcal{E}(1/T)^c ] E\_{\rho^{1:T}}\left[\widehat{Reg}\_T\mid \mathcal{E}(1/T)^c \right]\leq  \text{poly}(A,H,d, \log(|\Phi|/\delta)) \cdot T^{5/6}  + H,$ where we used that $P[ \mathcal{E}(1/T)^c]\leq 1/T$ and the bound on $\widehat{Reg}_T$ under $\mathcal{E}(1/T)$.
>
> Note that the high probability bound on $\widehat{Reg}_T$ above is still not good enough for an adaptive adversary; see the answer to Q3.
>
> We hope this clarifies things. Please let us know if you have any more questions.

---

> > ### Comment · Reviewer_3DuN · 2024-08-09
> >
> > I thank the authors for stating the corrected theorem and walking me through the proof sketch. I now understand that Theorem 4.1 is meant to state an in-expectation upper bound only, and $\\delta$ is just the probability with which some good event doesn't hold.
> >
> > I don't have any other questions as of now. As I promised, I am increasing my score from 5 to 6.

---

### Official Review · Reviewer_rtoB · 2024-07-10

**Soundness:** 3
**Presentation:** 3
**Contribution:** 3
**Rating:** 6
**Confidence:** 3

**Summary:**

This paper studies adversarial Low-Rank MDPs with unknown transition and bandit feedback. The authors give three main results, targeting either tighter regret or computational efficiency. The authors show that the linear structure of the reward function is necessary for the case of bandit feedback to achieve regret without dependence on the number of states.

**Strengths:**

1. The paper addresses a novel and challenging problem of adversarial Low-Rank MDPs with unknown transition and bandit feedback. The problem is well-motivated and has practical implications in real-world applications.
2. The theoretical results are strong and the analyses are sound. The authors provide a comprehensive analysis of the proposed algorithms and prove the theoretical guarantees. The authors combine several techniques to handle the unknown transition and bandit feedback, which is non-trivial.
3. The paper provides a lower bound to show we could not gain too much from a low-rank transition structure compared with tabular MDPs. This lower bound is insightful and provides a clear motivation for the linear reward structure in the bandit feedback setting.

**Weaknesses:**

1. The computational cost of the proposed algorithms is huge, even for the oracle efficient algorithms. The authors should provide more discussions on the computational complexity of the proposed algorithms.
2. Though this paper is the first work to study adversarial Low-Rank MDPs with unknown transition and bandit feedback, the $O(T^{5/6})$ regret bound is not very satisfying. Moreover, the results have a linear dependence on the number of actions, which is not ideal.

**Questions:**

1. Is polynomial dependence on the number of actions necessary? Can the authors provide some intuition about the reason or discuss the possibility of improving the dependence on the number of actions?

---

> ### Author Rebuttal · Authors · 2024-08-06
>
> Thank you for your support and valuable feedback. Your question about $poly(A)$ dependence in regrets is explained in **Q1** of the global response. We answer your other questions below.
>
>
>
> **Q1**: *The authors should provide more discussions on the computational complexity of the proposed algorithms.*
>
> **A**: The main result in [Mhammedi et al., 2023] guarantees that the number of calls to the optimization Oracle within VoX is polynomial in $d$, $A$, $\log |\Phi|$, and $1/\epsilon$. Due to this, VoX is an Oracle-efficient algorithm. Since our algorithm (Algorithm 2 in this context) makes a single call to VoX, it automatically inherits its Oracle efficiency. All the other steps in Algorithm 2 can be performed computationally efficiently; in fact, the most computationally expensive step is the call to VoX.
>
> Our Algorithm 1 is generally computationally inefficient. The computational complexity scales with $|\Pi|$ because we need to maintain exponential weights over policy class $\Pi$. For the linear policy class defined in Line 235, the computational complexity of Algorithm 1 has order $|\Phi| T^d$.
>
> [Mhammedi et al., 2023]  Mhammedi, Z., Block, A., Foster, D. J., and Rakhlin, A. (2023). Efficient model-free exploration in low-rank mdps. arXiv preprint arXiv:2307.03997

---

> > ### Comment · Reviewer_rtoB · 2024-08-09
> > **Thank you for your response.**
> >
> > I thank the authors for their response and have no further questions.

---

### Official Review · Reviewer_GQHM · 2024-07-11

**Soundness:** 3
**Presentation:** 3
**Contribution:** 3
**Rating:** 6
**Confidence:** 4

**Summary:**

This work initiates the study on learning adversarial low-rank MDPs with bandit feedback. The authors propose an inefficient algorithm with $T^{2/3}$ expected regret, and an oracle-efficient algorithm with $T^{5/6}$ expected regret. Further, the authors also show an oracle-efficient algorithm with $T^{5/6}$ high probability regret.

**Strengths:**

1.	This work further advances the understanding of learning adversarial MDPs with general function approximation and the algorithmic designs are of interest.

**Weaknesses:**

1.	The first algorithm is generally hard to be implemented due to its computational intractability and thus might be less appealing to practitioners in the area. Besides, the algorithmic design of the first algorithm is similar to those of [1,2] in the sense that learning adversarial MDPs is somewhat reduced to learning adversarial bandits by running exponential weights over the policy set of the corresponding MDP. I would suggest the authors give more comparisons and discussions between the design of the first algorithm and the algorithms in [1,2].

[1] Kong et al. Improved Regret Bounds for Linear Adversarial MDPs via Linear Optimization. TMLR, 24.

[2] Liu et al. Towards Optimal Regret in Adversarial Linear MDPs with Bandit Feedback. ICLR, 24.

**Questions:**

1.	In Eq. (12), log-barrier instead of the common Shannon entropy regularizer is used in FTRL. Is this due to that it is not feasible to bound the stability term in the analysis of FTRL when using Shannon entropy caused by the possibly very large magnitude of the constructed $Q$-function estimates?
2.	Intuitively, what is the reason that Algorithm 2 still needs to operate in epochs after the phase of reward-free exploration?

**Limitations:**

Not applicable.

---

> ### Author Rebuttal · Authors · 2024-08-06
>
> Thank you for your support and valuable feedback. Your question about why we use log-barrier is explained in **Q2** of the global response. We answer your other questions below.
>
>
> **Q1**: *The first algorithm is similar to learning adversarial MDPs by reducing to learning adversarial bandits.  I would suggest the authors give more comparisons and discussions between the design of the first algorithm and the algorithms in linear MDPs [1,2].*
>
> **A**: Our Algorithm 1 takes a different approach from reducing learning MDPs to linear bandits.
>
> In a linear MDP with a known feature map $\phi$ and a linear loss $\ell_h(x,a) = \phi(x,a)^\top g_h$, we have:
> $$V_1^\pi(x_1, \ell) = \mathbb{E}^\pi\left[\sum_{h=1}^H \ell_h(x_h, a_h)\right] = \mathbb{E}^\pi\left[\phi(x_h, a_h)\right]\sum_{h=1}^H g_h.$$
>
> This means that the linear MDP problem can be reduced to a linear bandit instance where the action set is $\\{\mathbb{E}^\pi\left[\phi(x_h, a_h)\right]: \forall \pi \\}$ and $\sum_{h=1}^H g_h$ is the hidden loss vector. This action set can be estimated by estimating occupancy measures; is the approach taken by [1,2]. With this, [1,2] use the standard linear bandit loss estimator with estimated actions.
>
> The reduction to linear bandits in [1,2] relies crucially on the linearity of the loss functions. In contrast, our Algorithm 1 does not require linear losses. Instead, it only relies on the linear structure of the transitions (i.e. the low-rank structure).
> Moreover, since the feature maps are unknown in our setting, we cannot use the standard linear bandit loss estimator as in [1,2]. To address this, Algorithm 1 learns a representation in an initial phase, which is then used to compute a new loss estimator carefully designed for low-rank MDPs.
>
> **Q2**: *Why Algorithm 2 still need to operate in epochs after the phase of reward-free exploration?*
>
> **A**: The role of the reward-free phase is to learn a policy cover; a set of policies with good coverage over the state space. In the non-adversarial setting, once such a policy set is computed, finding a near-optimal policy is relatively easy. However, in the adversarial setting, the losses change across rounds and the algorithm needs to constantly estimate the $Q$-functions to play good actions. The algorithm uses epoch for this estimation task, where the larger the epoch (the more episodes in an epoch) the more accurate are the estimated $Q$-functions.

---

> > ### Comment · Reviewer_GQHM · 2024-08-10
> >
> > Thanks for your feedback. Could you please elaborate further on the details of the results if the negative entropy instead of log-barrier regularizer is used? What would be the concrete dependence of the results on $T$ and $A$ if negative entropy is employed?

---

> > > ### Author Response · Authors · 2024-08-12
> > > **Use of the negative entropy vs the log-barrier**
> > >
> > > Thank you for your comments. After a more careful inspection, we found that using the negative entropy regularizer would lead to a better dependence in $A$ while keeping the dependence in $T$ unchanged compared with the log-barrier regularizer. As mentioned in the rebuttal, using the negative entropy regularizer requires $\eta |\widehat{Q}\_h^{(k)}(x, a)|\le 1$ for any $k,h,x,a$. This prevents one from choosing $\eta$ optimally for the standard loss estimator in the linear MDP setting (due to the large magnitude of the estimator). However, the estimator we use for the low-rank setting in Algorithm 2 has the property that $\widehat{Q}\_h^{(k)}(x,a) \le H\sqrt{d}$ for any $x,a,k,h$. This implies that the optimal choice of $\eta$ in our setting (which satisfies $\eta=1/\text{poly}(T)$) is well within the range implied by the constraint $\eta |\widehat{Q}\_h^{(k)}(x, a)|\le 1$. The same applies in the setting of Algorithm 3 where $\widehat{Q}\_h^{(k)}(x,a) \le 8Hd^2$.
> > >
> > >
> > > We would also like to correct a typo in our current proof. In the current analysis of the log-barrier (Lines 495 and 554), the penalty term is missing a factor of $A$. According to Lemma G.1, the correct penalty term should be $\frac{N\_{reg} A \log(T/\gamma)}{\eta}$ instead of $\frac{N\_{reg} \log(T/\gamma)}{\eta}$. On the other hand, for negative entropy, the penalty term is $\frac{N\_{reg} \log(A)}{\eta}$ from Lemma G.6. Thus, since the stability terms for both the log-barrier and the negative entropy are the same, this implies that the correct regret for log-barrier has a slightly worse dependence on $A$ compared with the negative entropy. We will change our algorithm and analysis to negative entropy in the revision (our final regret bounds will remain the same up log factors in $A$).

---

> > > > ### Comment · Reviewer_GQHM · 2024-08-12
> > > >
> > > > Thanks for your replies. I have no further questions.

---

### Official Review · Reviewer_XUkU · 2024-07-12

**Soundness:** 3
**Presentation:** 2
**Contribution:** 3
**Rating:** 5
**Confidence:** 3

**Summary:**

The paper explores regret minimization in low-rank MDPs with adversarial bandit loss and unknown transitions. This means that the transition can be expressed as $P(x' \mid x,a ) = \phi^\star (x,a)^T\mu^\star(x')$ for some unknown  $\phi^\star$ and $\mu^\star$ and the loss is also linear in $\phi^\star$. They consider three distinct settings:

1.  **Model-based setting**: Here, the authors assume access to function classes containing $\mu^\star$ and $\phi^\star$. They use some reward-free algorithm to obtain $\mu$ and $\phi$ that approximates the dynamics well, and then run EXP2-style algorithm on some $\epsilon$-cover of the class of linear policies. This algorithm is computationally inefficient and achieves $T^{2/3}$ regret,  with some logarithmic dependencies on the size of the function classes.

2. **Model-free setting**: In this setting, they don't assume access to a class that contains $\mu^\star$. Once again, the algorithm begins with some of-the-shelf exploration phase that aim to find a policy cover. That is, a (small) set of policies that guarantees some sense of minimal reachability to all reachable states. Next, they run an FTRL-style algorithm in each state with log-barrier regularization on blocks of certain size. Within each block, the algorithm plays the policy derived from the FTRL, mixed with a uniform policy from the policy cover. The policy cover's exploration guarantee allows for good least-squares estimation of the Q-function for the policy employed in that block, subsequently used in the FTRL. This achieves $T^{5/6}$ regret against oblivious adversary.

3.  **Model-free setting with Adaptive adversary**: Here, the adversary adjusts their strategy based on the learner's actions, and there is additional assumption that the the loss features are known (and the dynamics feature map is still unknown). The algorithm is similar to the algorithm in 2. but adds some representation learning algorithm on top of the reward-free algorithm which suggested to be beneficial against adaptive adversary. The regret bound obtained in this setting is also of order of $T^{5/6}$.

**Strengths:**

-   Low-rank MDPs and representation learning are important areas of study in reinforcement learning, particularly due to the relevance of representation learning to deep RL. The paper advances research in this regime by extending previous results from full-information settings to bandit feedback.

- The paper introduces three algorithms, each tailored to a distinct setting. These algorithms manage to achieve similar regret guarantees to previous works that dealt with the full-information case (although the settings are not directly comparable). Additionally, the authors provide a compelling justification for why they cannot handle arbitrary (non-linear) loss as in the full-information case by presenting a lower bound that scales with the number of states.

**Weaknesses:**

-   The approach taken in Algorithm 1 is somewhat generic, employing an off-the-shelf reward-free exploration phase followed by an inefficient EXP2 algorithm. Most of the complexity introduced by the unknown feature maps is largely addressed by leveraging existing reward-free algorithms, possibly underplaying the novelty or the specific challenges directly tackled by the new algorithm itself.

-   Section 5 lacks detailed explanations. The section does not adequately clarify what specifically becomes more challenging in the adaptive adversary setting, nor does it thoroughly explain how the spanner policies effectively address these challenges. The description merely states that these policies are necessary *"because the estimation of the Q-functions becomes much more challenging with an adaptive adversary"*. This lack of detail leaves the reader without a clear understanding of the problem mechanics or the solution’s efficacy in this context.

**Questions:**

Could you please clarify how (41) is obtained from Lemma G.5? (41) bounds the average distance, while, to my understanding, G.5 bounds the distance on the data. It feels like some argument on the generalization gap is missing.
- Is there a specific reason to use log-barrier and not negative entropy regularization in algorithm 2?

**Limitations:**

As far as I'm concerned, the authors address most of their limitations.

---

> ### Author Rebuttal · Authors · 2024-08-06
>
> Thank you for reviewing our paper and for your valuable feedback. Your questions about the use of the log-barrier and the difficulty of dealing with an adaptive adversary are explained in **Q2** and **Q3** of the global response. We answer your other questions below.
>
>
> **Q1**: *In Algorithm 1, most of the complexity introduced by the unknown feature maps is largely addressed by leveraging existing reward-free algorithms, possibly underplaying the novelty or the specific challenges directly tackled by the new algorithm itself.*
>
> **A**: The main novelty of Algorithm 1 lies in the design of the loss estimators; in a low-rank MDP where the feature map is unknown, standard loss estimators used for linear MDPs fail. The challenge here is that the loss estimators have to be carefully designed to account for any errors in the learned representation from Phase 1. This is a non-trivial challenge that existing methods before our paper do not address.
>
>
> **Q2**: *Could you please clarify how (41) is obtained from Lemma G.5? It feels like some argument on the generalization gap is missing.*
>
> **A**: You are right. Equation (41) follows from Lemma G.4 and Lemma G.5. We will add the reference to Lemma G.4.

---

> > ### Comment · Reviewer_XUkU · 2024-08-08
> >
> > Thank you for your rebuttal. I have read through all the other reviews and your responses. At this point, I do not have any further questions. I will reconsider my score based on the discussions and additional inputs provided during the current phase and the next phase of the review process.

---

### Author Rebuttal · Authors · 2024-08-06

## **Global Response**

We thank all the reviewers for their valuable feedback. We would like to clarify several common concerns here.

**Q1** : *The $poly(A)$ dependence in the regret bound looks not ideal. Is it common in low-rank MDP literature?*

**A**: We would like to point out that poly-dependence of action set size $A$ is unavoidable anyway even in non-adversarial low-rank MDPs with linear loss; this follows the lower bound from Theorem 4.2 and Appendix B of [Zhao et al. 2024].


[Zhao et al. 2024]  Zhao, C., Yang, R., Wang, B., Zhang, X., and Li, S. (2024). Learning adversarial low-rank markov decision processes with unknown transition and full-information feedback. NeurIPS.


**Q2**: *Why we use log-barrier?*

**A**: For the $Q$-function estimator $\widehat{Q}$ with learning rate $\eta$, negative entropy regularization requires $\eta \widehat{Q} \ge -1$ to ensure the stability term in the regret is small. Using the log barrier, we can get rid of this constraint, leading to more flexibility in the choice of $\eta$ and ultimately a better regret bound (compared to the negative entropy) after tuning $\eta$.

The price of using the log-barrier is a poly-dependence in the number of actions $A$. However, as discussed in **Q1**, such dependence is unavoidable for low-rank MDPs.



**Q3**:  *Section 5 lacks detailed explanations. Why does the estimation of the Q-functions become much more challenging against an adaptive adversary without knowing the loss feature map?*


**A**:  Our algorithm for oblivious adversaries works in epochs, where at each epoch $k$ the algorithm commits to a policy $\widehat{\pi}^{(k)}$ for $N_{reg}$ episodes. The algorithm then uses these trajectories to compute an estimate $\widehat{Q}^{(k)}$ of the average $Q$-function $\frac{1}{N_{reg}} \sum_{t \text{ in epoch } k}Q^{\widehat{\pi}^{(k)}}_h(\cdot,\cdot ;\ell^t)$. In the case of an oblivious adversary, this estimation problem can be reduced to a standard regression problem with i.i.d. data. This is possible because the regression target is an average of $Q$-functions over $t$ episodes in epoch $k$, and this target is invariant to any permutation of the episodes $t$ thanks to the fact that losses are oblivious. And so, by randomly shuffling the episodes in epoch $k$, one can reduce the regression problem to an i.i.d. one. When dealing with an adaptive adversary, the order of the episodes within an epoch matters and the average $Q$-function targets are no longer invariant to permutations of the episodes. For this reason, we cannot rely on standard regression guarantees with i.i.d. data.

---

### Comment · Area_Chair_MUgc · 2024-08-12

Dear Reviewers,

The deadline for the reviewer-author discussion period is approaching. Thank you for already providing responses to the rebuttal. If you have any additional questions, please discuss them with the authors at your earliest convenience.

Best wishes, AC

---

### Decision · Program_Chairs · 2024-09-25

**Decision:**

Accept (poster)

**Comment:**

The paper explores regret minimization in low-rank MDPs with adversarial bandit loss and unknown transitions. Several regret bounds are obtained under different settings. It is the first attempt to low-rank MDPs in the adversarial environment, and involves a nontrivial reduction to linear bandit (where the reduction differs from previous works). All reviews are positive, and during the discussion period all reviewers and I have reached a consensus that the manuscript has significant contributions. Concerns on Theorem 4.1 and on different regularizers have been adequately addressed during the rebuttal. I therefore recommend acceptance.